



# Source apportionment of carbonaceous aerosols in Xi'an, China: insights from a full year of measurements of radiocarbon and the stable isotope $^{13}$C

Haiyan Ni[1,2,3,4], Ru-Jin Huang[2,3*], Junji Cao[3,5*], Ting Zhang[3], Meng Wang[3], Harro A.J. Meijer[1], Ulrike Dusek[1]

[1]Centre for Isotope Research (CIO), Energy and Sustainability Research Institute Groningen (ESRIG), University of Groningen, Groningen, 9747 AG, the Netherlands
[2]State Key Laboratory of Loess and Quaternary Geology (SKLLQG), Institute of Earth Environment, Chinese Academy of Sciences, Xi'an, 710061, China
[3]Key Laboratory of Aerosol Chemistry & Physics (KLACP), Institute of Earth Environment, Chinese Academy of Sciences, Xi'an, 710061, China
[4]University of Chinese Academy of Sciences, Beijing, 100049, China
[5]Institute of Global Environmental Change, Xi'an Jiaotong University, Xi'an 710049, China

*Correspondence to*: Ru-Jin Huang (rujin.huang@ieecas.cn); Junji Cao (cao@loess.llqg.ac.cn)

**Abstract.** Sources of organic carbon (OC) and elemental carbon (EC) in Xi'an, China are investigated based on one-year radiocarbon and stable carbon isotope measurements. The radiocarbon results demonstrate that EC is dominated by fossil sources throughout the year, with a mean contribution of 83 ± 5 % (7 ± 2 µg m$^{-3}$). The remaining 17 ± 5 % (1.5 ± 1 µg m$^{-3}$) is attributed to biomass burning, with higher contribution in the winter (~24 %) compared to the summer (~14 %). Stable carbon isotopes of EC ($\delta^{13}C_{EC}$) are enriched in winter (-23.20 ± 0.35 ‰) and depleted in summer (-25.94 ± 0.46 ‰), indicating the influence of coal combustion in winter and liquid fossil fuel combustion in summer. By combining radiocarbon and stable carbon signatures, relative contributions from coal combustion and liquid fossil fuel combustion are estimated as 45 % (29–58 %, interquartile range) and 31 % (18–46 %) in winter, respectively, whereas in other seasons more than one half of EC are from liquid fossil combustion. In contrast with EC, the contribution of non-fossil sources to OC is much larger, with an annual average of 54 ± 8 % (12 ± 10 µg m$^{-3}$). Clear seasonal variations are seen in OC concentrations both from fossil and non-fossil sources, with maxima in winter and minima in summer, because of unfavourable meteorological conditions coupled with enhanced fossil and non-fossil activities in winter, mainly biomass burning and domestic coal burning. $\delta^{13}C_{OC}$ exhibited similar values with $\delta^{13}C_{EC}$, and showed strong correlations ($r^2 = 0.90$) in summer and autumn, indicating similar source mixtures with EC. In spring, $\delta^{13}C_{OC}$ is depleted (1.1–2.4 ‰) compared to $\delta^{13}C_{EC}$, indicating the importance of secondary formation of OC (e.g., from volatile organic compound precursors) in addition to primary sources. Modelled mass concentrations and source contributions of primary OC are compared to the measured mass and source contributions. There is strong evidence that both secondary formation and photochemical loss processes influence the final OC concentrations.





## 1 Introduction

Carbonaceous aerosols are an important component of fine particulate matter (PM$_{2.5}$, particles with aerodynamic diameter <2.5 μm), constituting 20–50 % of PM$_{2.5}$ mass in many urban areas in China (Cao et al., 2007; Huang et al., 2014a; Tao et al., 2017). They have adverse impacts on air quality (Watson, 2002), human health (Nel, 2005; Cao et al., 2012; Lelieveld et

al., 2015), and climate (Chung and Seinfeld, 2002; Bond et al., 2013). Carbonaceous aerosols are operationally divided into organic carbon (OC) and elemental carbon (EC). EC is exclusively emitted as primary aerosols from incomplete combustion of biomass (e.g., wood, crop residues, and grass) and fossil fuels (e.g., coal, gasoline, and diesel). OC includes both primary and secondary OC from non-fossil (e.g., biomass burning, biogenic emissions, and cooking) and fossil sources, where primary OC is emitted directly and secondary OC is formed in the atmosphere via atmospheric oxidation of volatile organic

compounds (Jacobson et al., 2000; Pöschl, 2005; Hallquist et al., 2009). Carbonaceous aerosols in PM$_{2.5}$ have been identified as critical contributors to severe air pollution events (Cao et al., 2003; Huang et al., 2014a; Elser et al., 2016; Liu et al., 2016a), but their sources and evolution remain poorly characterized. A better understanding of OC and EC sources is important for the mitigation of particulate air pollution.

Radiocarbon ($^{14}$C) analyses of OC and EC allow a quantitative and unambiguous measurement of their fossil and non-fossil

contributions, based on the fact that emissions from fossil sources are $^{14}$C-free, whereas non-fossil emissions contain the contemporary $^{14}$C content (e.g., Szidat, 2009; Dusek et al., 2013, 2017). The $^{14}$C content of an aerosol sample is usually reported relative to an oxalic acid standard, and expressed as fraction modern (F$^{14}$C). $^{14}$C content of the standard is related to the unperturbed atmosphere in the reference year of 1950 (Mook and van der Plicht, 1999; Reimer et al., 2004):

$$F^{14}C = \frac{(^{14}C/^{12}C)_{sample}}{(^{14}C/^{12}C)_{1950}}, \qquad (1)$$

where the $^{14}$C/$^{12}$C ratio of the sample and standard are both corrected for machine background and normalized for fractionation to δ$^{13}$C = -25 ‰. Aerosol carbon from living material should have F$^{14}$C ~1 in an undisturbed atmosphere, and carbon from fossil sources has F$^{14}$C = 0. Previous $^{14}$C measurements of carbonaceous aerosols in China found that EC in urban areas is dominated by fossil sources, which account for 66–87 % of EC mass, whereas OC is more influenced by non-fossil sources with fossil sources accounting for only 35–67 % (Table 1). Despite a fair number of $^{14}$C studies in China in

recent years, only two $^{14}$C datasets so far reported annual results and seasonal variations (Zhang et al., 2014b, 2017).

In addition to $^{14}$C source apportionment, analysis of the stable carbon isotope ($^{13}$C, expressed as δ$^{13}$C) can provide further information regarding sources and atmospheric processing of carbonaceous aerosol (Bosch et al., 2014; Kirillova et al., 2014b; Andersson et al., 2015; Masalaite et al., 2017). Different emission sources have their distinct source signature: the δ$^{13}$C signature of carbonaceous aerosols from coal combustion is less depleted (δ$^{13}$C~ -25 ‰ to -21 ‰) than from liquid

fossil sources (-28 ‰ to -24 ‰) and C3 plants (-35 ‰ to -24 ‰) (Andersson et al. (2015) and references therein).



Complementing $^{14}$C source apportionment with $^{13}$C measurements allows a better constraint of the contribution of different emission sources to carbonaceous aerosols (Kirillova et al., 2013, 2014a; Bosch et al., 2014; Andersson et al., 2015; Winiger et al., 2015, 2016; Bikkina et al., 2016, 2017; Yan et al., 2017). For example, EC is inert to chemical or physical transformations, thus the $\delta^{13}C_{EC}$ preserves the signature of emission sources (Huang et al., 2006; Andersson et al., 2015;

Winiger et al., 2015, 2016). EC from fossil sources (e.g., coal combustion, liquid fossil fuel burning) can be first separated from biomass burning by $F^{14}C$ of EC. Further, $\delta^{13}C$ of EC allows separation of fossil sources into coal and liquid fossil fuel burning (Andersson et al., 2015; Winiger et al., 2015, 2016), as they have their own distinct source signatures (Table S1). $\delta^{13}C$ signatures of OC are not only determined by the source signatures but also influenced by atmospheric processing. During formation of secondary organic aerosol (SOA), molecules depleted in heavy isotopes are expected to react faster,

leading to SOA depleted in $^{13}$C compared to its gaseous precursors, if the precursor is only partially reacted (Anderson et al., 2004; Irei et al., 2006; Fisseha et al., 2009). On the other hand, photochemical aging of particulate organics leads to $^{13}$C enrichment in the remaining aerosols due to a fractionation in the aerosol $^{13}$C (Irei et al., 2011; Pavuluri and Kawamura, 2016).

We present, to the best of our knowledge, the first one-year radiocarbon and stable carbon isotopic measurements to

constrain OC and EC sources in China. PM$_{2.5}$ samples were collected at Xi'an (33°29'–34°44' N, 107°40'–109°49' E), one of the most polluted megacities in the world (Zhang and Cao, 2015a). In this study, contributions from fossil and non-fossil sources to OC and EC are quantified by radiocarbon measurements. Fossil sources of EC are further distinguished into coal and liquid fossil fuel combustion by complementing radiocarbon with stable carbon signature. Sources and atmospheric processing of OC are qualitatively assessed by stable carbon signature. Further, mass concentrations and source

contributions of primary OC are estimated based on the apportioned EC and compared with measured OC mass concentrations and source contributions to give insights into OC sources and formation mechanisms.

## 2 Methods

### 2.1 Sampling

Sampling was carried out at Xi'an High-Tech Zone (34.23° N, 108.88° E, ~10 m above the ground), on a building rooftop of

the Institute of Earth Environment, Chinese Academy of Sciences. The sampling site is surrounded by a residential area ~15 km south of downtown and has no major industrial activities. Details about the sampling site can be found elsewhere (Bandowe et al., 2014; Zhang et al., 2014a).

24 h (from 10:00 a.m. to 10:00 a.m. the next day, local standard time [LST]) PM$_{2.5}$ samples were collected every sixth day from 5 July 2008 to 27 June 2009 using a high-volume sampler (TE-6070 MFC, Tisch Inc., Cleveland, OH, USA) operating

30 at 1.0 m$^3$ min$^{-1}$. PM$_{2.5}$ samples were collected on Whatman quartz fiber filters (20.3 cm×25.4 cm, Whatman QM/A, Clifton,



NJ, USA) that were pre-fired at 900 °C for 3 h to remove absorbed organic vapors (Watson et al., 2009; Chow et al., 2010). After sampling, all filters were packed in pre-baked aluminium foils, sealed in polyethylene bags and stored at -18 ºC in a freezer. To be consistent with previous studies, we designated 15 November to 14 March as winter, 15 March to 31 May as spring, 1 June to 31 August as summer, and 1 September to 14 November as autumn, based on the meteorological

characteristics and the typical residential heating period. Fifty-eight $PM_{2.5}$ samples were collected in total, with 13 in spring, 15 in summer, 12 in autumn, and 18 in winter. Six samples with varying $PM_{2.5}$ mass and carbonaceous aerosols loading were selected per season for [14]C analysis. We selected the samples carefully to cover periods of low, medium and high $PM_{2.5}$ concentrations to get samples representative of the various pollution conditions that did occur in each season. The 24 selected samples are highlighted in Fig. S1 with their OC and EC concentrations. In total, there are 48 radiocarbon data,

including 24 for OC and 24 for EC. Details on sample selection for [14]C analysis are presented in Supplemental S1.

### 2.2 Organic carbon (OC) and elemental carbon (EC) measurement

Filter pieces of 0.5 $cm^2$ were used to measure OC and EC by a Desert Research Institute (DRI) Model 2001 Thermal/Optical Carbon Analyzer (Atmoslytic Inc., Calabasas, CA, USA) following the IMPROVE_A (Interagency Monitoring of Protected Visual Environments) thermal/optical reflectance (TOR) protocol (Chow et al., 1993, 2007, 2011). Details of the OC/EC

measurement were described in Cao et al. (2005).

### 2.3 Stable carbon isotope ([13]C) analysis of OC and EC

The stable carbon isotopic composition of OC and EC was determined using a Finnigan MAT 251 mass spectrometer (Bremen, Germany) at the Stable Isotope Laboratory at the Institute of Earth Environment, Chinese Academy of Sciences. For OC, filter pieces were heated at 375 °C for 3 h in the presence of CuO catalyst grains. The evolved $CO_2$ from OC was

collected by a series of cold traps and quantified manometrically. The stable carbon isotopic composition of the $CO_2$ was determined as $\delta^{13}C_{OC}$. The carbon that remained on the filters was combusted at 850 °C for 5 h and quantified as $\delta^{13}C_{EC}$. Samples were analysed at least in duplicate, and all replicates showed differences less than 0.3 ‰. $\delta^{13}C$ values are reported in the delta notation with respect to the international standard Vienna Pee Dee Belemnite (VPDB):

$$\delta^{13}C \ (‰) = \left[ \frac{(^{13}C/^{12}C)_{sample}}{(^{13}C/^{12}C)_{VPDB}} - 1 \right] \times 1000, \tag{2}$$

Details of stable carbon isotope measurements are described elsewhere (Cao et al., 2008, 2011, 2013).

### 2.4 Radiocarbon ([14]C) measurement of OC and EC

***Combustion of OC, EC and standards*** OC and EC were extracted separately by our aerosol combustion system (ACS) (Dusek et al., 2014). In brief, the ACS consists of a combustion tube, where aerosol filter pieces are combusted at different



temperatures in pure $O_2$, and a purification line where the resulting $CO_2$ is isolated and separated from other gases, such as water vapor and $NO_x$. The purified $CO_2$ is then stored in flame-sealed ampoules until graphitization.

OC is combusted by heating filter pieces at 375 °C for 10 min. EC is combusted after complete OC removal. To remove OC completely, water-soluble OC is first removed from the filter by water extraction (Dusek et al., 2014) to minimize charring

of organic material (Yu et al., 2002). Subsequently, most water-insoluble OC is removed by heating the filter pieces at 375 °C for 10 min. Then the oven temperature is increased to 450 °C for 3 min, and in this step a mixture of the most refractory OC and less refractory EC is removed from the filter. The remaining EC is then combusted by heating at 650 °C for 5 min (Zenker et al., 2017).

Two standards with known $^{14}$C content are analyzed as quality control: an oxalic acid standard and a graphite standard. The

standards are directly put on the filter holder of the combustion tube and heated at 650 °C for 10 min. The contamination introduced by the combustion process can be estimated from the deviation of measured values from the nominal values. The contamination is below 1.5 µgC per combustion, which is relatively small compared the samples ranging between 50 and 270 µgC in this study.

*$^{14}$C analysis of OC and EC* Graphitization and AMS measurements were conducted at the Centre for Isotope Research (CIO)

at the University of Groningen. The extracted $CO_2$ is reduced to graphite by reaction with $H_2$ (g) at a molecular ratio $H_2/CO_2$ of 2 using a porous iron pellet as catalyst at 550 °C (de Rooij et al., 2010). The water vapor from the reduction reaction is cryogenically removed using Peltier cooling elements. The yield of graphite is higher than 90 % for samples of >50 µgC. Graphite formed on the iron pellet is then pressed into a 1.5 mm target holder, which is introduced into the AMS system for subsequent measurement. The AMS system (van der Plicht et al., 2000) is dedicated to $^{14}$C analysis, and simultaneously

measures $^{14}$C/$^{12}$C and $^{13}$C/$^{12}$C ratios.

Varying amounts of reference materials covering the range of sample mass are analyzed together with samples in the same wheel of AMS. Two such materials with known $^{14}$C content are used: the oxalic acid OXII calibration material ($F^{14}C$ = 1.3406) and a $^{14}$C-free $CO_2$ gas ($F^{14}C$ = 0). The differences between measured and nominal $F^{14}C$ values are used to correct the sample values (de Rooij et al., 2010) for contamination during graphitization and AMS measurement. The contamination

is typically smaller than 2 µgC (Prokopiou, 2010).

## 2.5 Source apportionment methodology using $^{14}$C

$F^{14}C$ of EC ($F^{14}C_{(EC)}$) was converted to the fraction of biomass burning ($f_{bb}(EC)$) by dividing the conversion factor of 1.10 ± 0.05 for EC (Lewis et al., 2004; Mohn et al., 2008; Palstra and Meijer, 2014), to eliminate the effect from nuclear bomb tests in the 1960s. EC is primarily produced from biomass burning ($EC_{bb}$) and fossil fuel combustion ($EC_{fossil}$), and absolute EC

concentrations from each source can be estimated once $f_{bb}(EC)$ is known:





$$EC_{bb} = EC \times f_{bb}(EC), \tag{3}$$

$$EC_{fossil} = EC - EC_{bb}, \tag{4}$$

Analogously, $F^{14}C$ of OC ($F^{14}C_{(OC)}$) was converted to the fraction of non-fossil ($f_{nf}(OC)$) by dividing the conversion factor of 1.09 ± 0.05 for OC (Lewis et al., 2004; Levin et al., 2010; Zhang et al., 2014b). OC can be apportioned between OC from

non-fossil sources ($OC_{nf}$) and from fossil-dominated combustion sources ($OC_{fossil}$) using $f_{nf}(OC)$:

$$OC_{nf} = OC \times f_{nf}(OC), \tag{5}$$

$$OC_{fossil} = OC - OC_{nf}, \tag{6}$$

A Monte Carlo simulation with 10,000 individual calculations was conducted to propagate uncertainties. For each individual calculation, $F^{14}C_{(OC)}$, $F^{14}C_{(EC)}$, and OC, EC concentrations are randomly chosen from a normal distribution symmetric around

the measured values with the experimental uncertainties as standard deviation. Random values for conversion factors are chosen from a triangular frequency distribution with its maximum at the central value, and is 0 at the lower limit and upper limit. In this way 10,000 different estimation of $f_{bb}(EC)$, $f_{nf}(OC)$, $EC_{bb}$, $EC_{fossil}$, $OC_{nf}$ and $OC_{fossil}$ can be calculated. The derived average represents the best estimate, and the standard deviation represents the combined uncertainties.

### 2.6 Source apportionment of EC using Bayesian statistics

$F^{14}C$ and $\delta^{13}C$ signatures of EC and a mass-balance calculation were used in combination with a Bayesian Markov chain Monte Carlo (MCMC) scheme to further constrain EC sources into biomass burning ($f_{bb}$), liquid fossil combustion ($f_{liq.fossil}$), and coal combustion ($f_{coal}$):

$$F^{14}C_{(EC)} = F^{14}C_{bb} \times f_{bb} + F^{14}C_{liq.fossil} \times f_{liq.fossil} + F^{14}C_{coal} \times f_{coal}, \tag{7}$$

$$f_{bb} + f_{liq.fossil} + f_{coal} = 1, \tag{8}$$

$$\delta^{13}C_{EC} = \delta^{13}C_{bb} \times f_{bb} + \delta^{13}C_{liq.fossil} \times f_{liq.fossil} + \delta^{13}C_{coal} \times f_{coal}, \tag{9}$$

where $f$ represents the fraction of EC mass contributed by a given source, and subscripts denote investigated sources, where "$_{bb}$" denotes biomass burning, "$_{liq.fossil}$" is liquid fossil, and "$_{coal}$" is fossil coal. $F^{14}C_{(EC)}$ is included in this model which allows separating the input from biomass ($f_{bb}$) from fossil sources ($f_{liq.fossil}$ and $f_{coal}$). $F^{14}C_{bb}$ is the $F^{14}C$ of biomass burning (1.10 ± 0.05, the conversion factor for EC in Sect. 2.5). $F^{14}C_{liq.fossil}$ and $F^{14}C_{coal}$ is zero due to the long-time decay. The MCMC

technique takes into account the variability in the source-signatures of $F^{14}C$ and $\delta^{13}C$ (Table S1), where $\delta^{13}C$ introduces a larger uncertainty than $F^{14}C$.

MCMC-driven Bayesian approaches have been recently implemented to account for multiple sources of uncertainties and variabilities for isotope-based source apportionment applications (Parnell et al., 2010; Andersson, 2011). MCMC works by repeatedly guessing the values of the source contributions and find those values, which fit the data best. The initial guesses


are usually poor and are discarded as part of an initial phase known as the burn-in. Subsequent iterations are then stored and used for the posterior distribution. MCMC was implemented in the freely available R software (https://cran.r-project.org/), using the *simmr* package (https://CRAN.R-project.org/package=simmr). Convergence diagnostics were created to make sure the model has converged properly. The simulation for each sample was run with 10,000 iterations, using a burn-in of 1000

steps, and a data thinning of 100.

### 3 Results

#### 3.1 Temporal variation of OC and EC mass concentrations

During the sampling period, extremely high OC and EC mass concentrations were sometimes observed (Fig. S1). OC mass concentrations ranged from 3.3 µg m$^{-3}$ to 67.0 µg m$^{-3}$, with an average of 21.5 µg m$^{-3}$. EC mass concentrations ranged from 2

µg m$^{-3}$ to 16 µg m$^{-3}$, with an average of 7.6 µg m$^{-3}$ (Table S2). OC and EC mass concentrations were comparable to those reported values in previous studies for Xi'an, which had an average of 19.7 ± 10.7 µg m$^{-3}$ (average ± standard deviation) OC and 8.0 ± 4.7 µg m$^{-3}$ EC from March, 2012 to March, 2013 (Han et al., 2016).

OC and EC concentrations showed a clear seasonal variation with higher concentrations in cold period than those in warm period. The differences between winter and summer concentrations were significant ($p<0.05$). The mean winter to summer

concentration ratios were 3 for OC and 1.5 for EC. Similar seasonal trends of OC and EC were also observed in Xi'an, China in earlier studies (e.g., Han et al. (2016) and Niu et al. (2016)).

#### 3.2 Temporal variation of fossil and non-fossil fractions of OC and EC

To investigate the sources of OC and EC, twenty-four samples representing different loadings of carbonaceous aerosols from different seasons were selected for radiocarbon measurement (Supplemental S1, Fig. S2, Table S3). The highest biomass

burning contribution to EC ($f_{bb}$(EC)) of 46 % was detected on 25 January 2009 (Fig. 1(a)). This can be related to enhanced biomass burning emissions indicated by the comparably high biomass-indicative levoglucosan/EC ratio, and relatively low fossil-fuel associated Σhopanes/EC ratio, picene/EC ratio (Fig. S3), along with unfavorable meteorological condition (e.g., substantially low wind speed (~1 m/s) and low temperature (-0.5 °C)). The highest non-fossil contribution to OC ($f_{nf}$(OC)) of 70 % was observed on the same day. Note that 25 January 2009 was the Chinese New Year eve with many fireworks. Since

the influence of fireworks on F$^{14}$C signature is not known yet, the following source apportionment will not include the Chinese New Year eve.

EC is predominantly influenced by fossil sources, with relative contribution of fossil fuel to EC ($f_{fossil}$(EC)) ranging from 71 % to 89 %, with an annual average of 83 ± 5 %. Lower $f_{fossil}$(EC) were observed in winter (77 ± 5 %) compared with other seasons. This is due to the substantial contribution from biomass burning to EC in winter, with a larger $f_{bb}$(EC) in winter (23



± 5 %) than other seasons (14 ± 2 %, 16 ± 1 % and 18 ± 5 % in summer, spring and autumn, respectively; Fig. 1(a)). This is consistent with the evaluated levoglucosan/EC ratios observed in winter (96 ng/µg), 1.6 times higher than that of yearly average (Fig. S3). Lowest $f_{bb}$(EC) in summer (14 ± 2 %) suggests the importance of fossil fuel sources for EC concentrations. Since the residential usage of coal in summer is much reduced compared with other seasons, we can expect

higher contribution from vehicle emissions than coal burning to fossil EC in summer. EC concentrations from fossil fuel ($EC_{fossil}$) varied by a factor of 4, ranging from 3.1 µg m$^{-3}$ to 11.6 µg m$^{-3}$ with a mean of 6.7 ± 2.0 µg m$^{-3}$, which was 4 times higher than averaged biomass-burning EC concentrations ($EC_{bb}$ = 1.5 ± 0.9 µg m$^{-3}$). A stronger variation was observed in the $EC_{bb}$, varying 9-fold from 0.5 µg m$^{-3}$ to 4.7 µg m$^{-3}$ (Table S4, Table S5).

The relative contribution of non-fossil sources to OC ($f_{nf}$(OC)) ranged from 31 % to 66 %, with an annual average of 54 ±

8 %, which is larger than that to EC (yearly average of 17 ± 5 %). Higher $f_{nf}$(OC) was observed in winter (62 ± 5 %) and autumn (57 ± 4 %), compared to summer and spring, when about half of OC was contributed by non-fossil sources (48 ± 3 % and 48 ± 8 %, respectively. Table S5). The lowest $f_{nf}$(OC) of 31 % was detected on 28 April 2009 (Fig. 1(b)), caused by the enhanced fossil emissions indicated by the highest Σhopanes/EC ratios (5 ng/µg. Fig. S3). Averaged OC concentrations from non-fossil sources ($OC_{nf}$) were 12 ± 10 µg m$^{-3}$, ranging from 2.3 µg m$^{-3}$ to 38.6 µg m$^{-3}$. OC concentrations from fossil

sources ($OC_{fossil}$) varied from 3.2 µg m$^{-3}$ to 20.4 µg m$^{-3}$, with an average of 9.0 ± 4.8 µg m$^{-3}$. Clear seasonal variations were seen in OC concentrations both from fossil fuel and non-fossil sources, with maxima in winter ($OC_{fossil}$ = 13.2 ± 6.0 µg m$^{-3}$, $OC_{nf}$ = 23.3 ± 13.3 µg m$^{-3}$) and minima in summer ($OC_{fossil}$ = 5.5 ± 1.0 µg m$^{-3}$, $OC_{nf}$ = 5.1 ± 1.4 µg m$^{-3}$), because of enhanced fossil and non-fossil activities in winter, mainly biomass burning and domestic coal burning (Cao et al., 2009, 2011; Han et al., 2010, 2016).

**3.3 $^{13}$C signature of OC and EC**

The δ$^{13}$C$_{EC}$ preserves the signature of emission sources, as EC is inert to chemical or physical transformations (Huang et al., 2006; Andersson et al., 2015; Winiger et al., 2015, 2016). Major EC sources in Xi'an include biomass burning, coal combustion, and liquid fossil fuel combustion (e.g., diesel and gasoline) (Cao et al., 2005, 2009, 2011; Han et al., 2010; Wang et al., 2016). C3 plants and C4 plants, biomass subtypes, have a different δ$^{13}$C signature. Aerosols from burning C4

plants are more enriched in δ$^{13}$C (-16.45 ± 1.4 ‰) than that of C3 plants (-26.7 ± 1.8 ‰, Table S1). C3 plants are the dominant biomass type (e.g., wood, wheat straw etc.) in North China (Cheng et al., 2013; Cao et al., 2016). This is also evident from our observation that δ$^{13}$C values of the ambient aerosol fall within the range of C3 plants, coal and liquid fossil fuel combustion (Fig. 2).

The annually averaged δ$^{13}$C$_{EC}$ is -24.92 ± 1.14 ‰, varying between -26.50 ‰ and -22.81‰. Considerable seasonal variation

is observed, suggesting a shift among combustion sources. The δ$^{13}$C$_{EC}$ signature for winter (-23.20 ± 0.35 ‰) clearly locates in the δ$^{13}$C range for coal combustion (-23.38 ± 1.3 ‰, Table S1), and is more enriched compared to other seasons. This



indicates a strong influence of coal combustion in winter, but the $^{14}$C values indicate that coal combustion cannot be the only source of EC. Moreover, the $\delta^{13}C_{EC}$ values in winter ranging from -23.72 ‰ to -22.81 ‰ are at the higher (i.e., enriched) end of coal combustion, indicating some additional contributions from C4 plants, such as corn stalk burning. In northern China, large quantities of coal are used for heating during a formal residential "heating season" in winter (Cao et al., 2007), and in

rural Xi'an, burning corn stalk (C4 plant) in "Heated Kang" (Zhuang et al., 2009) is a traditional way for heating in winter (Sun et al., 2017). The most depleted $\delta^{13}C_{EC}$ values in summer (-25.94 ± 0.46 ‰) and spring (-25.40 ± 0.33 ‰) falls into the overlap of liquid fossil fuel emission (-25.5 ± 1.3 ‰) and C3 plant combustion (-26.7 ± 1.8 ‰, Fig. 2), when little or no coal is used for residential heating but has some coal emissions from industries. As the biomass burning contribution to EC in summer and spring is relatively low (14 ± 2% and 16 ± 1%, respectively), we can expect liquid fossil fuel combustion

dominates EC emissions. $\delta^{13}C_{EC}$ signatures in autumn (-25.14 ± 0.66 ‰) fall in the overlapped area of C3 plant, liquid fuel and coal, implying EC is influenced by the mixed sources.

$\delta^{13}C_{OC}$ was in general similar to $\delta^{13}C_{EC}$: it varies from -27.42 ‰ to -23.23 ‰, with an annual average of -25.32 ± 1.19 ‰ (Fig. 2). This range overlaps with C3 plants, liquid fossil and coal combustion. Influence from marine sources (-21 ± 2 ‰; Chesselet et al., 1981; Miyazaki et al., 2011) should be minimal, as Xi'an is a far inland city in China. $\delta^{13}C_{OC}$ shows a

similar seasonal variation pattern as $\delta^{13}C_{EC}$. $\delta^{13}C_{OC}$ is most enriched in winter (-24.13 ± 0.83 ‰), followed by autumn (-24.85 ± 0.79 ‰), summer (-25.73 ± 0.90 ‰), and spring (-26.58 ± 0.57 ‰). In addition to source mixtures, atmospheric processing also influences $\delta^{13}C_{OC}$ (Irei et al., 2006, 2011; Fisseha et al., 2009). In spring, $\delta^{13}C_{OC}$ is much more depleted than $\delta^{13}C_{EC}$ (1.1−2.4 ‰), indicating the importance of secondary formation of OC (e.g., from volatile organic compound precursors) in addition to primary sources (Anderson et al., 2004; Iannone et al., 2010). In summer and autumn 2008, $\delta^{13}C_{OC}$

was very similar to $\delta^{13}C_{EC}$ (Table S3), and showed strong correlations ($r^2$=0.90), indicating that OC originates from a similar source mixture as EC. There are no depleted $\delta^{13}C_{OC}$ values in summer and autumn as expected due to the secondary OC formation. It is partially due to the high temperature: (i) high temperature favors equilibrium shifts to the gas phase, and the formed SOA less efficiently partitions to the particle phase; (ii) aging processes also increase which causes enriched $\delta^{13}C_{OC}$ in the particle phase. This is further discussed in Sect. 4.4.

**3.4 $\delta^{13}$C/F$^{14}$C-based statistical source apportionment of EC**

Figure 3 shows $^{14}$C-based $f_{fossil}$(EC) against $\delta^{13}C_{EC}$ together with the isotopic signature of their source endmembers. The source endmembers for $\delta^{13}$C are less well constrained than for $^{14}$C. For example, $\delta^{13}$C values for liquid fossil fuel combustion overlaps with $\delta^{13}$C values for both coal and C3 plant combustion. In contrast to $\delta^{13}$C, $f_{bb}$ and $f_{fossil}$ are clearly different and the uncertainties in the endmembers are related to the combined uncertainties of $^{14}$C measurements and the

factor used to eliminate the bomb test effect. All data points fall reasonably well within the "source triangle" of C3 plant, liquid fossil fuel (e.g., traffic) and coal combustion, except that $\delta^{13}C_{EC}$ in winter are on the higher (i.e., enriched) end of coal combustion, indicating possible influence of C4 plants combustion as discussed above in Sect. 3.3.



### 3.4.1 Selection of δ¹³C endmembers for C4 plants in the study area

To incorporate possible contribution from C4 plants into the source apportionment, we need to estimate the $\delta^{13}C$ signature of aerosols emitted by C4 biomass burning. Corn stalk is the dominant C4 plant in Xi'an and its surrounding areas (Guanzhong Plain), with little sugarcane and other C4 plants (Sun et al., 2017; Zhu et al., 2017). Estimates of $\delta^{13}C$ of corn stalk burning

emissions range from -19.3 ‰ to -13.6 ‰ (Chen et al., 2012; Kawashima and Haneishi, 2012; Liu et al., 2014a; Guo et al., 2016). $\delta^{13}C$ values of aerosols from corn stalk burning were compiled from literature (Fig. S4). The mean was computed as the average of the different data sets, and standard deviation analogously calculated. $\delta^{13}C$ source signatures for corn stalk burning are -16.45 ± 1.4 ‰ (Fig. S4).

### 3.4.2 Influence of C4 biomass on EC source apportionment

Bayesian Markov chain Monte Carlo techniques (MCMC) were used to account for the variability of the isotope signatures from the different sources (Andersson et al., 2015; Winiger et al., 2015; Fang et al., 2017). Results from a four-source (C3 biomass, C4 biomass, coal and liquid fossil fuel) MCMC4 model and a three-source (C3 biomass, coal and liquid fossil fuel) MCMC3 model were compared to underscore the influence of C4 biomass on source apportionment. The results of the Bayesian calculations are the posterior probability density functions (PDF) for the relative contributions from the sources

(Fig. S5, Fig. S6). For MCMC4, we did a posteriori combination of PDF for C3 biomass and C4 biomass, and named the combined PDF as biomass burning, to better compare results with MCMC3.

To estimate seasonal source contributions to EC, we combined all the data points from each season in the MCMC calculations. Yearly source apportionment was conducted by combining all the data points, to improve the precision of the estimated source contributions. The median was used to represent the best estimate of the contribution of any particular

source to EC. Uncertainties of this best estimate are expressed as inter-quartile range and 95 % range of corresponding PDF. For both MCMC4 and MCMC3, the MCMC-derived fraction of biomass burning EC ($f_{bb}$) is similar to that obtained from radiocarbon data as it is well-constrained by $F^{14}C$ (Table 2, Table S5, Table S6, Fig. S7). Compared to MCMC4, MCMC3 overestimated the contributions from coal combustion, and underestimated the contributions from liquid fossil fuel combustion (Fig. 4). In MCMC3, the $\delta^{13}C$ signature for biomass burning ($\delta^{13}C_{bb}$) is taken from C3 plants only (-26.7 ± 1.8

‰), and is therefore more depleted compared the $\delta^{13}C_{bb}$ of combined C3 (-26.7 ± 1.8 ‰) and C4 (-16.45 ± 1.4 ‰) signatures in MCMC4. With the same $f_{bb}$ in both MCMC3 and MCMC4, MCMC3 calculations apportion a bigger fraction of EC to $\delta^{13}C$-enriched coal combustion in order to explain the enriched winter $\delta^{13}C_{EC}$. As a result, MCMC3-derived contributions of liquid fossil fuel combustion to EC was only 14 % in winter, 5 times less than in summer. This implies the absolute EC concentrations from liquid fossil fuel combustion were much smaller in winter than in summer, considering that the total EC

concentrations in winter were only 1.5 times higher than that in summer. This is inconsistent with our expectation that absolute EC concentrations from liquid fossil fuel combustion should be roughly constant all over the year, or even higher in winter due to unfavourable meteorological conditions. If we do not include C4 biomass in calculation, coal combustion





contributions will be overestimated, and combustion of liquid fossil fuel be underestimated, especially in winter when $\delta^{13}C_{EC}$ are most enriched combined with highest contribution from biomass burning.

MCMC4 calculations reveal that on a yearly average the highest contribution to EC is from liquid fossil sources (median, 72 %; interquartile range, (65−77 %); Table 2), followed by biomass burning (17 %, 16−18 %), and coal combustion (11 %, 6–18 %). However, source patterns changed substantially between different seasons. Coal combustion was the dominant contributor to EC concentrations in winter, with a median of 45 % (29–58 %). Contrary to winter, EC in other seasons was mainly derived from liquid fossil usage, accounting for 67 % (56–74 %), 71 % (63–77 %) and 77 % (71–82 %) of EC in autumn, spring, and summer, respectively. The larger contribution from coal combustion in winter was associated with the extensive coal use for residential heating and cooking in Xi'an, in addition to contributions from coal-fired industries and power plants. This is in line with the findings from $\delta^{13}C$ results. We consider that EC from coal-fired industries and power plants are much lower than that from residential coal combustion, because they have high combustion efficiency and widely-used dust removal facilities. For example, a previous study reported that EC emission factors (emitted EC amount per kg fuel) from residential coal combustion are up to 3 orders of magnitudes higher than those from industries and power plants (Zhang et al., 2008). However, relative contributions from fossil combustion ($f_{coal}+f_{liq.fossil}$) were on average lower in winter than in other seasons (warm period), implying that contributions from biomass burning were also important for the EC increment in winter. By subtracting mean $EC_{bb}$ and $EC_{fossil}$ in the warm period from those in winter, the excess $EC_{bb}$ and $EC_{fossil}$ was 1.2 μg m$^{-3}$ and 0.8 μg m$^{-3}$, respectively. Biomass burning contributed on average 60 % of EC increment in winter.

### 3.5 Estimating mass concentrations and sources of primary OC

Comparing concentrations and sources of primary OC to total OC can give insights into the importance of secondary formation and other chemical processes, such as photochemical loss mechanisms. Based on the EC concentrations from biomass, coal, and liquid fossil fuel combustion derived from MCMC4 model, the total primary OC mass concentrations due to these three major combustion sources can be estimated ($OC_{pri,e}$; OC primary, estimated). The respective EC concentrations apportioned to each source are multiplied by the characteristic primary OC/EC ratios for each source (Eq. (10)). The non-fossil fraction (i.e., biomass burning) in $OC_{pri,e}$ ($f_{bb}(OC_{pri,e})$) is approximated by Eq. (11):

$$OC_{pri,e} = POC_{bb,e} + POC_{coal,e} + POC_{liq.fossil,e} = \left(r_{bb} \times f_{bb} + r_{coal} \times f_{coal} + r_{liq.fossil} \times f_{liq.fossil}\right) \times EC, \qquad (10)$$

$$f_{bb}\left(OC_{pri,e}\right) = \frac{POC_{bb,e}}{OC_{pri,e}} = \frac{r_{bb} \times f_{bb}}{r_{bb} \times f_{bb} + r_{coal} \times f_{coal} + r_{liq.fossil} \times f_{liq.fossil}}, \qquad (11)$$

where $POC_{bb,e}$, $POC_{coal,e}$, and $POC_{liq.fossil,e}$ are estimated primary OC mass concentrations from biomass burning, coal combustion and liquid fossil fuel combustion, respectively. $r_{bb}$, $r_{coal}$, and $r_{liq.fossil}$ are OC/EC ratios for primary emissions from biomass burning, coal combustion, and liquid fossil fuel combustion, respectively. The selection of $r_{bb}$ (5 ± 2), $r_{coal}$ (2.38 ± 0.44), and $r_{liq.fossil}$ (0.85 ± 0.16) is done by literature searches and described in Supplemental S3. $f_{bb}$, $f_{coal}$, and $f_{liq.fossil}$ are the




relative contribution to EC from combustion of biomass, coal, and liquid fossil fuel derived from MCMC4 model. EC denotes EC mass concentrations ($\mu g\ m^{-3}$).

A Monte Carlo simulation with 10,000 individual calculations of $OC_{pri,e}$ and $f_{bb}(OC_{pri,e})$ was conducted to propagate uncertainties. For each individual calculation input, EC concentrations are randomly chosen from a normal distribution symmetric around the measured values with uncertainties as standard deviation; the random values for $r_{bb}$, $r_{coal}$ and $r_{liq.fossil}$ are taken from a triangular distribution, which has its maximum at the central value and 0 at the upper and lower limits. For $f_{bb}$, $f_{coal}$ and $f_{liq.fossil}$, the PDF derived from MCMC4 model was used (Fig. S9). Then 10,000 different estimations of $OC_{pri,e}$ and $f_{bb}(OC_{pri,e})$ were calculated. The derived median represents the best estimate, and interquartile ranges (25th-75th percentile) were calculated to represent the combined uncertainties.

The observed OC concentrations and non-fossil fractions $f_{nf}(OC)$ as well as estimated $OC_{pri,e}$, $f_{bb}(OC_{pri,e})$ are shown in Fig. 5. $OC_{pri,e}$ tracks the observed concentrations and seasonality of OC very well, with correlation of $r^2=0.71$ ($p<0.05$). $OC_{pri,e}$ are only substantially lower than OC, when observed OC concentrations > 25 $\mu g\ m^{-3}$ (Fig. 5(a)). Observed OC mass concentrations that exceed $OC_{pri,e}$ can be explained by contribution from secondary OC from coal combustion ($SOC_{coal}$), and liquid fossil fuel usage ($SOC_{liq.fossil}$) and by other non-fossil OC ($OC_{o,nf}$) that includes secondary OC from biomass burning and biogenic sources ($SOC_{nf}$; SOC non-fossil), and primary OC from vegetative detritus, bioaerosols, resuspended soil organic matter, or cooking. So:

$$\text{Observed OC conentrations} - OC_{pri,e} = OC_{o,nf} + SOC_{coal} + SOC_{liq.fossil}, \tag{12}$$

In most cases, the contributions to $PM_{2.5}$ from vegetative detritus, bioaerosols and soil dust in the air are likely small, because their sizes are usually much larger than 2.5 $\mu m$. For example, Guo et al. (2012) estimated that vegetative detritus only accounts for ~1% of OC in $PM_{2.5}$ in Beijing, China, using chemical mass balance (CMB) modeling and tracer-yield method. Thus, this fraction of OC can be ignored (i.e., $OC_{o,nf} \approx SOC_{nf}$). A previous [14]C study in Xi'an during severe winter pollution days in 2013 also reveals that increased total carbon (TC = OC + EC) was mainly driven by enhanced SOC from fossil and non-fossil sources (Zhang et al., 2015a), that is $SOC_{coal}$, $SOC_{liq.fossil}$, and $SOC_{nf}$, all of which are not modelled in $OC_{pri,e}$.

$OC_{pri,e}$ was higher than the total observed OC in summer 2008, which may indicate an overestimate of primary OC/EC ratios, or loss of OC due to photochemical processing. Xi'an is one of the four "stove cities" in China. In summer, daily average temperature was 25–31°C, and occasionally exceeded 38°C. At these temperatures, semi-volatile OC from emission sources becomes volatilized more quickly owning to higher temperatures, leading to lower primary OC/EC ratios than other seasons. These low OC/EC ratios in summer are commonly observed in urban China (e.g. median, 2.7; interquartile range, (1.9−4) from an overview of $PM_{2.5}$ composition in China by Tao et al. (2017)). This evaporation can be compounded by loss through photochemical reactions that lead to fragmentation of organic compounds.



On the other hand, the estimated $f_{bb}(OC_{pri,e})$ are consistently lower than observed $^{14}$C-based $f_{nf}(OC)$, and weak correlations were observed ($r^2$=0.31). Differences between the non-fossil carbon fraction in primary aerosol ($f_{bb}(OC_{pri,e})$) and in the total organic aerosol $f_{nf}(OC)$ can in principle be expected due to secondary organic aerosol formation. A higher fraction of non-fossil carbon in total OC than in estimated primary OC implies that non-fossil sources contribute more strongly to SOC

formation than fossil sources. Some previous observations support this hypothesis. Zhang et al. (2015a) also reported that the relative contribution of $OC_{o,nf}$ is ~2 times higher than that of $SOC_{coal}$ and $SOC_{liq.fossil}$ in January 2013 at the same sampling site. In winter, $OC_{o,nf}$ is likely dominated by SOC from biomass-burning emissions, while contributions from biogenic SOC is small. In spring and summer, additional contributions from biogenic SOC can further elevate $f_{nf}(OC)$ compared to $f_{bb}(OC_{pri,e})$.

However, considering both $f_{nf}(OC)$ and OC concentrations, this simple model of total OC as the sum of primary and secondary OC leads to an apparent contradiction for spring and summer observations. $OC_{pri,e}$ already equals to or exceeds the total measured OC concentrations, whereas additional SOC is necessary to explain the observed higher $f_{nf}(OC)$. Spring and summer temperatures in Xi'an are generally high, which favours active photochemistry. The resulting loss of OC due to photochemistry probably also needs to be considered to explain the observations.

## 15   4 Discussion

### 4.1 Aerosol in Xi'an compared to other Chinese cities

There are few annual $^{14}$C measurements in China (Table 1). The annual average $f_{fossil}(EC)$ in Xi'an is 83 %. This falls in the range of annual $f_{fossil}(EC)$ measured in China, depending on the location. Comparable annual $f_{fossil}(EC)$ was reported at an urban site of Beijing (79 ± 6% (Zhang et al., 2015b); 82% ± 7% (Zhang et al., 2017)) and a background receptor site of

Ningbo (77 ± 15% (Liu et al., 2013)). Much lower $f_{fossil}(EC)$ was found at a regional background site in Hainan (38 ± 11 % (Zhang et al., 2014b)). The big differences between the two background sites are due to different air-mass transport to the receptor site. The background site in Ningbo was more often influenced by air-masses transported from highly urbanized regions of East China associated with lots of fossil-fuel combustion, whereas the decreased fossil contribution observed in Haian could be attributed to enhanced open burning of biomass in Southeast Asia or Southeast China.

In this study, $f_{fossil}(EC)$ was lowest in winter (77 %). This is comparable with previously reported $f_{fossil}(EC)$ at the same sampling site during winter 2013 (78 ± 3 % (Zhang et al., 2015a)), Shanghai (79 ± 4 %, Zhang et al., 2015a), Wuhan (74 ± 8 % (Liu et al., 2016b)), North China Plain (73–75 %, Andersson et al., 2015) and Guangzhou (71 ± 10 %, Liu et al., 2014b). Higher $f_{fossil}(EC)$ in winter are reported in Beijing (80–87%, Sun et al., 2012; 83 ± 4 %, Chen et al., 2013), Xiamen (87 ± 3 %, Chen et al., 2013). Lower winter $f_{fossil}(EC)$ was observed in Guangzhou (69 %, Zhang et al., 2015a), Yangtze River Delta

(66–69 %, Andersson et al., 2015), and Pearl River Delta (67–70 %, Andersson et al., 2015), indicating different influence of



biomass burning emissions over China during winter. $^{14}$C measurements in other seasons are still very scarce in China.

The annual average $f_{fossil}$(OC) in Xi'an is 46%, with the lowest values in winter (38 %) and the highest in summer (52 %). The annual average $f_{fossil}$(OC) in this study is comparable to the results found in an urban site of Beijing (48 ± 12 %) (Zhang et al., 2017), but higher than 19 ± 10 % at a background site of Hainan (Zhang et al., 2014b). Similar contributions from

fossil sources to OC were reported for the same sampling site at Xi'an in winter 2013 (38 ± 3%, Zhang et al., 2015a), Wuhan in January 2013 (38 ± 5 %, Liu et al., 2016b), and Guangzhou in winter 2012/2013 (37 ± 4 %, Liu et al., 2014b). A higher fossil contribution to OC was found in Beijing with $f_{fossil}$(OC) of 58 ± 5 % in winter 2013 and 59 ± 6 % in winter 2013/2014 (Zhang et al., 2015a, 2017), and in Shanghai with $f_{fossil}$(OC) of 49 ± 2 % in winter 2013 (Zhang et al., 2015a)). Previous studies in Beijing observed different seasonal trends, with higher contribution by fossil sources in winter (higher $f_{fossil}$(OC))

than in other seasons (Yan et al., 2017; Zhang et al., 2017). This is consistent with findings by online aerosol mass spectrometer analysis in winter 2013/2014 (Elser et al., 2016), where organic matter in Xi'an was found to be dominated by biomass burning, in contrast to Beijing where it is dominated by coal burning. This implies different pollution patterns over Chinese cities.

The $\delta^{13}C_{EC}$ is most enriched in winter (-23.20 ± 0.35 ‰), and most depleted in summer (-25.94 ± 0.46 ‰). This is consistent

with previous studies in northern China, with the winter-to-summer difference ranging from 0.76 to 2.79 ‰ for all the 7 northern Chinese cities (e.g., Cao et al., 2011; Table S8), supporting the important influence on EC from coal combustion in winter. By contrast, no notable difference between winter and summer $\delta^{13}C_{EC}$ is reported in southern China, where there is no official heating season. (e.g., Ho et al., 2006; Cao et al., 2011; Table S8). $\delta^{13}C_{OC}$ showed a seasonal variation pattern similar to $\delta^{13}C_{EC}$. $\delta^{13}C_{OC}$ is most enriched in winter (-24.13 ± 0.83 ‰), comparable with previously reported winter data in

North China, for example Beijing (-24.26 ± 0.29 ‰) by Yan et al. (2017), and seven northern cities in China (-25.54 ‰ to -23.08 ‰) by Cao et al. (2011), but our winter $\delta^{13}C_{OC}$ is more enriched than those found in South China, for example Hong Kong (-26.9 ± 0.6 ‰) by Ho et al. (2006), and seven southern cities in China (-26.62 ‰ to -25.79 ‰) by Cao et al. (2011) (Table S8). The differences in North and South China reveal the influence from coal burning to OC.

### 4.2 Correlations between F$^{14}$C$_{(EC)}$ and biomass burning markers

In $^{14}$C-based source apportionment, biomass burning is considered the only source of non-fossil EC. Here we evaluate F$^{14}$C$_{(EC)}$ with other biomass burning markers, including levoglocosan and water-soluble potassium (K$^+$). In summer, a very strong positive correlation ($r^2 = 0.98$) was found between F$^{14}$C$_{(EC)}$ and K$^+$/EC ratios, in contrast to the significant negative correlation ($r^2 = 0.96$) between F$^{14}$C$_{(EC)}$ and levoglucosan/EC ratios (Fig. 6). Previous studies have found that burning of crop residues emitted more K$^+$ than levoglucosan, with significantly lower levoglucosan/K$^+$ ratios than burning of wood

(Cheng et al., 2013; Zhu et al., 2017). The levoglucosan/K$^+$ ratio for wood is 23.96 ± 1.82, much higher than those for crop residues (0.10 ± 0.00 for wheat straw, 0.21 ± 0.08 for corn straw, 0.62 ± 0.32 for rice straw) (Cheng et al., 2013). Emissions



from crop residue burning therefore increase both the fraction of EC from non-fossil sources and K$^+$. This results in a positive correlation between K$^+$/EC ratios and F$^{14}$C$_{(EC)}$. At the same time emissions from crop residue burning contain relatively little levoglucosan and atmospheric levoglucosan concentrations are expected to be dominated by wood burning emissions. If wood burning emissions stay relatively constant, an increase in crop burning emissions will increase EC

concentrations, but have little effect on levoglucosan concentrations, leading to lower levoglucosan/EC ratios. The significant positive correlation of F$^{14}$C$_{(EC)}$ with K$^+$ /EC ratios coinciding with a negative correlation of F$^{14}$C$_{(EC)}$ with levoglucosan/EC ratios in summer therefore suggests strong impacts from crop residues burning and little influence from wood burning on the variability of EC. Variable crop burning activities superimposed on a relatively constant background contribution from wood burning can explain the observed correlations. In summer, extensive open burning in croplands is

also detected in the MODIS fire counts map (NASA, 2017) (Fig. S10), when farmers in the surrounding area of Xi'an (i.e., Guanzhong Plain) burned crop residues in fields. No significant correlations of F$^{14}$C$_{(EC)}$ with K$^+$/EC or levoglucosan/EC were found in other seasons (Fig. S11), suggesting a changing mixture of both biomass subtypes.

**4.3 Changes in emission sources in Xi'an, China (2008/2009 vs. 2012/2013)**

EC is a primary emission product, and thus changes in EC sources can reflect the changes in emission sources. The

contributions from biomass burning to EC was 24 % (median; interquartile range 22–26 %) in winter 2008/2009 (Fig. 7, Table 2). Comparable contributions were also reported at the same sampling site for winter 2012/2013 based on $^{14}$C measurements (22 ± 3 %, Zhang et al., 2015a), and positive matrix factorization (PMF) receptor model simulation (20.1 ± 7.9 %, Wang et al., 2016) (Fig. 7). This suggests that from 2008 to 2013, biomass burning contributions to EC remained rather stable, although with a slight decrease from 24 % (22–26 %)  to 20 % (SD=7.9 %). Biomass burning in Xi'an mainly

includes open burning of crop residues, and household usage of crop residues and wood. The slight decrease can be explained by more strict rules to minimize crop open burning, but implementation of regulations was still weak and slow. Moreover, there are no regulations yet that target household biomass usage (Zhang and Cao, 2015b).

The contributions of coal combustion to EC decreased from 45 % (29–58 %) in winter 2008/2009 to 33.9 % (SD=23.8 %) in winter 2012/2013, with increased contributions from vehicle emission from 31 % (18–46 %) to 46 % (SD=25.1 %) (Fig. 7).

Vehicle emissions become increasingly important and coal combustion less from 2008 to 2013. This change could not be detected from $^{14}$C measurements alone, since the total fossil contribution to EC stayed relatively constant. Further apportionment of fossil sources into coal combustion and vehicle emissions could be achieved by combining $^{14}$C measurements with $\delta^{13}$C (Andersson et al., 2015; Winiger et al., 2016) or organic source markers (Zhang et al., 2015b).

The decreased contribution from coal combustion to EC from 2008 to 2013 resulted from stepwise replacement of coal

combustion by natural gas for residential heating and cooking since the second half of the 2000s. Natural gas usage in Xi'an increased by 94 % from 2009 to 2013 (Xi'an Statistical Yearbook, 2010, 2014). Although coal combustion in Xi'an had been



increasing from 6.6 million ton 2008 to 10.3 million ton in 2013, the proportion of coal used as energy reduced from 71 % to 66 % (Xi'an Statistical Yearbook, 2009, 2014). The reinforcement of environmental laws and regulations, encouragement of using high-efficiency improved coal burners and high-quality coals are important factors as well. The decreased coal combustion emissions are also evidenced from the declined enrichment factor (EF) of As and Pb. As and Pb can indicate

coal combustion, as Pb-containing gasoline has been forbidden since 2000 in Xi'an (Xu et al., 2012). Annual EFs of As and Pb dropped from 802 and 804 in 2008 to 465 and 490 in 2010, respectively (Xu et al., 2016).

Vehicular emissions to EC increased from 31 % to 46 % (an absolute relative increase by roughly 50 %) from 2008 to 2013 (Fig. 7). This is supported by increasing levels of $NO_2$ in urban Xi'an, which is another indicator for the contribution of vehicular emissions to air pollution. The $NO_2$ concentrations in Xi'an increased by 15.5 % from 2006 to 2010 (Xu et al.,

2016). The increased vehicular contribution likely resulted from a strong increase in civil vehicles. The processing (registration) of civil vehicles increased > twofold from 0.9 million unit in 2008 to 1.9 million unit in 2013 (Xi'an Statistical Yearbook, 2009, 2014). However, vehicular contributions to EC and $NO_2$ concentrations have not increased to the same extent as the increase in vehicle numbers. This can be attributed to the upgrade of vehicle emission standard from National II to National III for light-duty gasoline and heavy-duty diesel vehicles in 2007 and for heady-duty gasoline vehicles in 2010 in

Xi'an (GB18352.3-2005, 2005; GB17691-2005, 2005), which somewhat offset the increase of vehicle numbers.

**4.4 Differences between observed and estimated primary OC concentrations and sources**

The estimated $OC_{pri,e}$ concentrations are comparable to the observed OC concentrations except for samples with observed OC concentrations >25 μg m$^{-3}$. However, $f_{bb}(OC_{pri,e})$ is considerably lower than the observed $f_{nf}(OC)$. It is worth investigating, whether this might be due to the model assumptions, for example the OC/EC emission ratios used for the

primary sources. OC/EC ratios are known to be dependent on the measurement protocol applied to the samples (Chow et al., 2001, 2004). For examples, Han et al. (2016) found that for fresh biomass burning emissions, OC/EC ratios by EUSAAR_2 (Cavalli et al., 2010) is 2 times higher than those by IMPROVE_A (Chow et al., 2007). According to Eq. (11), underestimated $r_{bb}$ or overestimated $r_{coal}$ and $r_{liq.fossil}$ would result in a $f_{bb}(OC_{pri,e})$ that is biased towards low values. Impacts of $r_{bb}$ on $f_{bb}(OC_{pri,e})$ are presented in Fig. 5(b). With higher $r_{bb}$= 5 (3–7, minimum–maximum; our best estimate from the

literature review presented in Supplemental S3) compared to $r_{bb}$= 4 (3–5), $f_{bb}(OC_{pri,e})$ increases only by 4 % to 7 %. Any further increase of $r_{bb}$ would result in a modelled $OC_{pri,e}$ that is substantially higher than that total measured OC.

On the other hand, $r_{liq.fossil}$ of 0.85 ± 0.16 was applied without considering its seasonal variations. However, it is found that $r_{liq.fossil}$ is lower in summer compared with other seasons, which is related to increased volatilization of semi-VOCs and faster catalyst and engine warm-up times in summer (Xie et al., 2017). Huang et al. (2014b) found OC/EC ratios from fresh

vehicular emissions in summer to be ~80 % of the yearly average, based on the lowest 5 % OC/EC ratios measured in a roadside environment in Hongkong, China. The $f_{bb}(OC_{pri,e})$ would increase 3 % to 5 % in summer, if we apply 80 % of the



yearly average $r_{\text{liq.fossil}}$ for the summer (Fig. 5(b)), which is also not a substantial increase. In summary, it is not feasible to model the observed $f_{\text{nf}}(OC)$ by primary emissions, even though the total OC concentrations are in the range of modelled primary OC for spring and summer. Moreover, in spring $\delta^{13}C_{OC}$ is lower than $\delta^{13}C_{EC}$ (Fig. 2). This points to a depleted OC source, which can be an indication of secondary formation of OC. In summary, the isotopic composition of OC makes a predominantly primary origin very unlikely.

A more realistic model for OC concentrations and $f_{\text{nf}}(OC)$ needs to account for $OC_{\text{o,nf}}$, $SOC_{\text{coal}}$ and $SOC_{\text{liq.fossil}}$:

$$f_{\text{nf}}(OC) = \frac{POC_{\text{bb,e}}+OC_{\text{o,nf}}}{OC_{\text{pri,e}}+O_{\text{o,nf}}+SOC_{\text{coal}}+SOC_{\text{liq.fossil}}} \tag{13}$$

Then the estimated total OC ($OC_e$) will be:

$$OC_e = OC_{\text{pri,e}} + OC_{\text{o,nf}} + SOC_{\text{coal}} + SOC_{\text{liq.fossil}}, \tag{14}$$

As a sensitivity study with minimum addition to $OC_{\text{pri,e}}$ (thus minimum $OC_e$, $OC_{\text{e,min}}$), we make the unrealistic assumption that there is no SOC from coal and liquid fossil fuel combustion ($SOC_{\text{coal}}$=0, $SOC_{\text{liq.fossil}}$=0). Only the required $OC_{\text{o,nf}}$ is added until the modelled $f_{\text{nf}}(OC)$ is equal to the measured one. Figure 8 presents the modelled $OC_{\text{e,min}}$ and observed OC concentrations. Nearly half of $OC_{\text{e,min}}$ are higher than observed OC and especially in summer, the OC concentrations are consistently overestimated. For many of the data points in fall and spring there is a reasonable agreement between model and measurements. There are only a few haze episodes in winter, where additional SOC formation would be required to explain observed OC concentrations. However, a previous study in winter 2013 at the same sampling site found the secondary fossil OC was 0.75–1.6 times that of primary fossil OC (Zhang et al., 2015a), which indicates that that fossil SOC is likely also of importance. If we also include $SOC_{\text{coal}}$ and $SOC_{\text{liq.fossil}}$, this leads to a further overestimate of absolute OC concentrations, if we simply estimate total OC as the sum of primary and secondary OC. Therefore, the more reasonable explanation is OC loss. The primary OC/EC ratios do not preserve the characteristics of sources any more in a warm period due to active photochemistry under high temperature and humidity. The conclusion will not change if we apply EC apportion results from MCMC3 (Fig. S12, Fig. S13).

## 5 Conclusions

Sources of OC and EC in Xi'an, China are constrained based on a full year of radiocarbon and stable carbon isotopic measurements for the year 2008–2009. Radiocarbon measurement reveals that EC is dominated by fossil sources with contributions ranging from 71 % to 89 %, with an average of 83 ± 5 %. Compared with EC, OC has much higher contribution from non-fossil sources (54 ± 8 %), with higher contribution in winter (62 ± 5 %). Fossil contributions to OC and EC in this study fall within the range of published values from other $^{14}$C-based source apportionments in Chinses cities. The annual fossil contribution to OC and EC in Xi'an was comparable to Beijing (Zhang et al., 2015b, 2017), but higher than





that from a regional background site in Hainan (Zhang et al., 2014b). In this study, the non-fossil contribution to OC in winter ($f_{nf}$(OC) = 62 ± 5 %) was observed to be higher than in summer (48 ± 3 %) in Xi'an. A different seasonal variation pattern for $f_{nf}$(OC) was reported in Beijing, where the fossil contribution to OC was higher in winter than in summer (Yan et al., 2017; Zhang et al., 2017). This implies that different pollution patterns exist in individual Chinese cities.

In summer, a strong positive correlation was found between $F^{14}C_{(EC)}$ and $K^+$/EC ratios, and a significant negative correlation between $F^{14}C_{(EC)}$ and levoglucosan/EC ratios. This suggests that burning of crop residues, with significant lower levoglucosan/$K^+$ ratios than wood, accounted for most of the variability in non-fossil EC in the summer. No significant correlations of $F^{14}C_{(EC)}$ with $K^+$/EC or levoglucosan/EC were found in other seasons (Fig. S11), suggesting a variable mixture of biomass subtypes.

The annual averaged $\delta^{13}C_{EC}$ is -24.92 ± 1.14 ‰, varying between −26.50 ‰ and -22.81 ‰. The $\delta^{13}C_{EC}$ is most enriched in winter, ranging from -23.72 ‰ to -22.81 ‰. Winter $\delta^{13}C_{EC}$ values are at the higher (i.e., less negative) end of pure coal combustion emissions even though considerable contributions from more depleted liquid fuel combustion and wood burning to EC are expected. This indicates some contribution from C4 plants, such as corn stalk burning, in addition to coal combustion. To further refine EC sources, radiocarbon and stable carbon signatures are combined and used in a Bayesian

Markov Chain Monte Carlo (MCMC) approach, in which burning of C4 plants is included as a subtype of biomass burning. The MCMC results indicate that coal combustion dominated EC in winter, and liquid fossil fuel combustion dominated EC in other seasons. However, increased contributions from biomass burning were important for the EC increment in winter as well. Comparisons with the results of other studies at the same sampling site suggest that the source contributions of fossil primary carbonaceous aerosol have changed from 2008/2009 to 2013/2014, with decreasing contributions from coal burning

and increasing contributions from motor vehicles. The changes in source contributions to EC in Xi'an is consistent with recent changes in the region: changes in energy consumption, and the expansion of the civil vehicular fleet resulting from urbanization and economic improvement.

    $\delta^{13}C_{OC}$ exhibited similar values to $\delta^{13}C_{EC}$, and showed strong correlations ($r^2$ = 0.90) in summer and autumn, indicating similar source mixtures as EC and influence of high temperature on atmospheric processing of OC. In spring, $\delta^{13}C_{OC}$ is more

depleted than $\delta^{13}C_{EC}$, indicating the possible importance of secondary formation of OC (e.g., from volatile organic compound precursors) in addition to primary sources. Comparing the observations (OC mass, $^{14}C$-based $f_{nf}$(OC)) with estimated total primary OC concentrations related to combustion sources (i.e., estimated by apportioned EC and corresponding OC to EC ratios) and the non-fossil fraction in the estimated primary OC makes it possible to provide some insights into the importance of secondary formation and other chemical processes, such as photochemical loss mechanisms.

It is found that estimated primary OC mass follows the observed total OC concentrations and seasonality ($r^2$ = 0.71), but source contributions to total OC differ from the estimated source contributions to primary OC ($r^2$ = 0.31). The estimated primary OC is similar to the observed OC concentrations except for samples with observed OC concentrations >25 μg m⁻³.



However, the non-fossil fraction in estimated primary OC is significantly lower than the observed $f_{nf}$(OC). Those differences can be explained by the contribution of other non-fossil primary OC (excluding biomass burning), or secondary non-fossil OC, which are not included in the estimation. But we cannot reconcile the differences between observed and estimated non-fossil OC fraction without overestimating the absolute OC concentrations, especially in summer. Therefore, we hypothesize that OC loss due to active photochemistry cannot be neglected, especially not in summer.

**Acknowledgments**

This work was supported by the KNAW project (Nr. 530-5CDP30). The authors acknowledge the financial support from the Gratama foundation. Special thanks to Henk Been and Marc Bleeker for their help with the AMS measurements at CIO, and to Anita Aerts-Bijma and Dicky van Zonneveld for their help with [14]C data correction at CIO.

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



**Table 1.** Relative fossil source contribution to OC and EC ($f_{fossil}$(OC), $f_{fossil}$(EC) in percentage) in China

| Location | Site type | PM fraction | Season | Year | $f_{fossil}$(OC) | $f_{fossil}$(EC) | Reference |
|---|---|---|---|---|---|---|---|
| Beijing | urban | $PM_{2.5}$ | winter | 2009/2010 | | $83 \pm 4$ | (Chen et al., 2013) |
| Beijing | urban | $PM_{2.5}$ | spring | 2013 | $41 \pm 4$ | $67 \pm 7$ | (Liu et al., 2016a) |
| Beijing | rural | $PM_{2.5}$ | winter | 2007 | | 80–87 | (Sun et al., 2012) |
| Beijing | rural | $PM_{2.5}$ | summer | 2007 | | 80–87 | (Sun et al., 2012) |
| Beijing | urban | $PM_{2.5}$ | winter | 2013 | $67 \pm 3$ | | (Yan et al., 2017) |
| Beijing | urban | $PM_{2.5}$ | summer | 2013 | $36 \pm 13$ | | (Yan et al., 2017) |
| Beijing | urban | $PM_4$ | annual | 2010/2011 | | $79 \pm 6$ | (Zhang et al., 2015b) |
| Beijing | urban | $PM_{2.5}$ | winter | 2013 | $58 \pm 5$ | $76 \pm 4$ | (Zhang et al., 2015a) |
| Beijing | urban | $PM_1$ | annual | 2013/2014 | $48 \pm 12$ | $82 \pm 7$ | (Zhang et al., 2017) |
| Guangzhou | urban | $PM_{2.5}$ | winter | 2012/2013 | $37 \pm 4$ | $71 \pm 10$ | (Liu et al., 2014b) |
| Guangzhou | urban | $PM_{2.5}$ | spring | 2013 | $46 \pm 6$ | $80 \pm 5$ | (Liu et al., 2016a) |
| Guangzhou | urban | $PM_{10}$ | winter | 2011 | 42 | | (Zhang et al., 2014c) |
| Guangzhou | urban | $PM_{2.5}$ | winter | 2013 | $35 \pm 7$ | 69 | (Zhang et al., 2015a) |
| Shanghai | urban | $PM_{2.5}$ | winter | 2009/2010 | | $83 \pm 4$ | (Chen et al., 2013) |
| Shanghai | urban | $PM_{2.5}$ | winter | 2013 | $49 \pm 2$ | $79 \pm 4$ | (Zhang et al., 2015a) |
| Xiamen | urban | $PM_{2.5}$ | winter | 2009/2010 | | $87 \pm 3$ | (Chen et al., 2013) |
| Xi'an | urban | $PM_{2.5}$ | winter | 2013 | $38 \pm 3$ | $78 \pm 3$ | (Zhang et al., 2015a) |
| Xi'an | urban | $PM_{2.5}$ | winter | 2008/2009 | $46 \pm 8$ | $83 \pm 5$ | This study |
| Wuhan | urban | $PM_{2.5}$ | winter | 2013 | $38 \pm 5$ | $74 \pm 8$ | (Liu et al., 2016b) |
| North China Plain (NCP) | urban | $PM_{2.5}$ | winter | 2013 | | 73–75 | (Andersson et al., 2015) |
| Yangtze River Delta (YRD) | urban | $PM_{2.5}$ | winter | 2013 | | 66–69 | (Andersson et al., 2015) |
| Pearl River Delta (PRD) | urban | $PM_{2.5}$ | winter | 2013 | | 67–70 | (Andersson et al., 2015) |
| Ningbo | background | $PM_{2.5}$ | annual | 2009/2010 | | $77 \pm 15$ | (Liu et al., 2013) |
| Hainan | background | $PM_{2.5}$ | annual | 2005/2006 | $19 \pm 10$ | $38 \pm 11$ | (Zhang et al., 2014b) |





**Table 2.** MCMC4 results[a] from the $F^{14}C$- and $\delta^{13}C$-based Bayesian Source Apportionment Calculations of EC (Median, interquartile range (25[th]-75[th] percentile), and 95% Credible Intervals).

|  | Seasons | summer | autumn | winter[c] | spring | annual[c] |
|---|---|---|---|---|---|---|
| biomass burning[b] | median | 0.135 | 0.177 | 0.239 | 0.156 | 0.173 |
| (combination of | 25[th]-75[th] percentile | (0.129–0.142) | (0.16–0.197) | (0.22–0.26) | (0.153–0.159) | (0.165–0.18) |
| C3 & C4 plants) | 95% credible intervals | (0.114–0.159) | (0.117–0.249) | (0.172–0.332) | (0.145–0.166) | (0.15–0.195) |
|  |  |  |  |  |  |  |
| coal combustion | median | 0.085 | 0.153 | 0.446 | 0.136 | 0.11 |
|  | 25[th]-75[th] percentile | (0.045–0.15) | (0.083–0.261) | (0.294–0.582) | (0.075–0.219) | (0.063–0.18) |
|  | 95% credible intervals | (0.012–0.412) | (0.02–0.589) | (0.074–0.739) | (0.019–0.492) | (0.016–0.353) |
|  |  |  |  |  |  |  |
| liquid fossil | median | 0.779 | 0.666 | 0.307 | 0.707 | 0.717 |
| fuel combustion | 25[th]-75[th] percentile | (0.713–0.82) | (0.555–0.74) | (0.18–0.457) | (0.627–0.768) | (0.647–0.765) |
|  | 95% credible intervals | (0.452–0.858) | (0.226–0.824) | (0.039–0.684) | (0.357–0.826) | (0.468–0.815) |

[a]Results from the four-sources (C3 biomass, C4 biomass, coal and liquid fossil fuel) MCMC4 model.

[b]Contribution of biomass burning is done by a posteriori combination of PDF for C3 plants and that for C4 plants (Fig. S6). Median and quartile ranges for C3 and C4 plants burning to EC is shown in Table S7.

[c]Sample taken from Chinese New Year eve (25 January 2009) was excluded.




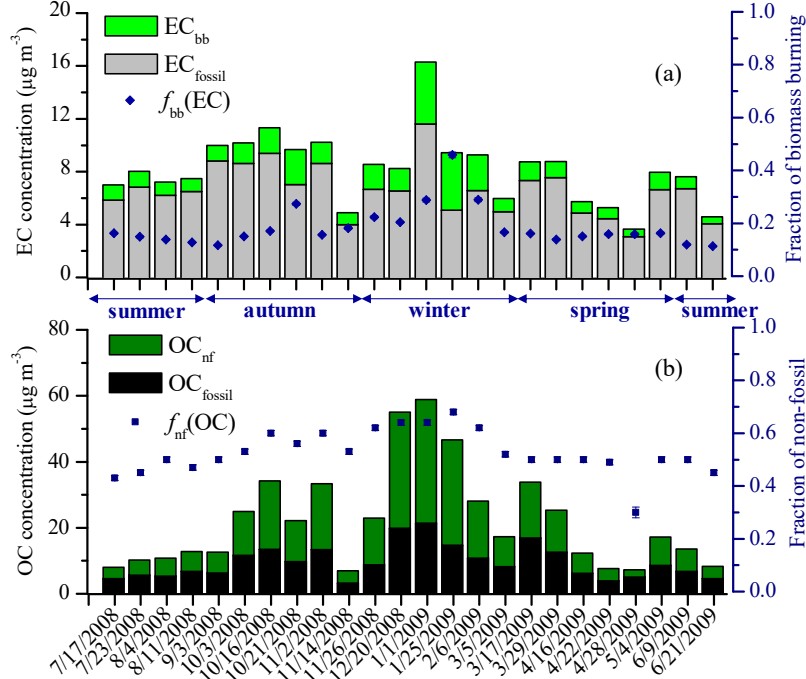

**Figure 1.** (a) Temporal variation of EC mass concentrations from biomass burning (EC_bb) and fossil-fuel sources (EC_fossil), and fraction of biomass burning contribution to EC ($f_{bb}$ (EC)). (b) Temporal variation of OC mass concentrations from non-fossil sources (OC_nf) and fossil-fuel sources (OC_fossil), and fraction of non-fossil OC to total OC ($f_{nf}$ (OC)).





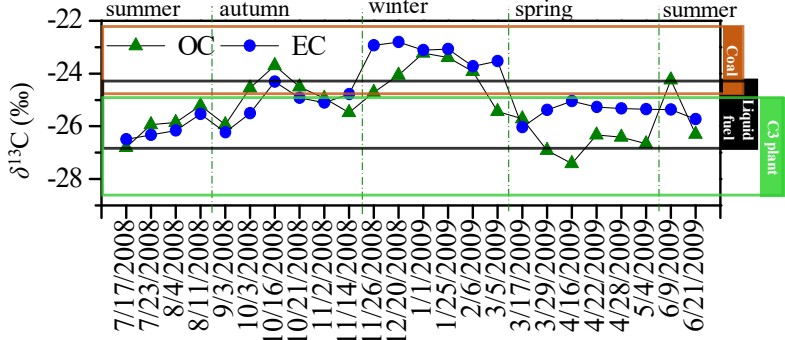

**Figure 2.** Stable carbon signatures ($\delta^{13}$C) in OC and EC for the samples selected for $^{14}$C measurements. The $\delta^{13}$C signatures of C3 plants (green rectangle), liquid fossil (e.g., oil, diesel, and gasoline, black rectangle), and coal (brown rectangle) are indicated as mean ± standard deviation in Table S1. The $\delta^{13}$C endmember ranges for C4 plant burning (-16.45 ± 1.4 ‰, Table S1) are much more enriched than other sources, and are not shown in this figure.





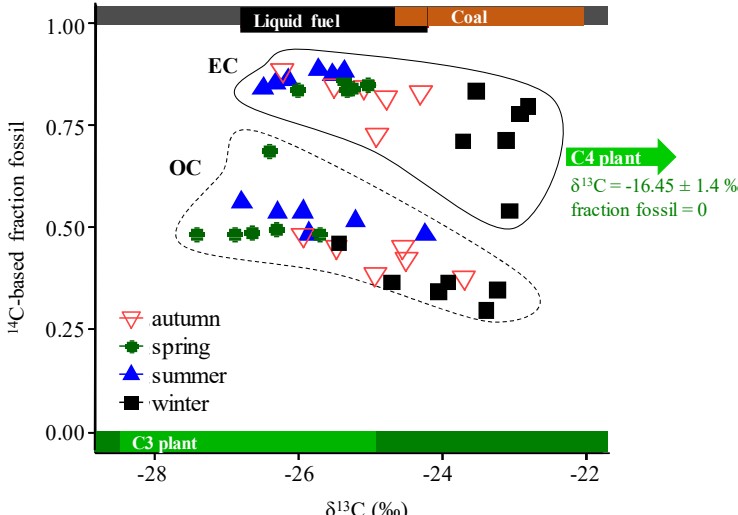

**Figure 3.** Two-dimensional isotope-based source characterization plot of OC and EC in different seasons. The fraction fossil ($f_{fossil}$(EC) and $f_{fossil}$(OC)) was calculated using radiocarbon data. The expected $\delta^{13}$C and $^{14}$C endmember ranges for biomass burning emissions, liquid fossil fuel combustion, and coal combustion are shown as green, black and brown bars, respectively. The $\delta^{13}$C signatures of C3 plants

5 (green rectangle), liquid fossil (e.g., oil, diesel, and gasoline, black rectangle), and coal (brown rectangle) are indicated as mean ± standard deviation in Table S1. The $\delta^{13}$C signatures of C4 plants burning is -16.45 ± 1.4 ‰ is not shown on x-axis.





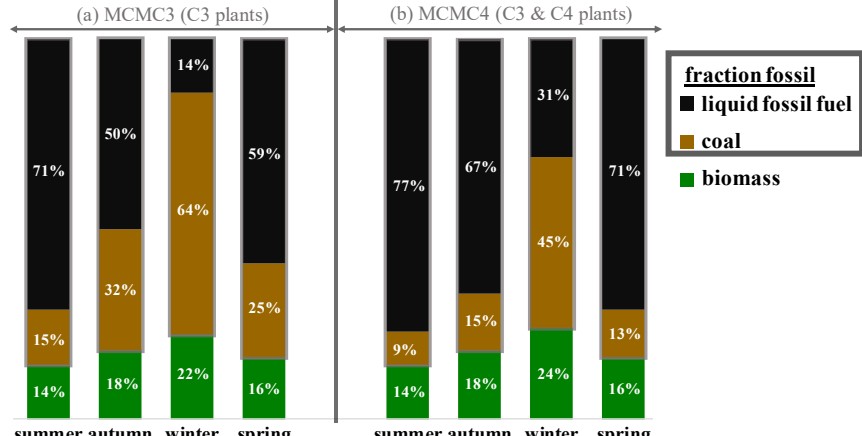

**Figure 4.** Sources of EC in different seasons. Results from the $F^{14}C$ and $\delta^{13}C$ based Bayesian source apportionment calculations of EC. The numbers in the bars represent the median contribution of liquid fossil fuel, coal and biomass burning. (a) results from the MCMC3 model, including C3 plants as biomass, coal and liquid fossil fuel; (b) Impact of C4 plants burning on EC source apportionment is tested

5    by including C4 biomass into the calculations (MCMC4). Including C4 plants in calculation does not affect the contribution of biomass burning to EC.  The relative fraction of C3 and C4 plants in biomass burning is shown in Fig. S8. In winter, the sample taken on Chinese New Year eve (25 January 2009) was excluded.




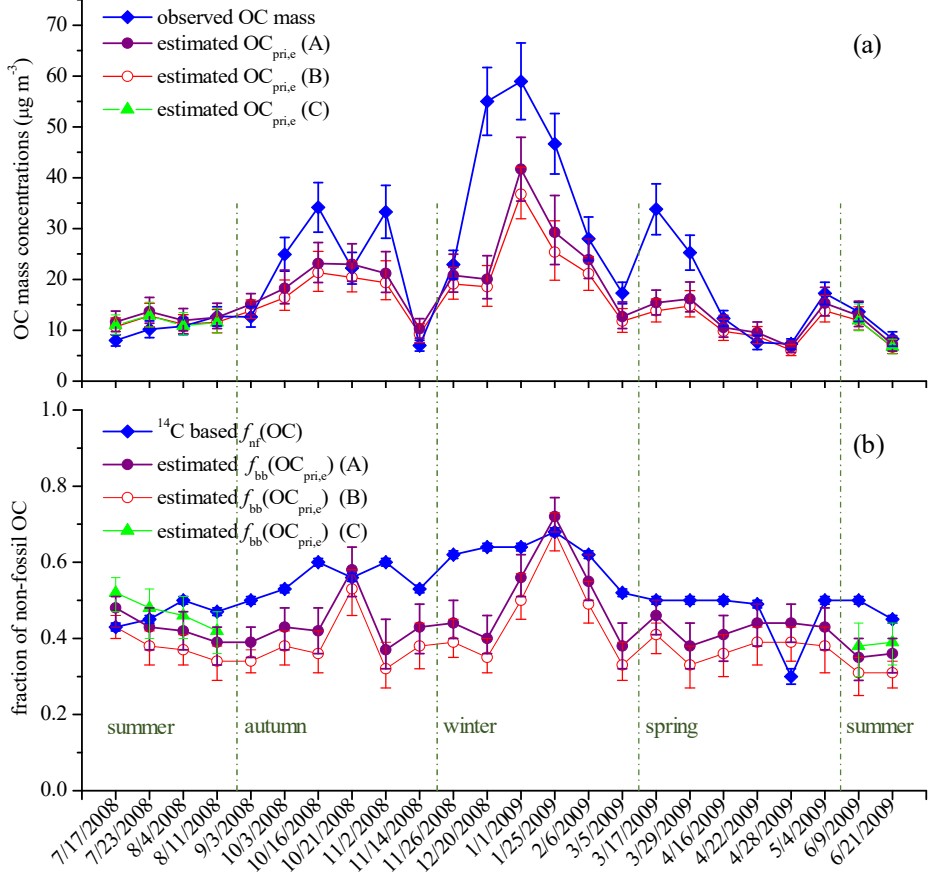

**Figure 5.** Estimated primary OC based on MCMC4 results. (a) measured OC concentrations (blue line and diamond symbols) with observational uncertainties (vertical bar) and estimated OC mass ($OC_{pri,e}$, circle and triangular symbols) from apportioned EC and OC/EC ratios for different sources (Eq. (10)). (b) [14]C-based fraction of non-fossil OC ($f_{nf}(OC)$) and modelled non-fossil fraction in $OC_{pri,e}$ ($f_{bb}(OC_{pri,e})$) derived from Eq. (11). Interquartile range (25th-75th percentile) of the median $OC_{pri,e}$ and $f_{bb}(OC_{pri,e})$ are shown in purple (A), red (B) and green (C) vertical bars. "A" and "B" denotes different OC/EC ratios applied to primary biomass burning emissions ($r_{bb}$): A. $r_{bb}$ = 5 (3–7, minimum-maximum); B. $r_{bb}$ = 4 (3–5); "C" denotes 80 % $r_{liq.fossil}$ applied in summer with $r_{bb}$= 5. $f_{nf}(OC)$ uncertainties are shown but not visible.




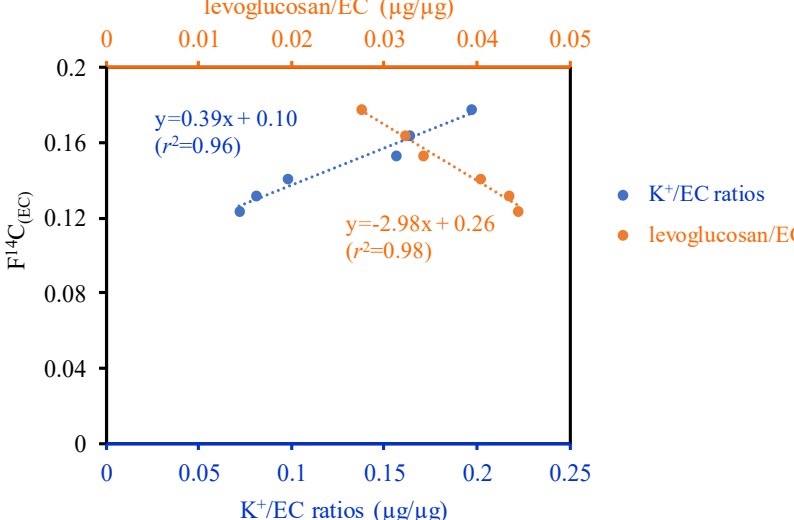

**Figure 6.** Correlation between $F^{14}C_{(EC)}$ and $K^+$/EC ratios and levoglucosan/EC ratios in summer. Data in other seasons are presented in Fig. S11.





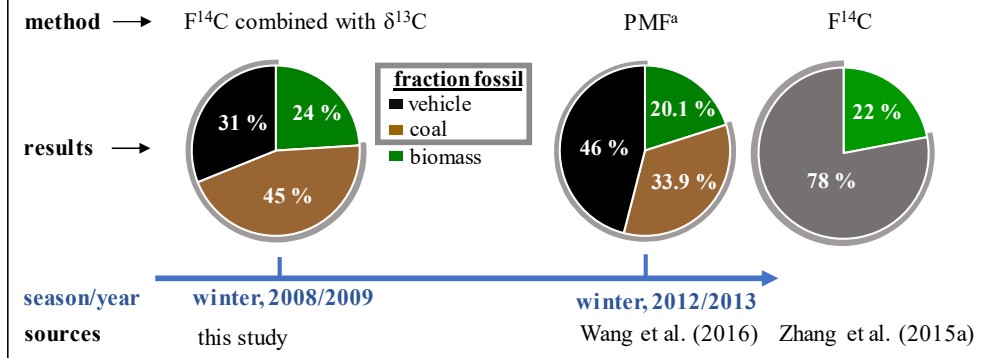

**Figure 7.** Comparison of EC source apportionment in winter 2008/2009 with two other studies in winter 2012/2013 at the same sampling site. [a]positive matrix factorization (PMF) receptor model simulation.





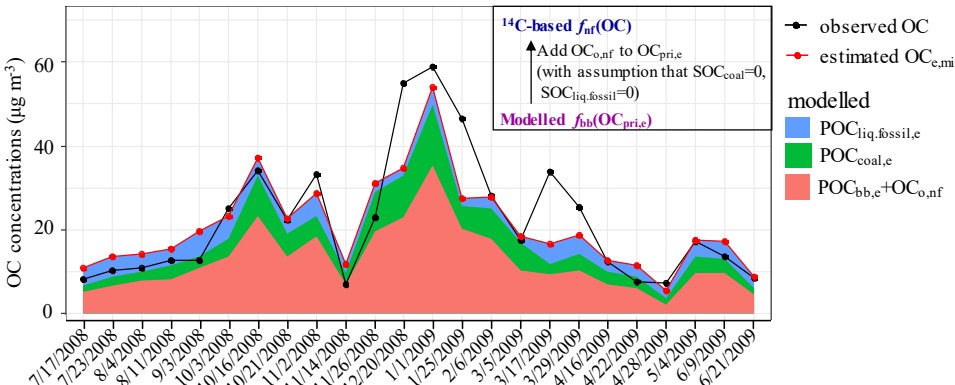

**Figure 8.** Observed and estimated OC concentrations. Modelled $OC_{e,min}$ is the sum of $OC_{pri,e}$ and $OC_{o,nf}$. $OC_{o,nf}$ accounts for the differences between $f_{nf}(OC)$ and $f_{bb}(OC_{pri,e})$, with an unrealistic assumption of no secondary fossil OC, leading to minimum addition to $OC_{pri,e}$. Coral area shows the $POC_{bb,e}$ and $OC_{o,nf}$, green area the $POC_{coal,e}$ and blue area the $POC_{liq.fossil,e}$. Estimation is based on MCMC4 results for EC source apportionment and primary OC/EC ratios corresponding to case (A) in Fig. 5.