# Peer review of "Source apportionment of carbonaceous aerosols in Xi'an, China: insights from a full year of measurements of radiocarbon and the stable isotope 13C"

_Atmospheric Chemistry and Physics, 2018_

## Referee Comment (RC1) · Anonymous Referee #2 · 1 May 2018

General comments This paper reports the results of source apportionment based on a 1-year campaign in China. Besides the specific results, the paper presents an interesting methodology, based on the synergic use of radioactive and stable carbon isotopes. The paper ends with an open question, but this may be the trigger to foster new research. Therefore I think the paper is worth the publication. Specific comments - Introduction, page 2 lines 16-18: "The 14C content of an aerosol sample is usually reported relative to an oxalic acid standard, and expressed as fraction modern (F14C). 14C content of the standard is related to the unperturbed atmosphere in the reference year of 1950 (Mook and van der Plicht, 1999; Reimer et al., 2004)"; please change to "The 14C content of an aerosol sample is usually reported relative to an oxalic acid

standard, and expressed as fraction modern (F14C). 14C content of the standard is related to the unperturbed atmosphere in the reference year of 1950 (Mook and van der Plicht, 1999; Reimer et al., 2004), and this is usually done with/by means of/similar a standard"; this sentence is more correct and actually describes much better the definition formula in the following line. - Introduction, page 2 lines 21-22: the assumptions on F14C are reported in a far too simplistic in several points (later on they become simple "conversion factors"). I suggest the authors to better introduce these quantities to facilitate the reader and also the writing of the following sections. The same for d13C: it is introduced in the following sections, it is true, but at this stage there are already some sentences maybe not clear to readers not too familiar with stable carbon isotopes (e.g. "signature depleted", but it is not clear with respect with which reference.) - Introduction, page 3 line 7 and table S1: actually, source signatures are not that well distinct, as they have overlaps: may the authors discuss deeper this point? - Sampling, page 3 line 28: why was the sampling time chosen to be 10 am to 10 am next day? Due to manual change? How long were the samples kept inside the sampler after sampling? - Sampling, page 4 line 4: citations missing ("previous studies" are not cited) - Stable carbon isotope (13C) analysis of OC and EC, page 4: some more details on the analysis would be welcome. Further, the title is maybe misleading, as it suggests that only 13C is measured, while I guess that also 12C is assessed for determining 13C/12C ratios. - Radiocarbon (14C) measurement of OC and EC, page 5, line 9: "Two standards with known 14C content are analyzed as quality control: an oxalic acid standard and a graphite standard.": maybe I did not understand, but I believe these standards are respectively for normalization and blank evaluation; if this is correct, they cannot be defined as "for quality control". In case further standards are measured as unknown, these can be defined "for quality control". - Radiocarbon (14C) measurement of OC and EC, page 5, lines 11-13 and 24-25: there is a repetition of the information, and actually not completely in the same way: please correct it. Further, is this contamination modern or fossil? - Source apportionment methodology using 14C: as already aforementioned, the use of the definition "conversion factors" is misleading,

as they have a physical meaning (as it is clear at page 6, line 24). Authors should introduce this concept earlier in the text, so that they can also explain the use of different values for their "conversion factors". This would definitely make the paper easier to read. - Temporal variation of fossil and non-fossil fractions of OC and EC, page 7: levoglucosan, hopanes and picen are cited for the first time, with no reference to S2, where the measurements are described. The existence of ancillary/additional measurements deserves to be introduced as part of the methodology. - 13C signature of OC and EC, page 9, line 12: "$\delta$13COC was in general similar to $\delta$13CEC": this means that the biogenic source is roughly negligible: can the author comment with finding also in relation to the radiocarbon measurement results? - I suggest moving section 4.4 straight after 4.2, as this discussion follows directly from the last sentences of 4.2. - Changes in emission sources in Xi'an, China (2008/2009 vs. 2012/2013), pages 15-16: the cited papers taken for comparison focus, respectively, on a big haze episode and on an intensive campaign (2 winter months), and not on a campaign aiming at being representative for a year, therefore I think this comparison is not very useful. Further, contributions are roughly the same within the uncertainties. - Supplement, table S1: far as I get, the reported interval for C4 plants is wide as different plants (corn, sugar cane, grass and maybe more) have different signatures: why do the authors "decrease" this range to -16.4 +- 1.4 permil? (Futher, please pay attention to number of digits, e.g. -23.4 +-1.3 and not -23.38 +-1.3)

---

## Referee Comment (RC2) · Anonymous Referee #3 · 18 Jul 2018

General comments:

This manuscript reported datasets of carbon isotopes (13C and 14C) of OC and EC in a major inland city of China, Xi'an, during one-year sampling, which were used to study the source apportionment of carbonaceous aerosols by combining 13C and 14C with Bayesian Markov chain Monte Carlo (MCMC) scheme. The data and methodology are reliable and novel. This paper shed some new light on the source apportionment of carbonaceous aerosols by distinguishing coal and liquid fossil fuel contributions to EC, C3 and C4 plant to biomass burning. The paper is relatively well-written, and it should be acceptable for publication after some moderate to major revision.

Major points: 1. Be clear re mean or median values of source contribution, E.g., P1/L23, the 45% and 31% are median values in Figure 4. Need to be consistent in the manuscript. 2. The flow of introduction is not well organized, and some part of the 14C introduction should be moved to the method part. I suggest re-organization and strengthening of the scientific objectives of this study in the introduction. In addition, you need to explain why we need to further distinguish the coal and fuel combustion in EC but not OC? Fossil sources contribute averagely 46% to OC based on your results, so it is important. 3. Provide more clear details of blank/contamination evaluation for 14C analysis, instrumental analytical precision and mention of source markers (S2) in the method part. 4. The result section needs to be better structured and written. There are many parts, specially, sections 3.4 and 3.5, should be moved to discussion; and formulas could be moved to methods part. 5. There are many comparisons without in-depth discussion in the discussion section. And comparison among different methods and different climate event seems not reasonable in 4.3. 6. The Conclusion part was too long. I suggest summarizing the key points.

Minor points and suggestions: Introduction: P2/L1-10 This part didn't emphasize the importance of carbonaceous aerosol very well. And the structure and description are very similar to the second paragraph of the introduction from Zhang et al., (2015a, Atmos. Chem. Phys). Need to revise. P2/L16: The definition and expression of fraction modern is not explicit. "The 14C content" is not a ratio as "fraction modern". I don't think the standard need normalize for fractionation to 13C=-25‰ You can refer to Stuiver and Polach (1977) and modify this sentence. I also suggest move this "14C result report" part to the Methods part. P2/L25 Clarify which kind of 14C studies only have two datasets. . . . . . . seasonal variations? TC or other? For example, Zhang et al. (2015b, EST) also reported annual and seasonal variations of EC in Beijing. P2/L27-30 It's better to introduce 13C first and introduce 14C as a novel tool. The 13C values of distinct sources you listed overlap with each other. P3/L7: same question as above. P3/L9-13 You need to provide 13C numbers for example to make readers to remember the trend between different processes.

Methods: P4/L1 replace "pre-fired" with "pre-baked" P4/L3 what's the standard for season's classification? Reference? How can you classify autumn to middle day? P4/L15 instrumental analytical precision? Do you have field blank? P5/L9-13 and L21-25 The whole blank/contamination should include all blank produced during experimental process and it is very important to know if the contamination is modern or fossil for 14C analysis. For combustion process, are the stds modern and fossil? Give the mass and F14C value of the stds. Also, provide the F14C value of combustion contamination. For 14C analysis, give the mass and F14C value of contamination you got in this study. Did you have F14C value of blank filter? Provide which blank you used to correct your 14C data. P6/L18 Is the "fbb" the same as "fbb" at L1?

Results P7/L10 The number of seasonal samples in Table S2 is not the same as in sampling part, need to clarify this inconsistency. P8/L20 unify the decimal place to one in the whole manuscript. P8/L30 Is it possible that combustion of a mixture of C4 and C3 plants or liquid fuel will results in the 13CEC values of around -24‰ P9/L26 what do the grey and dark green rectangle mean in Figure 3? P10/L1 Sections 3.4.1 is not real results of this study P10/L10 Provide the four-source calculation formula in 3.4.2 section and change the name "MCMC3" in 2.6 section. P10/L28 "5 times less than in summer" should use "lower than" P10/L29-32 The proportion of liquid fossil fuel combustion in winter (more coal burning) lower than summer (more traffic emission) make sense, why is not your expectation? P11/L16 mean or median? P12/L14 Does the OCo, nf mean observed non-fossil OC?

Discussion P13/L16 Add "Characteristics" in front of 4.1 title. P13/L17 Clarify which ffossil (EC) you used in comparison, the MCMC4 or F14C? Because others use the F14C deduced values. P13/L25 should be 76% as shown in Figure 4. P15/L4-8 Discussion here is not very convincing. The contribution of biomass burning to EC is the lowest in summer, but the highest contribution of biomass burning to EC occurred in winter (most corn stalk burning in winter, Figures 4 and S5), why no significant correlation was found in winter? P15/L16-20 I don't think it's reasonable to directly compare

results of different methods, e.g., you got contribution of biomass burning to EC by MCMC4 with 4 sources while Zhang et al. (2015) got the fraction by 2 sources. Furthermore, taking into account of the error bar, the fraction fossil (76%)/ biomass (24%) of this study are the same to Zhang et al. (2015). Finally, Zhang et al. (2015) studied samples during the extreme winter haze episode of 2013. P15/L23-28 The same question as above. Because the PMF model didn't use 14C, is this reasonable for comparison? P16/L7 and Figure 7 clarify the relationship between vehicular emissions and liquid fossil fuel combustion somewhere before discussion. P17/L3 Will the biogenic emission to OC result in lower 13C values than EC?

Conclusions condense and summarize the key points of this study in this part.

References Check carefully the papers of the same author, e.g., you have two Zhang et al. (2015a) and where is Zhang et al. (2014a)? I think in section 4.3, you refer to Zhang et al., (2015a, Atmos. Chem. Phys.,)

---

## Author Comment (AC1) · 12 Sep 2018

**Response to reviewer 2**

**General comments** This paper reports the results of source apportionment based on a 1-year campaign in China. Besides the specific results, the paper presents an interesting methodology, based on the synergic use of radioactive and stable carbon isotopes. The paper ends with an open question, but this may be the trigger to foster new research. Therefore, I think the paper is worth the publication.

**Response**: We appreciate the reviewer's thoughtful and valuable comments, which are very helpful for revising and improving our manuscript. We have carefully addressed the reviewer's comments. Below are point-to-point responses.

**Specific comments**

1) Introduction, page 2 lines 16-18: "The $^{14}$C content of an aerosol sample is usually reported relative to an oxalic acid standard and expressed as fraction modern (F$^{14}$C). $^{14}$C content of the standard is related to the unperturbed atmosphere in the reference year of 1950 (Mook and van der Plicht, 1999; Reimer et al., 2004)"; please change to "The $^{14}$C content of an aerosol sample is usually reported relative to an oxalic acid standard and expressed as fraction modern (F$^{14}$C). $^{14}$C content of the standard is related to the unperturbed atmosphere in the reference year of 1950 (Mook and van der Plicht, 1999; Reimer et al., 2004), and this is usually done with/by means of/similar a standard"; this sentence is more correct and actually describes much better the definition formula in the following line.

**Response**: We agree with the reviewer, that this formulation is not entirely clear. Text and Eq. (1) is revised to clarify the definition of F$^{14}$C as shown in the revised manuscript (marked-up copy, page 2, line 26–31; page 3, line 1–3). We tried to be even more specific and refer now concretely to OxII standard multiplied by 0.7459.

2) Introduction, page 2 lines 21-22: the assumptions on F$^{14}$C are reported in a far too simplistic in several points (later on they become simple "conversion factors"). I suggest the authors to better introduce these quantities to facilitate the reader and also the writing of the following sections. The same for δ$^{13}$C: it is introduced in the following sections, it is true, but at this stage there are already some sentences maybe not clear to readers not too familiar with stable carbon isotopes (e.g. "signature depleted", but it is not clear with respect with which reference.)

**Response**: We introduce the F$^{14}$C of contemporary sources in details as follows (page 3 line 5–14):

"However, F$^{14}$C values of the contemporary (or non-fossil) carbon sources are bigger than 1 due to the nuclear bomb tests that nearly doubled the $^{14}$CO$_2$ in the atmosphere in the 1960s and 1970s. Currently, F$^{14}$C of the atmospheric CO$_2$ is approximately 1.04 (Levin et al., 2010).This value is decreasing every year, because the $^{14}$CO$_2$ produced by bomb testing is taken up by oceans and the biosphere and diluted by $^{14}$C-free CO$_2$ produced by fossil fuel burning . For biogenic aerosols, aerosols emitted from cooking as well as annual crop, the F$^{14}$C is close to the value of current atmospheric CO$_2$. F$^{14}$C of

wood burning is higher than that, because a significant fraction of carbon in the wood burned today was fixed during times when atmospheric $^{14}C/^{12}C$ ratios were substantially higher than today. Estimates of $F^{14}C$ for wood burning are based on tree-growth models (e.g., Lewis et al., 2004; Mohn et al., 2008) and found to range from 1.08 to 1.30 (Szidat et al., 2006; Genberg et al., 2011; Gilardoni et al., 2011; Minguillón et al., 2011; Dusek et al., 2013)."

The new citations are included in the revised reference list.

Text in the method section is revised to clarify the "conversion factor" by adding their physical meanings, as follows (page 7 line 11–13; line 18–22):

"$F^{14}C$ of EC ($F^{14}C_{(EC)}$) was converted to the fraction of biomass burning ($f_{bb}(EC)$) by dividing with $F^{14}C$ of biomass burning ($F^{14}C_{bb}= 1.10 \pm 0.05$; Lewis et al., 2004; Mohn et al., 2008; Palstra and Meijer, 2014) given that biomass burning is the only non-fossil source of EC, to eliminate the effect from nuclear bomb tests in the 1960s." (page 7 line 11–13)

"$F^{14}C$ of OC ($F^{14}C_{(OC)}$) was converted to the fraction of non-fossil ($f_{nf}(OC)$) by dividing the $F^{14}C$ of non-fossil sources including both biogenic and biomass burning ($F^{14}C_{nf} =1.09 \pm 0.05$; Lewis et al., 2004; Levin et al., 2010; Y. Zhang et al., 2014a). The lower limit of 1.04 corresponds to current biospheric sources as the source of OC, the upper limit corresponds to burning of wood as the main source of OC, with only little input from annual crops." (page 7 line 18–22)

$F^{14}C_{bb}$ (1.10 ± 0.05) for EC is slightly smaller than $F^{14}C_{nf}$ (1.09 ± 0.05) for OC, because except biomass burning, biogenic emissions also contribute to OC, but have a smaller $F^{14}C$ than that of biomass burning.

For stable carbon isotopes, we change "the stable carbon isotope ($^{13}C$, expressed as $\delta^{13}C$)" to "the stable carbon isotope composition (namely the $^{13}C/^{12}C$ ratio, expressed as $\delta^{13}C$ in Eq. (2))" in the revised text (page 3 line 19–20). $\delta^{13}C$ is useful to distinguish sources of aerosols. The distinction is possible because for example, $\delta^{13}C$ values of carbon from coal combustion are less depleted (or enriched, i.e., enrichment in $^{13}C/^{12}C$ ratio) compared to the aerosol carbon emitted by other sources, for example, liquid fossil fuel combustion (-28 ‰ to -24 ‰) and C3 plants burning (-35 ‰ to -24 ‰) (Andersson et al. (2015) and references therein). The "signature depleted" is based on the comparison between different emission sources. To clarify, we explain the term "enriched" and "depleted" $\delta^{13}C$ as follows:

"Different emission sources have their own source signature: carbonaceous aerosol from coal combustion is enriched in $^{13}C$ (i.e., has higher $\delta^{13}C$ values of ~ -25 ‰ to -21 ‰) compared to aerosol from liquid fossil fuel combustion ($\delta^{13}C$ ~ -28 ‰ to -24 ‰) and from burning of C3 plants ($\delta^{13}C$ ~ -35 ‰ to -24 ‰) (Andersson et al. (2015) and references therein)." (page 3 line 21–25)

3) Introduction, page 3 line 7 and table S1: actually, source signatures are not that well distinct, as they have overlaps: may the authors discuss deeper this point?

**Response:** We appreciate this point. If it is ok with the reviewer, we will put the deeper discussion into the methods section 2.6, where we introduce the use of $^{13}$C data for source apportionment and can further elaborate on the consequences of the overlapping signatures.

Though with some overlaps, different emission sources have their own source signature. To avoid misunderstanding, we change "Different emission sources have their *distinct* source signature" in the introduction section to "Different emission sources have their _own_ source signature" (page 3 line 22).

Source signatures of $\delta^{13}$C presented in Table S1 (shown as mean ± standard deviation) are established for the purpose of the source apportionment for EC (see revised Sect. 2.6). For burning C3 plants, coal and liquid fossil fuel, their $\delta^{13}$C source signatures for EC are fully complied and established in Andersson et al. (2015) by a thorough literature search. In brief, the mean and standard deviation for $\delta^{13}$C endmembers for the different sources are estimated as the average and standard deviation of the different data sets, respectively. For burning corn stalk residues in the study area, our $\delta^{13}$C values of aerosols from corn stalk burning were complied in Fig. S6 and its source signatures are established as -16.4 ± 1.4 ‰ (mean ± standard deviation), as described in Sect 4.3.1 (Sect. 3.4.1 in the draft manuscript) and as addressed in Question 14. To clarify, we add ranges of source signatures of $\delta^{13}$C into Supplemental Table S1, besides the established mean ± standard deviation for source apportionment of EC.

The source endmembers for $\delta^{13}$C are less well-constrained than for $F^{14}$C, as $\delta^{13}$C varies with fuel types and combustion conditions. For example, $\delta^{13}$C values for liquid fossil fuel combustion overlaps with $\delta^{13}$C values for both coal and C3 plant combustion. In this study, to account for the variability of the isotope signatures of $\delta^{13}$C and $F^{14}$C from the different sources, the Bayesian Markov chain Monte Carlo techniques (MCMC) were used as explained in the Method section. Uncertainties of both source endmembers for each source and the measured ambient $\delta^{13}C_{EC}$ and $F^{14}C_{(EC)}$ are propagated. These serve as input of MCMC to estimate the source contributions to EC. The MCMC results are the posterior probability density functions (PDF) for the relative contributions from the sources (Fig. S7, Fig. S8). The PDF of the relative source contributions of liquid fossil fuel combustion (vehicle) and coal combustion is skewed. By contrast, the PDF of the relative source contributions of biomass burning is symmetric as it is well-constrained by $F^{14}$C (Fig. S8(a)). In this study, the median with its interquartile range was used to give the estimates of the contribution of any particular source to EC throughout the manuscript (e.g., Table 1, Sect. 4.3.2).

This point is clarified in the Method Sect. 2.6 by adding the following (page 8 line 14-22):

"$\delta^{13}C_{bb}$, $\delta^{13}C_{liq.fossil}$ and $\delta^{13}C_{coal}$ are the $\delta^{13}$C signature emitted from biomass burning, liquid fossil fuel combustion and coal combustion, respectively. The means and the standard deviations for $\delta^{13}C_{bb}$ (-26.7 ± 1.8 ‰ for C3 plants, and -16.4 ± 1.4 ‰ for corn stalk), $\delta^{13}C_{liq.fossil}$ (-25.5 ± 1.3 ‰) and $\delta^{13}C_{coal}$ (-23.4 ± 1.3 ‰) are presented in Table S1

(Andersson et al., 2015 and reference therein; Sect. 4.3.1), and serves as input of MCMC. The source endmembers for $\delta^{13}C$ are less well-constrained than for $F^{14}C$, as $\delta^{13}C$ varies with fuel types and combustion conditions. For example, the range of possible $\delta^{13}C$ values for liquid fossil fuel combustion overlaps to a small extent with the range for coal combustion, although liquid fossil fuels are usually more depleted than coal. The MCMC technique takes into account the variability in the source-signatures of $F^{14}C$ and $\delta^{13}C$ (Table S1), where $\delta^{13}C$ introduces a larger uncertainty than $F^{14}C$. Uncertainties of both source endmembers for each source and the measured ambient $\delta^{13}C_{EC}$ and $F^{14}C_{(EC)}$ are propagated."

Further, to give the readers an idea of this point, we also changed the sentence in the Introduction section "Further, $\delta^{13}C$ of EC allows separation of fossil sources into coal and liquid fossil fuel burning (Andersson et al., 2015; Winiger et al., 2015, 2016), as they have their distinct source signatures" to:

"Further, $\delta^{13}C$ of EC allows separation of fossil sources into coal and liquid fossil fuel burning (Andersson et al., 2015; Winiger et al., 2015, 2016), due to their different source signatures. Typical $\delta^{13}C$ values for EC from previous studies are summarized in Table S1." (page 3 line 30–32).

4) Sampling, page 3 line 28: why was the sampling time chosen to be 10 am to 10 am next day? Due to manual change? How long were the samples kept inside the sampler after sampling?

**Response:** The sampling time was long chosen to be 10:00 a.m. to 10:00 a.m. the next day (e.g., Cao et al., 2009; Han et al., 2016) due to manual change and safety reasons in accessing the site at midnight.

Only one filter can be loaded into the sampler, so we took the filter out of the sampler after 24 hr sampling and did not keep it in the sampler. The revised text (page 5 line 1) shows that:

"After sampling, we immediately removed the filter from the sampler. All filters were packed in pre-baked aluminum foils, sealed in polyethylene bags and stored at -18 °C in a freezer"

5) Sampling, page 4 line 4: citations missing ("previous studies" are not cited)

**Response:** Citations are added on page 5 line 2–3.

6) Stable carbon isotope ($\delta^{13}C$) analysis of OC and EC, page 4: some more details on the analysis would be welcome. Further, the title is maybe misleading, as it suggests that only $^{13}C$ is measured, while I guess that also $^{12}C$ is assessed for determining $^{13}C/^{12}C$ ratios.

**Response:** More details on the $\delta^{13}C$ analysis are added as shown in the revised manuscript (page 5 line 21–30; page 6 line 1–7).

The title is changed to "2.3 Stable carbon isotopic composition of OC and EC", to indicate that $^{13}C/^{12}C$ ratios are determined, not only $^{13}C$. (page 5 line 20)

7) Radiocarbon ($^{14}$C) measurement of OC and EC, page 5, line 9: "Two standards with known $^{14}$C content are analyzed as quality control: an oxalic acid standard and a graphite standard.": maybe I did not understand, but I believe these standards are respectively for normalization and blank evaluation; if this is correct, they cannot be defined as "for quality control". In case further standards are measured as unknown, these can be defined "for quality control".

**Response:** Two standards with known $^{14}$C content that extracted using our aerosol combustion system (ACS) are analyzed to assess the contamination introduced by the combustion process, and they are treated exactly like the samples (e.g., normalized to the oxalic acid OXII calibration material). The measured deviation in F$^{14}$C from the nominal values is caused by contamination introduced by the combustion process. The contamination is assessed but not used for further data correction, because the contamination is relatively small (typically below 1.5 μgC per combustion) compared the sample sizes (ranging between 50 and 270 μgC). To clarify, the revised texts show:

> "Two standards with known $^{14}$C content are combusted as quality control for the combustion process: an oxalic acid standard and a graphite standard. Small amounts of solid standard material are directly put on the filter holder of the combustion tube and heated at 650 °C for 10 min. In the further $^{14}$C analysis, the CO$_2$ derived from combustion of the standards is treated exactly like the samples. Therefore, the contamination introduced by the combustion process can be estimated from the deviation of measured values (Table S9) from the nominal values of the standards. The contamination is below 1.5 μgC per combustion, which is relatively small compared the samples ranging between 50 and 270 μgC in this study." (page 6, line 19–24)

8) Radiocarbon ($^{14}$C) measurement of OC and EC, page 5, lines 11-13 and 24-25: there is a repetition of the information, and actually not completely in the same way: please correct it. Further, is this contamination modern or fossil?

**Response:** $^{14}$C measurements of aerosol samples are subject to contaminations, which can be introduced during the combustion process using ACS, or during the graphitization and AMS measurements. For contamination caused by the combustion process, it is already explained in the response to Question 7. Here we addressed the contamination during the graphitization and AMS measurement, thus it is not a repetition of the information.

F$^{14}$C of aerosols samples was corrected for contamination that occurred during graphitization and AMS measurement. For AMS measurements, samples are usually analyzed together with varying amounts of reference material covering the range of sample mass. Two such materials with known $^{14}$C content are used: the oxalic acid OXII calibration material (F$^{14}$C = 1.3406) and a $^{14}$C-free CO$_2$ gas (F$^{14}$C = 0). Contamination during the graphitization and AMS measurement results into the differences between measured and nominal F$^{14}$C values. The magnitude of these deviations can be used to quantify the contamination with fossil carbon (F$^{14}$C$_F$ = 0) and modern carbon (F$^{14}$C$_M$ = 1), which in turn are used for correcting the sample values (de Rooij et al., 2010; Dusek et al., 2014).

The contamination with fossil carbon and modern carbon is quantified using isotope mass balance (Dusek et al., 2014):

$$F^{14}C_m \cdot M_m = F^{14}C_{st} \cdot M_{st} + F^{14}C_F \cdot M_F + F^{14}C_M \cdot M_M. \qquad (S1)$$

$M_m$ and $M_{st}$ stand for the experimentally determined mass and the mass of reference materials either the oxalic acid OXII calibration material ($F^{14}C = 1.3406$) or a $^{14}C$-free $CO_2$ gas ($F^{14}C = 0$) with a unit of µgC, respectively. $F^{14}C_m$ and $F^{14}C_{st}$ represent the experimentally determined $F^{14}C$ measured by AMS and nominal $F^{14}C$ of reference materials (Table S9).

The relationships among all masses are described as Eq. (S2):

$$M_m = M_{st} + M_F + M_M, \qquad (S2)$$

where $M_M$ is calculated using Eq. (S1) by substituting $F^{14}C_{st} = 0$ for a $^{14}C$-free $CO_2$ gas as:

$$M_M = F^{14}C_m \cdot M_m. \qquad (S3)$$

Substitute $F^{14}C_{st} = 1.3406$ for OXII and the derived $M_M$ from Eq. (S3), $M_F$ is derived by combining Eq. (S1) and Eq. (S2) as:

$$M_F = ((1.3406 - F^{14}C_m) \cdot M_m - (1.3406 - 1) \cdot M_M)/1.3406. \qquad (S4)$$

$M_M$ and $M_F$ is calculated by applying Eq. (S3) and Eq. (S4), and they are mass dependent. The modern carbon contamination ($M_M$) is between 0.35 and 0.50 µg C, and the fossil carbon contamination ($M_F$) is around 2 µg C for sample bigger than 100 µgC.

In the revised manuscript, we add the detailed calculation of modern and fossil contamination in the supplemental material (Supplemental S3). The revised manuscript adds:

> "The differences between measured and nominal $F^{14}C$ values are used to correct the sample values (de Rooij et al., 2010; Dusek et al., 2014) for contamination during graphitization and AMS measurement (Supplemental S3). The modern carbon contamination is between 0.35 and 0.50 µg C, and the fossil carbon contamination is around 2 µg C for sample bigger than 100 µgC." (page 7 line 5–9)

9) Source apportionment methodology using $^{14}C$: as already aforementioned, the use of the definition "conversion factors" is misleading, as they have a physical meaning (as it is clear at page 6, line 24). Authors should introduce this concept earlier in the text, so that they can also explain the use of different values for their "conversion factors". This would definitely make the paper easier to read.

**Response:** As addressed in the response to Question 2, the revised manuscript uses the physical meanings (i.e., $F^{14}C$ of biomass burning ($F^{14}C_{bb}$) for EC, $F^{14}C$ of non-fossil sources ($F^{14}C_{nf}$) for OC) instead of "conversion factors" (page 7 line 11–13, line 18–22). This concept is added in the revised Introduction section (page 3 line 5–13).

10) Temporal variation of fossil and non-fossil fractions of OC and EC, page 7: levoglucosan, hopanes and picene are cited for the first time, with no reference to S2, where the measurements

are described. The existence of ancillary/additional measurements deserves to be introduced as part of the methodology.

**Response:** We add briefly the measurement of source markers in the Sect. 2.2 as part of the methodology and change the title to "2.2 Organic carbon (OC), elemental carbon (EC) and source markers measurement". S2 is kept in the supplemental material for details on the measurements.

Further, S2 is mentioned (page 9 line 17) when organic markers (levoglucosan, hopanes and picene) are cited in the main text for the first time.

11) [13]C signature of OC and EC, page 9, line 12: "$\delta^{13}C_{OC}$ was in general similar to $\delta^{13}C_{EC}$": this means that the biogenic source is roughly negligible: can the author comment with finding also in relation to the radiocarbon measurement results?

**Response:** $\delta^{13}C_{OC}$ was in general similar to $\delta^{13}C_{EC}$: it suggests that biogenic OC is probably not very important, as could be expected from the high TC concentrations and strong anthropogenic sources. This can be true, as we would expect a bit different $\delta^{13}C_{OC}$ from $\delta^{13}C_{EC}$ if biogenic sources play an important role on OC.

In light of [14]C, we still measure a considerable fraction of non-fossil OC, and it would seem that this is more related to the biomass burning. Or, if there is biogenic OC, but by chance their $\delta^{13}C$ signatures are relatively similar with those for the source mixture of EC, which is not very likely.

The following statements were added to the revised manuscript:

"$\delta^{13}C_{OC}$ was in general similar to $\delta^{13}C_{EC}$. This suggests that biogenic OC is probably not very important, as could be expected from the high TC concentrations. [14]C analysis indicates a considerable fraction of non-fossil OC than non-fossil EC, and it would seem that this is mainly related to the biomass burning, which has higher OC/EC ratios than fossil fuel burning. If the contribution of biogenic OC plays an important role, then the biogenic $\delta^{13}C$ signatures should be relatively similar to the source mixture of EC, which is rather unlikely, especially as this source mixture is not constant" (page 11 line 5–9)

12) I suggest moving section 4.4 straight after 4.2, as this discussion follows directly from the last sentences of 4.2.

**Response:** The order of the two sections has been changed. The order of Fig. 7 and Fig. 8 is also changed accordingly.

13) Changes in emission sources in Xi'an, China (2008/2009 vs. 2012/2013), pages 15-16: the cited papers taken for comparison focus, respectively, on a big haze episode and on an intensive campaign (2 winter months), and not on a campaign aiming at being representative for a year, therefore I think this comparison is not very useful. Further, contributions are roughly the same within the uncertainties.

**Response:** We did the whole year measurement in Xi'an for the year 2008/2009. In this study, we selected samples with varying PM$_{2.5}$ mass and carbonaceous aerosols for [14]C analysis. The

selected samples cover periods of low, medium and high $PM_{2.5}$ concentrations to get samples representative of the various pollution conditions that did occur in each season. Here we only compare the winter season due to the limited source apportionment results for EC in Xi'an. For Xi'an, we see from this study (see Fig. 1(a), except the Chinese New Year eve, which is not included in the comparison) but also some studies in preparation that the $^{14}C$ values do not change very much between polluted days and clean days. In addition, two months (the intensive campaign by Wang et al. (2016)) almost cover the whole winter (in total, 3 months). Thus, we think that it makes sense to compare the results from this study with the two cited paper.

For EC source apportionment, it is noted that the quartile ranges for 2008/2009 values (this study) overlaps ranges for 2012/2013 values (average ± SD). Compared to the uncertainties of radiocarbon measurements, the uncertainties of PMF results are always larger, making the overlapped ranges very likely. However, comparing the probability distribution functions for both cases give a more complete picture. Figure S14 and Figure S15 shows the probability density functions (PDF) of the relative source contributions to EC from coal combustion and vehicle emissions, respectively. Results from this study for the year 2008/2009 are shown in grey (this is also shown in the original Fig. S8), and from Wang et al. (2016) for the year 2012/2014 shown in yellow. For the PDF by Wang et al. (2016), we assume normal distribution as their source apportionment results are not known and given in the form of average ± SD. As shown in Fig. S14 and Fig. S15, though with some overlaps, the PDF of the relative source contribution of coal combustion (vehicle emissions) does clearly shift to the lower side (higher side) from the year 2008/2009 to 2012/2013. With the current inherent uncertainties in these two states of the art source apportionment methods it will not be possible to draw more firm conclusions than that these probability distributions show a certain trend, despite some possible overlap.

We also have some additional observation data to support the conclusion as discussed in Sect. 4.6. The decreased contributions of coal combustion are also evidenced from the declined enrichment factor of As and Pb, indicators of coal combustion. Increasing vehicular emissions is supported by the increasing level of $NO_2$, an indicator for the contribution of vehicular emissions.

In the revised manuscript, the above discussion is added in the section of Changes in emission sources in Xi'an, China (2008/2009 vs. 2012/2013) (page 18 line 25–27; page 19, line 7–14). The Fig. S14 and Fig. S15 is added to the supplemental material.

[Figure]

**Figure S14.** Probability density functions (PDF) of the relative source contributions of coal combustion to EC in winter in the year 2008/2009 (this study, shown in grey; this is also shown in Fig. S8) and 2012/2013 by Wang et al. (2016), shown in yellow.

[Figure]

**Figure S15.** Probability density functions (PDF) of the relative source contributions of vehicle emissions to EC in winter in the year 2008/2009 (this study, shown in grey; this is also shown in Fig. S8.) and 2012/2013 by Wang et al. (2016), shown in yellow

14) Supplement, table S1: far as I get, the reported interval for C4 plants is wide as different plants (corn, sugar cane, grass and maybe more) have different signatures: why do the authors "decrease" this range to -16.4 ± 1.4 per mil? (Further, please pay attention to number of digits, e.g. -23.4 ±1.3 and not -23.38 ±1.3)

**Response:**  In this study, $\delta^{13}$C for corn stalk is used as it is the dominant C4 plant in Xi'an and its surrounding areas (Sun et al., 2017; Zhu et al., 2017), with little sugarcane and other C4 plants as explained in Sect. 4.3.1 where details on selection of $\delta^{13}$C endmembers for C4 plants in the study area are described. $\delta^{13}$C values of aerosols from corn stalk burning were compiled from literature, ranging from -19.3 ‰ to -13.6 ‰ (Fig. S6). $\delta^{13}$C source signatures for emissions from burning corn stalk were determined as -16.4 ± 1.4 ‰ (mean ± standard deviation): the mean (-16.4 ‰) was computed as the average of the different data sets, and standard deviation (1.4 ‰) analogously calculated.

In the revised manuscript, we change the title of Sect. 4.3.1 to "4.3.1 Selection of $\delta^{13}$C endmembers for aerosols from corn stalk burning in the study area", to clarify that the $\delta^{13}$C=-16.4 ± 1.4 ‰ is specific for burning corn stalk, which is a subtype of C4 plant. Further, in the notation of Table S1, we add the following as a reminder:

> "In this study, $\delta^{13}$C for corn stalk is used as it is the dominant C4 plant in Xi'an and its surrounding areas (Sun et al., 2017; Zhu et al., 2017), with little sugarcane and other C4 plants."

Number of digits are all corrected for $\delta^{13}$C values throughout the manuscript, according the reviewer's comments.

**References:**

Cao, J. J., Zhu, C. S., Chow, J. C., Watson, J. G., Han, Y. M., Wang, G.H., Shen, Z. X., and An, Z. S.: Black carbon relationships with emissions and meteorology in Xi'an, China, Atmos. Res., 94, 194–202, 2009.

de Rooij, M., van der Plicht, J., and Meijer, H.: Porous iron pellets for AMS $^{14}$C analysis of small samples down to ultra-microscale size (10–25μgC), Nucl. Instrum. Meth. B: Beam Interactions with Materials and Atoms, 268, 947-951, 2010.

Dusek, U., Ten Brink, H., Meijer, H., Kos, G., Mrozek, D., Röckmann, T., Holzinger, R., and Weijers, E.: The contribution of fossil sources to the organic aerosol in the Netherlands, Atmos. Environ., 74, 169-176, 2013.

Dusek, U., Monaco, M., Prokopiou, M., Gongriep, F., Hitzenberger, R., Meijer, H.A.J. and Röckmann, T.: Evaluation of a two-step thermal method for separating organic and elemental carbon for radiocarbon analysis, Atmos. Meas. Tech., 7(7), 1943–1955, https://doi.org/10.5194/amt-7-1943-2014, 2014.

Genberg, J., Hyder, M., Stenström, K., Bergström, R., Simpson, D., Fors, E., Jönsson, J. Å., and Swietlicki, E.: Source apportionment of carbonaceous aerosol in southern Sweden, Atmospheric Chemistry and Physics, 11, 11387-11400, 2011.

Gilardoni, S., Vignati, E., Cavalli, F., Putaud, J. P., Larsen, B. R., Karl, M., Stenström, K., Genberg, J., Henne, S., and Dentener, F.: Better constraints on sources of carbonaceous aerosols using a combined $^{14}$C – macro tracer analysis in a European rural background site, Atmos. Chem. Phys., 11, 5685-5700, https://doi.org/10.5194/acp-11-5685-2011, 2011.

Han, Y. M., Chen, L.W., Huang, R.J., Chow, J. C., Watson, J. G., Ni, H. Y., Liu, S. X., Fung, K.

K., Shen, Z. X, Wei, C., Wang, Q. Y., Tian, J., Zhao, Z. Z., Prévôt, A. S. H., and Cao, J. J.: Carbonaceous aerosols in megacity Xi'an, China: implications of thermal/optical protocols comparison, Atmos. Environ., 132, 58–68, 2016.

Minguillón, M. C., Perron, N., Querol, X., Szidat, S., Fahrni, S. M., Alastuey, A., Jimenez, J. L., Mohr, C., Ortega, A. M., Day, D. A., Lanz, V. A., Wacker, L., Reche, C., Cusack, M., Amato, F., Kiss, G., Hoffer, A., Decesari, S., Moretti, F., Hillamo, R., Teinilä, K., Seco, R., Peñuelas, J., Metzger, A., Schallhart, S., Müller, M., Hansel, A., Burkhart, J. F., Baltensperger, U., and Prévôt, A. S. H.: Fossil versus contemporary sources of fine elemental and organic carbonaceous particulate matter during the DAURE campaign in Northeast Spain, Atmos. Chem. Phys., 11, 12067-12084, https://doi.org/10.5194/acp-11-12067-2011, 2011.

Sun, J., Shen, Z., Cao, J., Zhang, L., Wu, T., Zhang, Q., Yin, X., Lei, Y., Huang, Y., Huang, R., Liu, S., Han, Y., Xu, H., Zheng, C., and Liu, P.: Particulate matters emitted from maize straw burning for winter heating in rural areas in Guanzhong Plain, China: current emission and future reduction, Atmos. Res., 184, 66–76, 2017.

Szidat, S., Jenk, T. M., Synal, H. A., Kalberer, M., Wacker, L., Hajdas, I., Kasper-Giebl, A., and Baltensperger, U.: Contributions of fossil fuel, biomass-burning, and biogenic emissions to carbonaceous aerosols in Zurich as traced by [14]C, J.Geophys. Res.-Atmos., 111, 2006.

Zhu, C. S., Cao, J. J., Tsai, C. J., Zhang, Z. S., and Tao, J.: Biomass burning tracers in rural and urban ultrafine particles in Xi'an, China, Atmos. Pollut. Res., 8, 614–618, http://dx.doi.org/10.1016/j.apr.2016.12.011, 2017.

---

## Author Comment (AC2) · 12 Sep 2018

**Response to reviewer 3**

**General comments:**

This manuscript reported datasets of carbon isotopes ($^{13}$C and $^{14}$C) of OC and EC in a major inland city of China, Xi'an, during one-year sampling, which were used to study the source apportionment of carbonaceous aerosols by combining $^{13}$C and $^{14}$C with Bayesian Markov chain Monte Carlo (MCMC) scheme. The data and methodology are reliable and novel. This paper shed some new light on the source apportionment of carbonaceous aerosols by distinguishing coal and liquid fossil fuel contributions to EC, C3 and C4 plant to biomass burning. The paper is relatively well-written, and it should be acceptable for publication after some moderate to major revision.

**Response:** We appreciate the constructive suggestion. We carefully considered all the comments of the reviewer in the revised manuscript.

**Major points: 1.** Be clear re mean or median values of source contribution, E.g., P1/L23, the 45% and 31% are median values in Figure 4. Need to be consistent in the manuscript.

**Response:** We agree with the reviewers that the mean or median values of source contribution is not clear enough to readers. Throughout the manuscript, the average with one standard deviation is used to represent the best estimate and uncertainties of radiocarbon results, respectively, as stated in Sect. 2.5: Source apportionment methodology using $^{14}$C (page 8 line 1–2). Thus, we used the average and one standard deviation for $f_{bb}$(EC), $f_{fossil}$(EC), $f_{nf}$(OC), $f_{fossil}$(OC), $EC_{bb}$, $EC_{fossil}$, $OC_{nf}$ and $OC_{fossil}$ throughout the manuscript.

For EC source apportionment results derived from the MCMC model (Sect 2.6), the median was used to represent the best estimate of the contribution of any particular source to EC. Uncertainties of this best estimate are expressed as inter-quartile range and 95 % range of corresponding PDF, as stated on page 14 line 19–20: "The median was used to represent the best estimate of the contribution of any particular source to EC. Uncertainties of this best estimate are expressed as inter-quartile range and 95 % range of corresponding PDF."

The revision in the abstract lines (page 1, line 22):

"relative contributions from coal combustion and liquid fossil fuel combustion are estimated as 45 % (median; 29–58 %, interquartile range) and 31 % (18–46 %) in winter"

The revised texts show that (page 14 line 21–23):

"For both MCMC4 and MCMC3, the MCMC-derived fraction of biomass burning EC ($f_{bb}$, median with interquartile range calculated by Eq. (7)) is similar to that obtained from radiocarbon data ($f_{bb}$(EC), average with one standard deviation by Eq. (3)) as both of them are well-constrained by F$^{14}$C. "

The revised caption of Fig. S9:

[Figure]

**Figure S9.** Comparison between the MCMC-derived fraction of biomass burning EC ($f_{bb}$ derived from MCMC4) and that obtained from radiocarbon data ($^{14}$C-based $f_{bb}$(EC)). Average and one standard deviation is shown for $f_{bb}$(EC), median with interquartile range is shown for $f_{bb}$ . $f_{bb}$ derived from MCMC3 is also very similar to $f_{bb}$(EC)."

Also, we checked the rest of the manuscript with care. When the MCMC results are discussed, the median with the interquartile range is clearly stated (e.g., page 15, line 4–5; page 18, line 25; Table 2; Fig. 5). We believe that after revision, the mean and median of the source apportion results is clear.

**2.** The flow of introduction is not well organized, and some part of the $^{14}$C introduction should be moved to the method part. I suggest re-organization and strengthening of the scientific objectives of this study in the introduction. In addition, you need to explain why we need to further distinguish the coal and fuel combustion in EC but not OC? Fossil sources contribute averagely 46% to OC based on your results, so it is important.

**Response:** We improved the introduction according to the comments and suggestions by the reviewer (page 2, line 2–23). The question regarding the $^{14}$C introduction is addressed in the response of question 8. The objectives of this study are re-organized in the introduction (page 4, line 14–20).

We did not further distinguish the coal and liquid fossil fuel combustion in fossil OC, because OC is chemically reactive, and $\delta^{13}$C of OC can be affected by atmospheric processing (Kirillova et al., 2013). The revised texts show (page 4, line 1–3):

"The interpretation of the stable carbon isotope signature for OC source apportionment is more difficult, because OC is chemically reactive and $\delta^{13}$C signatures of OC are not only determined by the source signatures but also influenced by atmospheric processing."

**3.** Provide more clear details of blank/contamination evaluation for $^{14}$C analysis, instrumental analytical precision and mention of source markers (S2) in the method part.

**Response: The** revised manuscript adds the following (page 7, line 5–9) in the method part:

"The differences between measured and nominal $F^{14}C$ values are used to correct the sample values (de Rooij et al., 2010; Dusek et al., 2014) for contamination during graphitization and AMS measurement (Supplemental S3). The modern carbon contamination is between 0.35 and 0.50 µg C, and the fossil carbon contamination is around 2 µg C for sample bigger than 100 µgC"

The detailed calculation of modern and fossil contamination is added in the supplemental material (Supplemental S3):

**"S3 Determination of modern and fossil contamination for radiocarbon measurement**

$F^{14}C$ of aerosols samples was corrected for contamination that occurred during graphitization and AMS measurement. For AMS measurements, samples are usually analyzed together with varying amounts of reference material covering the range of sample mass. Two such materials with known $^{14}C$ content are used: the oxalic acid OXII calibration material ($F^{14}C = 1.3406$) and a $^{14}C$-free $CO_2$ gas ($F^{14}C = 0$). Contamination during the graphitization and AMS measurement results into the differences between measured and nominal $F^{14}C$ values. The magnitude of these deviations can be used to quantify the contamination with fossil carbon ($F^{14}C_F = 0$) and modern carbon ($F^{14}C_M = 1$), which in turn are used for correcting the sample values (de Rooij et al., 2010).

The contamination with fossil carbon and modern carbon is quantified using isotope mass balance (Dusek et al., 2014):

$$F^{14}C_m \cdot M_m = F^{14}C_{st} \cdot M_{st} + F^{14}C_F \cdot M_F + F^{14}C_M \cdot M_M. \quad (S1)$$

$M_m$ and $M_{st}$ stand for the experimentally determined mass and the mass of reference materials either the oxalic acid OXII calibration material ($F^{14}C = 1.3406$) or a $^{14}C$-free $CO_2$ gas ($F^{14}C = 0$) with a unit of µgC, respectively. $F^{14}C_m$ and $F^{14}C_{st}$ represent the experimentally determined $F^{14}C$ measured by AMS and nominal $F^{14}C$ of reference materials (Table S9).

The relationships among all masses are described as Eq. (S2):

$$M_m = M_{st} + M_F + M_M, \quad (S2)$$

where $M_M$ is calculated using Eq. (S1) by substituting $F^{14}C_{st} = 0$ for a $^{14}C$-free $CO_2$ gas as:

$$M_M = F^{14}C_m \cdot M_m. \quad (S3)$$

Substitute $F^{14}C_{st} = 1.3406$ for OXII and the derived $M_M$ from Eq. (S3), $M_F$ is derived by combining Eq. (S1) and Eq. (S2) as:

$$M_F = ((1.3406 - F^{14}C_m) \cdot M_m - (1.3406 - 1) \cdot M_M)/1.3406. \quad (S4)$$

$M_M$ and $M_F$ is calculated by applying Eq. (S3) and Eq. (S4), and they are mass dependent. The modern carbon contamination ($M_M$) is between 0.35 and 0.50 μg C, and the fossil carbon contamination ($M_F$) is around 2 μg C for sample bigger than 100 μgC."

For OC and EC analysis the precision is determined as follows:

We repeat the analysis of samples, and the differences between the replicated analysis for the same sample (n=10) are smaller than 5% for TC, 5% for OC, and 10% for EC, respectively. This description is added in the Sect. 2.2 (page 5, line 15–16).

Following the reviewer's suggestion, we add briefly the measurement of source markers in the Sect. 2.2 as part of the methodology (page 5, line 17–19) and change the title to "2.2 Organic carbon (OC), elemental carbon (EC) and source markers measurement". S2 is kept in the supplemental material for details on the measurements.

**4.** The result section needs to be better structured and written. There are many parts, specially, sections 3.4 and 3.5, should be moved to discussion; and formulas could be moved to methods part.

**Response:** We have moved the sections 3.4 and 3.5 to the discussion following the reviewer's suggestion. In the revised manuscript, they are Sect. 4.3 and Sect. 4.4, respectively. The order of figures in the main text and supplemental material is adapted accordingly.

However, if the reviewer allows, we prefer to leave the formulas in the result and discussion section. In the Sect 4.3 (the original Sect. 3.4) and Sect 4.4 (the original Sect. 3.5), we detailed the way to model concentrations and sources of primary OC using those formulas and tried to explain the differences between the observed and modelled OC concentrations and sources. We hope that it would enhance the clarity and flow of this manuscript by leaving the equations in the part where they are need to know. This can also save the readers some time by avoiding going back to the method part, because those equations to model concentrations and sources of primary OC are not familiar to everyone. Further, there are equations used for illustrating the concept but not for calculation, for example:

$$\text{Observed OC conentrations} - OC_{pri,e} = OC_{o,nf} + SOC_{coal} + SOC_{liq.fossil}, \quad (12)$$

PS: we think the section of "**Sect. 3.4 $\delta^{13}$C/F$^{14}$C-based statistical source apportionment of EC**" and "**Sect. 3.5 Estimating mass concentrations and sources of primary OC**" (Sect. 4.3 and 4.4 of the revised manuscript) are results of this study. If we move these two sections to the discussion part, then the result section is very short and the discussion section is very long. If the reviewer agrees, we would like to move the two sections back the the result section.

**5.** There are many comparisons without in depth discussion in the discussion section. And comparison among different methods and different climate event seems not reasonable in 4.3.

**Response:** We did the whole year measurement in Xi'an for the year 2008/2009. In this study, we selected samples with varying PM$_{2.5}$ mass and carbonaceous aerosols for $^{14}$C analysis. The selected samples cover periods of low, medium and high PM$_{2.5}$ concentrations to get samples

representative of the various pollution conditions that did occur in each season (see details in Sect. 2.1 and Supplemental S1 for sample selection). Here we only compare the winter season due to the limited source apportionment results for EC in Xi'an. For Xi'an, we see from this study (Fig. 1(a), except the Chinese New Year eve, which is not included in the comparison) but also some studies in preparation that the $^{14}$C values do not change very much between polluted days and clean days. In addition, two months (the intensive campaign by Wang et al. (2016)) almost cover the whole winter (in total, 3 months). Thus, we think that it makes sense to compare the results from this study with the two cited paper.

Further, we address this question by answering the following questions 32 and 33 from the reviewer.

(a) **32.** P15/L16-20 I don't think it's reasonable to directly compare results of different methods, e.g., you got contribution of biomass burning to EC by MCMC4 with 4 sources while Zhang et al. (2015) got the fraction by 2 sources. Furthermore, taking into account of the error bar, the fraction fossil (76%)/ biomass (24%) of this study are the same to Zhang et al. (2015). Finally, Zhang et al. (2015) studied samples during the extreme winter haze episode of 2013.

**Response:** For the 4 sources solved by MCMC4, we did a posteriori combination of PDF of burning C3 and C3 plant, and named the combined PDF as biomass burning and denote the biomass burning contribution to EC as $f_{bb}$ (Sect. 4.3.2). As addressed in Question 1, we compared the MCMC-derived $f_{bb}$ with the $^{14}$C-derived $f_{bb}$(EC) i.e. two sources similar to Zhang et al. (2015), and found they were very similar to each other (Fig. S9), as both of them are well-constrained by $F^{14}C_{(EC)}$ (page 14, line 21–23 in the revised text):

> "For both MCMC4 and MCMC3, the MCMC-derived fraction of biomass burning EC ($f_{bb}$, median with interquartile range calculated by Eq. (7)) is similar to that obtained from radiocarbon data ($f_{bb}$(EC), median with one standard deviation by Eq. (3)) as both of them are well-constrained by $F^{14}C$ (Table 2, Table S5, Table S7, Fig. S9)."

Thus, we think it is reasonable to compare the $f_{bb}$ derived by MCMC4 with that from $^{14}$C-derived fraction of biomass burning and fossil fuel burning in EC by Zhang et al. (2015).

We agree with the reviewer that taking into account of the error bar, the fraction fossil (76%) and biomass burning (24%) in EC of this study are the same to Zhang et al. (2015). The revised text shows that (page 18, line 27–29):

> "Taken into account of the uncertainties, comparable contributions were also reported at the same sampling site for winter 2012/2013 based on $^{14}$C measurements (22 ± 3 %, Zhang et al., 2015a) and positive matrix factorization (PMF) receptor model simulation (20.1 ± 7.9 %, Wang et al., 2016) (Fig. 8)."

We noted that this study and Zhang et al. (2015) with different sampling focus. We conducted the campaign aiming at being representative for a year, and Zhang et al. (2015a) focus on the extreme winter haze. In this study, we selected samples with varying PM$_{2.5}$ mass and carbonaceous aerosols for $^{14}$C analysis. The selected samples cover periods of low, medium and high PM$_{2.5}$ concentrations to get samples representative of the various pollution conditions that did occur in each season. And we compare the sources of EC in winter from this study with Zhang et al.

(2015a). We think that this comparison is reasonable, because for Xi'an, we see from this study (see Fig. 1(a), except the Chinese New Year eve, which is not included in the comparison) but also some studies in preparation that the $^{14}$C values do not change very much between polluted days and clean days. We add the following clarification to the manuscript:

> "The contributions from biomass burning to EC was 24 % (median; interquartile range 22–26 %) in winter 2008/2009 (Fig. 8, Table 2) with no considerable change in $^{14}$C values between polluted days and clean days (Fig. 1(a), except the Chinese New Year eve)." (page 18, line 25–27)

(b) **33.** P15/L23-28 The same question as above. Because the PMF model didn't use $^{14}$C, is this reasonable for comparison?

**Response:** In our opinion it is reasonable to compare two different state of the art methods, provided that the uncertainty analysis for each is carefully done. If this was not possible, we would not be able, or justified to draw general conclusions from either method.

For EC source apportionment, it is noted that the quartile ranges for 2008/2009 values from MCMC4 (this study) overlaps ranges for 2012/2013 values (average ± SD) from the PMF model by Wang et al. (2016). Compared to the uncertainties of radiocarbon measurements, the uncertainties of PMF results are always larger, making the overlapped ranges very likely. But this will not change our conclusion that the source contributions of fossil EC are likely to have changed from 2008/2009 to 2013/2014, with decreasing contributions from coal burning and increasing contributions from motor vehicles. Figure S14 and Figure S15 shows the probability density functions (PDF) of the relative source contributions to EC from coal combustion and vehicle emissions, respectively. Results from this study for the year 2008/2009 are shown in grey (this is also shown in the original Fig. S8), and from Wang et al. (2016) for the year 2012/2014 shown in yellow. For the PDF by Wang et al. (2016), we assume normal distribution as their source apportionment results are not known and given in the form of average ± SD. As shown in Fig. S14 and Fig. S15, though with some overlaps, the PDF of the relative source contribution of coal combustion (vehicle emissions) does clearly shift to the lower side (higher side) from the year 2008/2009 to 2012/2013. Those discussions are added to the revised manuscript ((page 18 line 25–27; page 19, line 7–14).). Figure S14 and Figure S15 is added to the supplemental material.

We also have some additional observation data to support the conclusion as discussed in Sect. 4.6. The decreased contributions of coal combustion are also evidenced from the declined enrichment factor of As and Pb, indicators of coal combustion. Increasing vehicular emissions is supported by the increasing level of $NO_2$, an indicator for the contribution of vehicular emissions. (page 19, line 25–28; 29–31)

[Figure]

**Figure S14.** Probability density functions (PDF) of the relative source contributions of coal combustion to EC in winter in the year 2008/2009 (this study, shown in grey; this is also shown in Fig. S8) and 2012/2013 by Wang et al. (2016), shown in yellow.

[Figure]

**Figure S15.** Probability density functions (PDF) of the relative source contributions of vehicle emissions to EC in winter in the year 2008/2009 (this study, shown in grey; this is also shown in Fig. S8) and 2012/2013 by Wang et al. (2016), shown in yellow.

**6.** The Conclusion part was too long. I suggest summarizing the key points.

**Response:** thank you for this comment. The conclusion was shortened and the key points were summarized in the revision of the manuscript (page 20, page 21)

**Minor points and suggestions:**

**7. Introduction:** P2/L1-10 This part didn't emphasize the importance of carbonaceous aerosol very well. And the structure and description are very similar to the second paragraph of the introduction from Zhang et al., (2015a, Atmos. Chem. Phys). Need to revise.

**Response:** Thank you for spotting this. We re-write this part in the revised manuscript (page 2, line 2–23).

**8.** P2/L16: The definition and expression of fraction modern is not explicit. "The $^{14}C$ content" is not a ratio as "fraction modern". I don't think the standard need normalize for fractionation to $^{13}C$=-25‰. You can refer to Stuiver and Polach (1977) and modify this sentence. I also suggest move this "$^{14}C$ result report" part to the Methods part.

**Response:** Thank you for pointing this out. The revised manuscript shows that "The $^{14}C/^{12}C$ ratio of an aerosol sample is reported as fraction modern ($F^{14}C$)" (page 2, line 26).
Isotope fractionations occurs for $^{13}C$ and $^{14}C$ during the sample pre-treatment and measurements. To correct the isotope fractionations, samples are normalized to $\delta^{13}C$ = -25 ‰ with respect to VPDB, and the standards are normalized in the same way (Mook and van der Plicht, 1999; Reimer et al., 2004). $\delta^{13}C$ = -25 ‰ is chosen for normalization for isotope fractionations because it is a representative average of the majority of organic samples in nature. The only exception is that, due to the historical reasons, the old Oxalic Acid standard (OX1) is normalized to its own $\delta^{13}C$= -19‰.
We report the $^{14}C$ results of our aerosol samples as fraction modern ($F^{14}C$), following the nomenclature of Reimer et al. (2004). We read the Stuiver and Polach (1977) very carefully and find that $F^{14}C$ is referred to as $A_{SN}/A_{ON}$ in Stuiver and Polach (1977), where the subscript "$N$" denotes normalization for isotope fractionation for samples and standards as shown in Table 1 in Stuiver and Polach (1977). We also note that $A_{SN}/A_{ON}$ in Stuiver and Polach (1977) are normalized to $\delta^{13}C$= -19‰ of OX1. We think this citation is very helpful for readers to understand the $^{14}C$ data reporting, and we add the new citation in the revised text (page 2, line 28).

We moved the "$^{14}C$ result part" from the method part to the introduction part, following one reivewer's comment in the preliminary review. In the revised manuscript, we clarify the definition of $F^{14}C$ by explaining it in more detail (page 2 line 26–31; page 3 line 1–3). Further, we introduce the assumptions on $F^{14}C$ for contemporary sources (page 3, line 5–13) to facilitate the reader and also the writing of the following sections, following the comments by the reviewer 2. We think it probably better to leave the $F^{14}C$ definition and assumptions on $F^{14}C$ for contemporary sources in the introduction, if possible. We believe by doing this, the readers can better understand why $^{14}C$ is a very useful tool in the aerosol source apportionment and how.

**9.** P2/L25 Clarify which kind of $^{14}C$ studies only have two datasets: seasonal variations? TC or other? For example, Zhang et al. (2015b, EST) also reported annual and seasonal variations of EC in Beijing.
**Response:** This is an oversight. The revised manuscript adds the Y. Zhang et al.(2015b) and shows that:

"Despite a fair number of $^{14}$C studies in China in recent years, only a few $^{14}$C datasets so far reported annual results and seasonal variations of OC and EC (Y. Zhang et al., 2014a, 2015b, 2017)" (page 3 line 17–18)

**10.** P2/L27-30 It's better to introduce $^{13}$C first and introduce $^{14}$C as a novel tool. The $^{13}$C values of distinct sources you listed overlap with each other.

**Response:** We agree with the reviewer than the $^{14}$C is a novel tool and can constrain fossil and biomass burning contribution to EC very well. But in this study, we can only separate fossil EC into EC from coal combustion and EC from vehicular emission by complementing $^{14}$C with $^{13}$C. Using $^{14}$C alone can not separate the fossil EC into EC from coal combustion and liquid fossil fuel combustion.

In the introduction session, we introduce how the dual-carbon isotope-based (14C&13C) constrains on EC sources:

"EC from fossil sources (e.g., coal combustion, liquid fossil fuel burning) can be first separated from biomass burning by F$^{14}$C of EC. Further, $\delta^{13}$C of EC allows separation of fossil sources into coal and liquid fossil fuel burning (Andersson et al., 2015; Winiger et al., 2015, 2016), due to their different source signatures." (page 3 line 29–32)

The order of this description is the same of Eq. (7)–(9) for MCMC model in Sect. 2.6. Thus, we prefer to introduce F$^{14}$C first followed by $^{13}$C, if the reviewer allows.

It is noted that $^{13}$C source signatures are not that well distinct and they have some overlaps. If it is ok with the reviewer, we will put the deeper discussion into the methods section 2.6, where we introduce the use of $^{13}$C data for source apportionment and can further elaborate on the consequences of the overlapping signatures.

The source endmembers for $\delta^{13}$C are less well-constrained than for F$^{14}$C, as $\delta^{13}$C varies with fuel types and combustion conditions. For example, $\delta^{13}$C values for liquid fossil fuel combustion overlaps with $\delta^{13}$C values for both coal and C3 plant combustion. In this study, to account for the variability of the isotope signatures of $\delta^{13}$C and F$^{14}$C from the different sources, the Bayesian Markov chain Monte Carlo techniques (MCMC) were used as explained in the Method section. Uncertainties of both source endmembers for each source and the measured ambient $\delta^{13}C_{EC}$ and F$^{14}C_{(EC)}$ are propagated. These serve as input of MCMC to estimate the source contributions to EC. The MCMC results are the posterior probability density functions (PDF) for the relative contributions from the sources (Fig. S7, Fig. S8). The PDF of the relative source contributions of liquid fossil fuel combustion (vehicle) and coal combustion is skewed. By contrast, the PDF of the relative source contributions of biomass burning is symmetric as it is well-constrained by F$^{14}$C (Fig. S8(a)). In this study, the median with its interquartile range was used to give the estimates of the contribution of any particular source to EC throughout the manuscript (e.g., Table 2, Sect. 4.3.2).

This point is clarified in the Method Sect. 2.6 by adding the following (page 8 line 14-22):

"$\delta^{13}C_{bb}$, $\delta^{13}C_{liq.fossil}$ and $\delta^{13}C_{coal}$ are the $\delta^{13}C$ signature emitted from biomass burning, liquid fossil fuel combustion and coal combustion, respectively. The means and the standard deviations for $\delta^{13}C_{bb}$ (-26.7 ± 1.8 ‰ for C3 plants, and -16.4 ± 1.4 ‰ for corn stalk), $\delta^{13}C_{liq.fossil}$ (-25.5 ± 1.3 ‰) and $\delta^{13}C_{coal}$ (-23.4 ± 1.3 ‰) are presented in Table S1 (Andersson et al., 2015 and reference therein; Sect. 4.3.1), and serves as input of MCMC. The source endmembers for $\delta^{13}C$ are less well-constrained than for $F^{14}C$, as $\delta^{13}C$ varies with fuel types and combustion conditions. For example, the range of possible $\delta^{13}C$ values for liquid fossil fuel combustion overlaps to a small extent with the range for coal combustion, although liquid fossil fuels are usually more depleted than coal. The MCMC technique takes into account the variability in the source-signatures of $F^{14}C$ and $\delta^{13}C$ (Table S1), where $\delta^{13}C$ introduces a larger uncertainty than $F^{14}C$. Uncertainties of both source endmembers for each source and the measured ambient $\delta^{13}C_{EC}$ and $F^{14}C_{(EC)}$ are propagated."

**11.** P3/L7: same question as above.

**Response:** This is addressed as above. We also changed the sentence in the Introduction section "Further, $\delta^{13}C$ of EC allows separation of fossil sources into coal and liquid fossil fuel burning (Andersson et al., 2015; Winiger et al., 2015, 2016), as they have their distinct source signatures" to:

"Further, $\delta^{13}C$ of EC allows separation of fossil sources into coal and liquid fossil fuel burning (Andersson et al., 2015; Winiger et al., 2015, 2016), due to their different source signatures. Typical $\delta^{13}C$ values for EC from previous studies are summarized in Table S1." (page 3 line 30-32).

**12.** P3/L9-13 You need to provide [13]C numbers for example to make readers to remember the trend between different processes.

**Response:** How depleted the SOA is dependent on the extent of how much the precursors is reacted. As stated in the draft manuscript: "During formation of secondary organic aerosol (SOA), molecules depleted in heavy isotopes are expected to react faster, leading to SOA depleted in [13]C compared to its gaseous precursors, if the precursor is only partially reacted (Anderson et al., 2004; Irei et al., 2006; Fisseha et al., 2009)."
To the best of our knowledge, the $\delta^{13}C$ values for the SOA from different processes have not been well studied. So, we can not really give the exact [13]C numbers for the different processes. But we agree with the reviewer that some examples of $\delta^{13}C$ values for the SOA would be useful to readers. We add:

"For example, Irei et al. (2006) found that the $\delta^{13}C$ values of particulate SOA formed by OH radical-induced reactions of toluene ranged from -32.2‰ to -32.9 ‰, on average 5.8‰ lighter than those of parent toluene, when 7–29% toluene was reacted." (page 4 line 5–7)

"For example, Bosch et al. (2014) observed the more enriched $\delta^{13}C$ signature of water-soluble OC (-20.8 ± 0.7 ‰) than EC (-25.8 ± 0.3 ‰) at a receptor station for the South

Asian outflow, due to aging of OC during the long-range transport of aerosols." (page 4 line 9–11)

**Methods:**

**13.** P4/L1 replace "pre-fired" with "pre-baked"

**Response:** Done (page 4 line 30)

**14.** P4/L3 what's the standard for season's classification? Reference? How can you classify autumn to middle day?

**Response:** We classify the seasons based on the meteorological characteristics and the residential heating period in Xi'an, China. The official residential heating period starts from 15 November to 14 March next year and can be slightly changed with the temperature. The autumn is classified to the middle day of November as it was the last day before the winter heating.

The classification of seasons in this study is also consistent with earlier studies in Xi'an, China (Han et al., 2016; T. Zhang et al., 2014). The citations are added in the revised manuscript (page 5 line 2–3).

**15.** P4/L15 instrumental analytical precision? Do you have field blank?

**Response:** The question on the instrumental analytical precision is addressed in the response of Question 3.

Unfortunately, no field blank was collected during this sampling campaign. From more recent sampling campaigns we know that the typical TC on the field blank is usually smaller than 1μg cm$^{-2}$ with little EC, which is often below the detection level. This is much smaller than the TC loading on the collected samples ranging 20 μg cm$^{-2}$ to 300 μg cm$^{-2}$ with an average of 113 μg cm$^{-2}$. We can not conduct the blank correction, but this will only change the concentrations of OC slightly and even less for EC.

16. P5/L9-13 and L21-25 The whole blank/contamination should include all blank produced during experimental process and it is very important to know if the contamination is modern or fossil for $^{14}$C analysis. For combustion process, are the stds modern and fossil? Give the mass and F$^{14}$C value of the stds. Also, provide the F$^{14}$C value of combustion contamination. For $^{14}$C analysis, give the mass and F$^{14}$C value of contamination you got in this study. Did you have F$^{14}$C value of blank filter? Provide which blank you used to correct your $^{14}$C data.

**Response:** Thank you for this comment.

(a). the contamination introduced by the combustion process

For combustion processes, two sets of standard material: the oxalic acid HOxII and anthracite with known $^{14}$C contents (F$^{14}$C = 1.3406 and F$^{14}$C = 0, respectively) were combusted using ACS and used for quality control. The standard materials were put on the filter holder directly before heating in the oven in O$_2$ at 650°C for 10 minutes.
The F$^{14}$C deviations between the nominal values and measured values are attributed to the contamination introduced by the combustion process. The deviations are used to estimate the

contamination by applying Eq. (S1)–(S4) where the actual contamination can be divided into two components: fossil contamination ($M_F$ in µg C) with $F^{14}C_F = 0$ and modern contamination ($M_M$ in µg C) with $F^{14}C_M = 1$ (Dusek et al. 2014). The measured $F^{14}C$ and mass of the standards for quality control of the combustion process is shown in Table S9 in the revised manuscript. Measurements of the anthracite standard ($F^{14}C=0$) yield  modern contamination $M_M$ of 0.2–1.2 µg C per extraction. Measurements of OX II standard yield fossil contamination $M_F$ of 0.62–1.47 µg C per extraction.

We can conclude that the ACS system in fact introduced some contamination to the extracted samples but the contamination is not very large: (1) the measured $F^{14}C$ of the standards are deviating the nominal value only by less than 0.02 (the absolute values); (2). compared with our sampling size of 117 µg C (57–157 µg C; mean (min-max)) OC per extraction, 131 µg C (58–267 µg C) EC per extraction, the modern and fossil contamination is relatively small.

In this study, we did not correct the contamination introduced during the combustion process, because the contamination introduced from the combustion processes is small compared with our sample size.  To determine the actual contamination introduced during the combustion process, a series of blanks and standards made by ACS system covering the mass range of our sample size must be used for the mass-dependent correction.

 (b) contamination introduced by graphitization and AMS measurements

The method to estimate contamination is detailed in Supplemental S3 in the revised manuscript. The measure $F^{14}C$ and mass of standards are given in Table S9 in the revised manuscript. In the main text, we add:

> "The differences between measured and nominal $F^{14}C$ values are used to correct the sample values (de Rooij et al., 2010; Dusek et al., 2014) for contamination during graphitization and AMS measurement (Supplemental S3). The modern carbon contamination is between 0.35 and 0.50 µg C, and the fossil carbon contamination is around 2 µg C for sample bigger than 100 µgC."

Table S9. The measured $F^{14}C$ values and masses of the standards with their nominal $F^{14}C$ values.

| Standards | | nominal $F^{14}C$ | measured $F^{14}C$ ($F^{14}C_m$) | measured mass ($M_m$, μgC) |
|---|---|---|---|---|
| Combustion processes[a] | OXII | 1.3406 | 1.327 ± 0.022 | 65 |
| | OXII | 1.3406 | 1.321 ± 0.012 | 117 |
| | anthracite | 0 | 0.020 ± 0.001 | 51 |
| | anthracite | 0 | 0.002 ± 0.001 | 75 |
| | anthracite | 0 | 0.004 ± 0.001 | 219 |
| | anthracite | 0 | 0.005 ± 0.001 | 254 |
| Graphitization and $^{14}C$ measurements[b] | $^{14}C$-free $CO_2$ gas | 0 | 0.008 ± 0.001 | 42 |
| | $^{14}C$-free $CO_2$ gas | 0 | 0.004 ± 0.000 | 81 |
| | $^{14}C$-free $CO_2$ gas | 0 | 0.005 ± 0.000 | 91 |
| | $^{14}C$-free $CO_2$ gas | 0 | 0.004 ± 0.000 | 123 |
| | $^{14}C$-free $CO_2$ gas | 0 | 0.003 ± 0.000 | 162 |
| | $^{14}C$-free $CO_2$ gas | 0 | 0.002 ± 0.000 | 186 |
| | $^{14}C$-free $CO_2$ gas | 0 | 0.003 ± 0.000 | 287 |
| | OXII | 1.3406 | 1.268 ± 0.013 | 45 |
| | OXII | 1.3406 | 1.270 ± 0.012 | 81 |
| | OXII | 1.3406 | 1.280 ± 0.011 | 96 |
| | OXII | 1.3406 | 1.305 ± 0.010 | 128 |
| | OXII | 1.3406 | 1.337 ± 0.010 | 162 |
| | OXII | 1.3406 | 1.306 ± 0.006 | 214 |
| | OXII | 1.3406 | 1.311 ± 0.005 | 321 |

[a] For combustion processes, two sets of standard material: the oxalic acid HOxII and anthracite with known $^{14}C$ contents ($F^{14}C$ = 1.3406 and $F^{14}C$ = 0, respectively) were combusted using ACS and used for quality control;
[b] Varying amounts of reference materials covering the range of sample mass are graphitized and analyzed together with samples in the same wheel of AMS, to correct contamination during graphitization and AMS measurement.

(c) blank filter

As addressed in the question 15, unfortunately, no field blank was collected during this sampling campaign. Typical TC on the field blank is usually smaller than 1μg cm$^{-2}$ with little EC. This is much smaller than the TC loading on the collected samples ranging 20 μg cm$^{-2}$ to 300 μg cm$^{-2}$ with an average of 113 μg cm$^{-2}$. We can not conduct the blank correction, but this will only change the concentrations and $F^{14}C$ of OC slightly and even less for EC.

We measured the OC of one field blank filter for another sampling campaign in Xi'an, China. This OC field blank was analyzed by pooling several extracts of the same filter. The $F^{14}C$ of the OC on the field blank filter ($F^{14}C_{OC,blank}$) was 0.553 ± 0.003. The $F^{14}C_{OC,blank}$ is close to the ambient $F^{14}C$ of OC ($F^{14}C_{OC}$) values for this study, with an average of 0.567 ranging from 0.330

to 0.696. The blank correction will therefore not shift the ambient $F^{14}C_{OC}$ in this study strongly. It is relatively common in the literature that $^{14}C$ measurements of aerosol filter samples are not blank corrected, either because the blank values are too small to be measured for $^{14}C$, or because the correction would be very small compared to the relatively larger carbon amounts that need to be sampled for $^{14}C$ analysis (e.g., Szudat et al., 2004; 2006). A recent study of Dusek et al. (2017) found that the blank correction for $F^{14}C$ was very small and only slightly change $F^{14}C$ values of ambient samples.

In summary, both the measurements of the blank filter (from a different campaign, but conducted by the same group in the same location, with similar sampling setup) and the combustion standards show that the blank correction would shift the measured $^{14}C$ only by an insignificant amount, the correction was therefore omitted.

**17.** P6/L18 Is the "$f_{bb}$" the same as "$f_{bb}$" at L1?

**Response:** Thank you for this comment. Eq. (7) at L18 can be formulated as: $F^{14}C_{(EC)}=F^{14}C_{bb} \times f_{bb}$, because $F^{14}C_{liq.fossil}$ and $F^{14}C_{coal}$ is zero due to the long-time decay. The "$f_{bb}$" in Eq. (7) at L18 is by definition the same as $^{14}C$-based "$f_{bb}(EC)$" in Eq. (3) at L1, as both of them are calculated by dividing $F^{14}C_{(EC)}$ with $F^{14}C$ of biomass burning ($F^{14}C_{bb}$). However, we use different symbols, because in the first case, $f_{bb}(EC)$ is estimated by a Monte Carlo simulation, based on $^{14}C$ measurements alone, and in the second case $f_{bb}$ is estimated by the full 4 source MCMC model.

The main difference is that the we use average to represent the best estimate for $^{14}C$-derived "$f_{bb}(EC)$" in Eq. (3), and use median to represent the best estimate for $f_{bb}$ ( Eq. (7) at L18) derived from the MCMC. The median is used to represent the best estimate of the contribution of any particular source to EC derived from the MCMC, because the probability density functions (PDF) of the relative source contributions of liquid fossil fuel combustion (vehicle) and coal combustion is skewed. By contrast, the PDF of the relative source contributions of biomass burning is symmetric as it is well-constrained by $F^{14}C$ (Fig. S8(a)). Thus the MCMC-derived $f_{bb}$ (median) is very similar to that $f_{bb}(EC)$ (median), as shown in Fig. S9. This is clarified in the manuscript and the revised text shows that:

> "For both MCMC4 and MCMC3, the MCMC-derived fraction of biomass burning EC ($f_{bb}$, median with interquartile range calculated by Eq. (7)) is similar to that obtained from radiocarbon data ($f_{bb}(EC)$, median with one standard deviation by Eq. (3)) as both of them are well-constrained by $F^{14}C$ (Table 2, Table S5, Table S6, Fig. S9)." (page 14 line 21–23)

In addition, the caption of Figure S9 is revised to clarify the similarity between $f_{bb}$ and $f_{bb}(EC)$:

> "Figure S9. Comparison between the MCMC-derived fraction of biomass burning EC ($f_{bb}$ derived from MCMC4) and that obtained from radiocarbon data ($^{14}C$-based $f_{bb}(EC)$). Average and one standard deviation is shown for $f_{bb}(EC)$, median with interquartile range is shown for $f_{bb}$ . $f_{bb}$ derived from MCMC3 is also very similar to $f_{bb}(EC)$."

**Results:**

**18.** P7/L10 The number of seasonal samples in Table S2 is not the same as in the sampling part, need to clarify this inconsistency.

**Response:** Thank you for pointing this out. This is a mistake in the number of the samples. The sampling campaign in Xi'an was conducted for 13 months from 5 July 2008 to 8 August 2009 following every-sixth-day sampling schedule. In total, 65 $PM_{2.5}$ samples were collected during the 13 months as shown in the Table S2 in the draft manuscript.

However, this study focuses on the yearly cycle, and it will be confusing that there are two "July" and two "August" in a year cycle if we use the 65 samples collected the 13 months. Thus, at the end, we decide to use the 58 $PM_{2.5}$ samples collected from 5 July 2008 to 27 June 2009 to represent the one-year cycle.

The revised Table S1 shows there are 58 $PM_{2.5}$ samples in total, with 13 in spring, 15 in summer, 12 in autumn, and 18 in winter. We also check the mass concentrations in Table S2, and the mass concentrations of $PM_{2.5}$, OC and EC are the average and standard deviation calculated from the 58 samples collected from 5 July 2008 to 27 June 2009 with the season classification rules stated in the method Sect. 2.1.

**19.** P8/L20 unify the decimal place to one in the whole manuscript.

**Response:** Thank you for this comment. The decimal place is unified to one for $\delta^{13}C$ values throughout the manuscript.

**20.** P8/L30 Is it possible that combustion of a mixture of C4 and C3 plants or liquid fuel will results in the $^{13}C_{EC}$ values of around -24‰.

**Response:** To investigate if the mixture of C4 and C3 plants or liquid fossil fuel combustion can result in the $\delta^{13}C_{EC}$ values in winter (around -24‰), we run an additional adjusted simulation. In this simulation we assume that coal combustion does not contribute to EC at all in winter, even though coal combustion for heating is known as an important contributor to EC in winter (contributing around 45% to EC, see Sect. 4.3 and Fig. 5). That is, liquid fossil fuel combustion is the only fossil source of EC:

$$f_{\text{fossil}}(\text{EC}) = f_{\text{liq.fossil}} \qquad (R1)$$

where $f_{\text{fossil}}(\text{EC})$ is the relative contribution of fossil fuel to EC. Biomass burning contribution to EC ($f_{\text{bb}}(\text{EC})$) includes contribution from burning C3 ($f_{\text{bb\_C3}}$) and C4 plant ($f_{\text{bb\_C4}}$):

$$f_{\text{bb}}(\text{EC}) = f_{\text{bb\_C3}} + f_{\text{bb\_C4}} \qquad (R2)$$

Further, from the results of MCMC4, we know that C4 plant burning contribute more than that of C3 plant to EC in winter (Fig. S10b; Table S8), that is,

$$f_{\text{bb\_C4}} > f_{\text{bb\_C3}} \qquad (R3)$$

Based on the $\delta^{13}C$ source signature for EC from burning C4 plant ($\delta^{13}C_{\text{bb\_C4}}$ = -16.4 ± 1.4 ‰; mean ± standard deviation), C3 plant ($\delta^{13}C_{\text{bb\_C3}}$ =-26.7 ± 1.8 ‰) and liquid fossil fuel ($\delta^{13}C_{\text{liq.fossil}}$ = -25.5 ± 1.3 ‰), the Eq.(R1)–(R2) and the constraints of Eq.(R3), $\delta^{13}C$ of EC due to burning C4

plant, C3 plant and liquid fossil fuel can be estimated ($\delta^{13}C_{EC,e}$; the subscript "$_e$" denotes "estimated") by Eq.(R4):

$$\delta^{13}C_{EC,e} = \delta^{13}C_{bb\_C3} \times f_{bb\_C3} + \delta^{13}C_{bb\_C4} \times f_{bb\_C4} + \delta^{13}C_{liq.fossil} \times f_{liq.fossil} \qquad (R4)$$

A Monte Carlo simulation with 10,000 individual calculations was conducted to estimate propagated uncertainties. For each individual calculation, $\delta^{13}C_{bb\_C4}$, $\delta^{13}C_{bb\_C3}$, $\delta^{13}C_{liq.fossil}$, $f_{liq.fossil}(=f_{fossil}(EC))$ and $f_{bb}(EC)$ are randomly chosen from a normal distribution symmetric around the mean with the standard deviation. Mean and standard deviation of $f_{liq.fossil}(=f_{fossil}(EC))$ and $f_{bb}(EC)$ are derived from the radiocarbon measurements and $F^{14}C$ of biomass burning, and are given in Table S4. In this way 10,000 different estimation of $\delta^{13}C_{EC,e}$ can be calculated. The median was used to represent the best estimate of the contribution of any particular source to EC. Uncertainties of this best estimate are expressed as inter-quartile range ($25^{th}$-$75^{th}$) of the 10,000 estimated $\delta^{13}C_{EC,e.}$

Figure R1(A) shows the observed $\delta^{13}C_{EC}$ and the estimated $\delta^{13}C_{EC,e}$ values in winter. The observed $\delta^{13}C_{EC}$ in winter is shown as the filled black squares. The modelled $\delta^{13}C_{EC,e}$ is shown as the median (blue squares) with interquartile range ($25^{th}$-$75^{th}$ percentile, blue horizontal bars). Five from six observed $\delta^{13}C_{EC}$ values in winter are either outside the interquartile range of the estimated $\delta^{13}C_{EC,e}$ or on the higher/lower end of the range. This makes the assumption of no coal combustion contribution to EC very unlikely, at least for the 3 sample with high $f_{fossil}(EC)$.

Further, the conclusion will not change if no constraint of $f_{bb\_C4} > f_{bb\_C3}$ applys (Fig. R1(B)).

[Figure]

**Figure R1.** Two-dimensional isotope-based source characterization plot of EC in winter, with the assumption that no coal burning in winter contributed to EC. The observed $\delta^{13}C_{EC}$ in winter is shown as the filled black squares. The estimated $\delta^{13}C_{EC,e}$ with (A) and without (B) the constraint of $f_{bb\_C4} > f_{bb\_C3}$ is shown as the median (blue squares) with interquartile range (25th-75th percentile, blue horizontal bars). The fraction fossil in EC ($f_{fossil}$(EC) was calculated using radiocarbon data. The expected $\delta^{13}C$ and $^{14}C$ endmember ranges for C3 plant burning, and liquid fossil fuel combustion are shown as green and black bars, respectively, within the $^{14}C$-based endmember ranges for non-fossil (dark green rectangle, bottom) and fossil fuel combustion (grey rectangle, top). The $\delta^{13}C$ signatures of C4 plants burning is -16.4 ± 1.4 ‰ (mean ± standard deviation) is indicated in the plot but not shown on x-axis. The $\delta^{13}C$ signatures of C3 plants (green rectangle) and liquid fossil (e.g., oil, diesel, and gasoline, black rectangle) are indicated as mean ± standard deviation in Table S1.

**21.** P9/L26 what do the grey and dark green rectangle mean in Figure 3?

**Response:** we add the description of the grey and dark green rectangle in the caption of the Fig. 4 (the Fig. 3 in the draft manuscript):

> "The expected $\delta^{13}C$ and $^{14}C$ endmember ranges for biomass burning emissions, liquid fossil fuel combustion, and coal combustion are shown as green, black and brown bars, respectively, within the $^{14}C$-based endmember ranges for non-fossil (dark green rectangle, bottom) and fossil fuel combustion (grey rectangle, top)."

**22.** P10/L1 Sections 3.4.1 is not real results of this study

**Response:** In the revised manuscript, we move the Sect. 3.4 to the discussion section following the reviewer's suggestion. The original Sect. 3.4.1 is Sect. 4.3.1 in the revised manuscript. We think the Sect. 4.3.1 is not a problem now in the discussion section.

**23.** P10/L10 Provide the four-source calculation formula in 3.4.2 section and change the name MCMC3" in 2.6 section.

**Response:** Sect 2.6 provides the principle of the MCMC, where $f_{bb}$ in Eq. (7)–(9) represents the relative contributions of biomass burning to EC. $f_{bb}$ represents the relative contributions of C3 plant burning for MCMC3 and represented the sum contribution of C3 and C4 plant burning to EC for MCMC4, respectively.

This is explained as follows:

> "Results from a four-source (C3 biomass, C4 biomass, coal and liquid fossil fuel) MCMC4 model and a three-source (C3 biomass, coal and liquid fossil fuel) MCMC3 model were compared to underscore the influence of C4 biomass on source apportionment. The results of the Bayesian calculations are the posterior probability density functions (PDF) for the relative contributions from the sources (Fig. S7, Fig. S8). For MCMC4, we did a posteriori combination of PDF for C3 biomass and C4 biomass, and named the combined PDF as biomass burning, to better compare results with MCMC3. " (page 14 line 11–16 in the revised manuscript).

In the main text, we would prefer not to separate the "biomass" into "biomass from C3 plants" and "biomass from C4 plants" to avoid distracting the readers, if possible. Because we tried to say that including C4 plants in calculation (MCMC4) does not affect the contribution of biomass burning to EC but affect the separation between contributions from coal burning and liquid fossil fuel combustion. Further, the relative fraction of C3 and C4 plants in biomass burning is not the main purpose of this study. If we separate $f_{bb}$ into $f_{bb}$_C3 (contribution of C3 plant burning to EC) and $f_{bb}$_C4 (contribution of C4 plant burning to EC) in the formulas for MCMC4, then there will be two more symbols but they are not used in the manuscript later on.

 In addition, the MCMC3 does not include C4 plant burning, thus does not represent the real-world conditions in Xi'an, China and leads to biased EC source apportionment as discussed on page 14 line 25–32, page 15 line 1–3. So, we would prefer to leave the title of Sect. 2.6 and not to change the title of Sect. 2.6 to "MCMC3".

**24.** P10/L28 "5 times less than in summer" should use "lower than"

**Response:** Corrected (page 14 line 29).

**25.** P10/L29-32 The proportion of liquid fossil fuel combustion in winter (more coal burning) lower than summer (more traffic emission) make sense, why is not your expectation?

**Response:** Yes, the proportions are still according to expectations, but not the absolute values of EC. We expect that the absolute EC concentrations ($\mu g\ m^{-3}$) from liquid fossil fuel combustion ($EC_{liq.fossil}$) should be roughly constant all over the year, or even higher in winter due to unfavorable meteorological conditions.

The total EC concentrations in winter were only 1.5 times higher than that in summer. To meet our expectation, that is, $EC_{liq.fossil}$ in winter should be roughly equal to or a bit higher than that in summer, the relative contributions of liquid fossil fuel combustion to EC ($f_{liq.fossil}$) in winter should be 1.5 times (or less) lower than that in summer. But the MCMC3-derived contributions of liquid fossil fuel combustion to EC was only 14 % in winter, 5 times lower than in summer. This implies the absolute EC concentrations ($\mu g\ m^{-3}$) from liquid fossil fuel combustion were much smaller in winter than in summer, which is inconsistent with our expectation that absolute EC concentrations from liquid fossil fuel combustion should be roughly constant all over the year, or even higher in winter due to unfavorable meteorological conditions (page 14 line 28–32; page 15 line 1).

**26.** P11/L16 mean or median?

**Response:** It is mean values. $EC_{bb}$ ($\mu g\ m^{-3}$) and $EC_{fossil}$ ($\mu g\ m^{-3}$) of individual samples are calculated using Eq. (3) and Eq. (4), respectively. As stated in the method Sect 2.5, the average of $EC_{bb}$ ($\mu g\ m^{-3}$) and $EC_{fossil}$ ($\mu g\ m^{-3}$) derived from 10,000 individual calculations of a Monte Carlo simulation is used to represent the best estimate.

The mean $EC_{bb}$ and $EC_{fossil}$ in the winter is calculated by averaging the $EC_{bb}$ and $EC_{fossil}$ of the selected winter samples for $^{14}C$ analysis, respectively. The mean $EC_{bb}$ and $EC_{fossil}$ in the warm period is also calculated like this.

**27.** P12/L14 Does the $OC_{o,nf}$ mean observed non-fossil OC?

**Response:** The $OC_{o,nf}$ denotes other non-fossil OC except primary OC from biomass burning. We make this definition of $OC_{o,nf}$ more clear in the revised text:

> "Observed OC mass concentrations that exceed $OC_{pri,e}$ can be explained by contribution from secondary OC from coal combustion ($SOC_{coal}$), and liquid fossil fuel usage ($SOC_{liq.fossil}$) and by other non-fossil OC ($OC_{o,nf}$). $\underline{OC_{o,nf}}$ includes secondary OC from biomass burning and biogenic sources ($\underline{SOC_{nf}}$; SOC non-fossil), and primary OC from vegetative detritus, bioaerosols, resuspended soil organic matter, or cooking." (page 16 line 12–**16**)

**28. Discussion** P13/L16 Add "Characteristics" in front of 4.1 title.

**Response:** Thank you for this comment. Done (page 11 line 24).

**29.** P13/L17 Clarify which $f_{fossil}$ (EC) you used in comparison, the MCMC4 or $F^{14}C$? Because others use the $F^{14}C$ deduced values.

**Response:** Thank you for this comment. The $f_{fossil}$(EC) used in comparison is calculated from $F^{14}C_{(EC)}$. To clarify, the revised text shows:

"The annual average $f_{fossil}$(EC) derived from $^{14}$C data in Xi'an is 83 %." (page 11 line 25)

In addition, we try to make this more clear by adding the following note to Table 1 (page 29):

"a$f_{fossil}$(OC) and $f_{fossil}$(EC) in this study is calculated from the $F^{14}$C data (see details in Sect. 2.5)."

**30.** P13/L25 should be 76% as shown in Figure 4.

**Response:** As addressed in Question 29, the $f_{fossil}$(EC) is calculated from $F^{14}C_{(EC)}$. The averaged $f_{fossil}$(EC) in winter is 76.6 % ± 5.4% as shown in Table S5, and round to 77% at P13/L25 in the draft manuscript.

Due to the asymmetrical PDFs (Fig. S8), the individual median contributions of coal and liquid fossil fuel combustion ($f_{coal}$ and $f_{liq.fossil}$) do not add up to the median of the combined PDF for fossil fuel burning (slight difference) and we would like to avoid a discussion on that in the main manuscript, if possible. The $f_{fossil}$(EC) (average) deduced from $^{14}$C data and from MCMC (the sum of $f_{liq.fossil}$ and $f_{coal}$, median) can be a slight difference from each other, but still very similar as the $f_{bb}$(EC) and $f_{fossil}$(EC) is well-constrained by $F^{14}C_{(EC)}$.

**31.** P15/L4-8 Discussion here is not very convincing. The contribution of biomass burning to EC is the lowest in summer, but the highest contribution of biomass burning to EC occurred in winter (most corn stalk burning in winter, Figures 4 and S5), why no significant correlation was found in winter?

**Response:** Yes, no significant correlation was found in winter between $F^{14}C_{(EC)}$ and $K^+$/EC ratios and also not for $F^{14}C_{(EC)}$ and levoglucosan/EC ratios (Fig. S5).

In winter, the biomass burned in the studied area are mixture of crop residues (e.g., wheat straw, corn stalk) and wood. The levoglucosan/$K^+$ ratio for corn stalk burning and wheat straw burning is 0.21 ± 0.08 and 0.1 ± 0.0, respectively, much lower than those for wood burning (24.0 ± 1.8). No significant correlations between $F^{14}C_{(EC)}$ and $K^+$/EC ratios or between $F^{14}C_{(EC)}$ and levoglucosan/EC ratios suggest a changing mixture of biomass subtype (e.g., C3 plant (wheat straw and wood), C4 plant (corn stalk)) with different levoglucosan/$K^+$ ratios. A good correlation can only be expected, if one main type of biomass is burned. The revised manuscript shows:

"No significant correlations of $F^{14}C_{(EC)}$ with $K^+$/EC or levoglucosan/EC were found in other seasons (Fig. S5), suggesting a changing mixture of biomass subtypes with different levoglucosan/$K^+$ ratios. In this case the same amount of modern carbon contribution in EC (i.e., same $F^{14}C_{(EC)}$) can be associated with very different $K^+$/EC and levoglucosan/EC ratios, depending on which type of biomass is dominating at a given time" (page 13 line 19–23)

The correlations between $F^{14}C_{(EC)}$ with $K^+$ /EC ratios and $F^{14}C_{(EC)}$ with levoglucosan/EC ratios are used to infer the reason for the variability of EC, rather the absolute EC. Because $F^{14}C_{(EC)}$, $K^+$ /EC

ratios and levoglucosan/EC ratios are all relative term. Discussion on P15/L4-8 in the draft manuscript was discussing the variability of EC in summer, not absolute EC concentrations.

**32.** P15/L16-20 I don't think it's reasonable to directly compare results of different methods, e.g., you got contribution of biomass burning to EC by MCMC4 with 4 sources while Zhang et al. (2015) got the fraction by 2 sources. Furthermore, taking into account of the error bar, the fraction fossil (76%)/ biomass (24%) of this study are the same to Zhang et al. (2015). Finally, Zhang et al. (2015) studied samples during the extreme winter haze episode of 2013.

**Response:** This is addressed in Question 5(a).

**33.** P15/L23-28 The same question as above. Because the PMF model didn't use $^{14}$C, is this reasonable for comparison?

**Response:** This is addressed in Question 5(b).

**34.** P16/L7 and Figure 7 clarify the relationship between vehicular emissions and liquid fossil fuel combustion somewhere before discussion.

**Response:** In this study, liquid fossil fuel combustion and vehicular emission is used interchangeably. Because the $\delta^{13}C_{EC}$ signature of liquid fossil fuel combustion was compiled from literature where EC emitted from vehicles were collected and $\delta^{13}C_{EC}$ was measured. We add the following clarification in the revised manuscript where the liquid fossil fuel combustion is mentioned for the first time in the result section:

> "Major EC sources in Xi'an include biomass burning, coal combustion, and liquid fossil fuel (e.g., diesel and gasoline) combustion (i.e., vehicular emissions) …" (page 10 line 15–16)

We repeat this clarification with notes in parentheses several times, for example:
> "This is also evident from our observation that $\delta^{13}C$ values of the ambient aerosol fall within the range of C3 plants, coal and liquid fossil fuel combustion (i.e., vehicular emissions) (Fig. 2)." (page 10 line 20–21)

**35.** P17/L3 Will the biogenic emission to OC result in lower $^{13}$C values than EC?

**Response:** Thank you for this comment. We agree with the reviewer that if biogenic emissions play an important role on OC, we can expect a bit different $\delta^{13}C_{OC}$ from $\delta^{13}C_{EC.}$ Typically, biogenic OC concentrations are estimated on the order of a few microgram per cubic meter (μg /m$^3$), which is small compared to the observed TC in this study (Fig. 1 and Table S2). So probably primary and secondary biomass OC is responsible for most of the modern OC. And secondary OC in general might be responsible for more depleted $\delta^{13}C_{OC}$ (e.g., Irei et al., 2006; Fisseha et al., 2009). However, results on $\delta^{13}C$ of biogenic OC are sparse in the literature, and are so far inconclusive weather $\delta^{13}C$ is enriched or depleted. From our data we cannot make a firm conclusion that biogenic OC (mainly secondary) is the main cause.

**36. Conclusions** condense and summarize the key points of this study in this part.

**Response:** This is addressed in the Question 5.

**37. References** Check carefully the papers of the same author, e.g., you have two Zhang et al. (2015a) and where is Zhang et al. (2014a)? I think in section 4.3, you refer to Zhang et al., (2015a, Atmos. Chem. Phys.,)

**Response:** We check the references very carefully. There is one "Zhang and Cao, 2015a" and one "Zhang and Cao, 2015b" cited. And We find there is one "Zhang et al., 2015a" and one "Zhang et al., 2015b)" as follows:

> "**Zhang, Y. L. and Cao, F.**: Fine particulate matter ($PM_{2.5}$) in China at a city level, Sci. Rep., 5, 14884, **2015a**. (page 28 line 3)
>
> **Zhang, Y. L. and Cao, F.**: Is it time to tackle $PM_{2.5}$ air pollutions in China from biomass-burning emissions?, Environ. Pollut. , 202, 217–219, **2015b**. (page 28line 4–5)
>
> **Zhang, Y. L.,** Huang, R. J., El Haddad, I., Ho, K. F., Cao, J. J., Han, Y., Zotter, P., Bozzetti, C., Daellenbach, K. R., Canonaco, F., Slowik, J. G., Salazar, G., Schwikowski, M., Schnelle-Kreis, J., Abbaszade, G., Zimmermann, R., Baltensperger, U., Prévôt, A. S. H., and Szidat, S.: Fossil vs. non-fossil sources of fine carbonaceous aerosols in four Chinese cities during the extreme winter haze episode of 2013, Atmos. Chem. Phys., 15, 1299–1312, https://doi.org/10.5194/acp-15-1299-2015, **2015a**. (page 28 line 12–16)
>
> **Zhang, Y. L.,** Schnelle-Kreis, J., Abbaszade, G., Zimmermann, R., Zotter, P., Shen, R. R., Schäefer, K., Shao, L., Prévôt, A , and Szidat, S.: Source apportionment of elemental carbon in Beijing, China: insights from radiocarbon and organic marker measurements, Environ. Sci. Technol., 49, 8408–8415, **2015b**." (page 28 line 17–19)"

Zhang et al. (2014a) is cited in the Sect. 2.1 and it is revised to "T. Zhang et al., 2014" to differentiate "Y. Zhang et al., 2014a" and "Y. Zhang et al., 2014b":

> "Details about the sampling site can be found elsewhere (Bandowe et al., 2014; T. Zhang et al., 2014)." (page 4 line 26)

And in the reference list:

> "**Zhang, T.,** Cao, J.-J., Chow, J. C., Shen, Z.-X., Ho, K.-F., Ho, S. S. H., Liu, S.-X., Han, Y.-M., Watson, J. G., Wang, G.-H., and Huang, R.-J.: Characterization and seasonal variations of levoglucosan in fine particulate matter in Xi'an, China, J. Air Waste Manage., 64, 1317–1327, 10.1080/10962247.2014.944959, 2014." (page 27 line 41; page 28 line 1–2)

In Sect. 4.6 (original Sect. 4.3), we refer to Zhang et al. (2015a, Atmos. Chem. Phys.)

Except the citations pointed out by the reviewer, we find out more mistakes, and correct them all thoughtout the manuscript:

> (1) Zhang et al., 2014b  (revised to "Y. Zhang et al., 2014a")
>
> (2) Zhang et al., 2014c  (revised to "Y. Zhang et al., 2014b")
>
> (3) Liu et al., 2014a (revised to "G. Liu et al., 2014")
>
> (4) Liu et al., 2014b (revised to "J. Liu et al., 2014")
>
> (5) Huang et al., 2014a (revised to "R. Huang et al., 2014")
>
> (6) Huang et al., 2014b (revised to "X. Huang et al., 2014")

**References:**

de Rooij, M., van der Plicht, J., and Meijer, H.: Porous iron pellets for AMS [14]C analysis of small samples down to ultra-microscale size (10–25μgC), Nucl. Instrum. Meth. B: Beam Interactions with Materials and Atoms, 268, 947-951, 2010.

Dusek, U., Monaco, M., Prokopiou, M., Gongriep, F., Hitzenberger, R., Meijer, H.A.J. and Röckmann, T.: Evaluation of a two-step thermal method for separating organic and elemental carbon for radiocarbon analysis, Atmos. Meas. Tech., 7(7), 1943–1955, https://doi.org/10.5194/amt-7-1943-2014, 2014.

Dusek, U., Hitzenberger, R., Kasper-Giebl, A., Kistler, M., Meijer, H. A., Szidat, S., Wacker, L., Holzinger, R., and Röckmann, T.: Sources and formation mechanisms of carbonaceous aerosol at a regional background site in the Netherlands: insights from a year-long radiocarbon study, Atmospheric Chemistry and Physics, 17, 3233-3251, 2017.

Fisseha, R., Saurer, M., Jäggi, M., Siegwolf, R. T. W., Dommen, J., Szidat, S., Samburova, V., and Baltensperger, U.: Determination of primary and secondary sources of organic acids and carbonaceous aerosols using stable carbon isotopes, Atmos. Environ., 43, 431–437, http://dx.doi.org/10.1016/j.atmosenv.2008.08.041, 2009.

Irei, S., Huang, L., Collin, F., Zhang, W., Hastie, D., and Rudolph, J.: Flow reactor studies of the stable carbon isotope composition of secondary particulate organic matter generated by OH-radical-induced reactions of toluene, Atmos. Environ., 40, 5858–5867, 2006.

Mook, W. G. and van der Plicht, J.: Reporting [14]C activities and concentrations, Radiocarbon, 41, 227–239, 1999.

Prokopiou, M.: Characterization of a thermal method for separating organic and elemental carbon from aerosol samples using 14-C analysis, MS thesis, University of Groningen, the Netherlands, 2010.

Reimer, P. J., Brown, T. A., and Reimer, R. W.: Discussion: reporting and calibration of post-bomb [14]C data, Radiocarbon, 46, 1299–1304, 2004.

Szidat, S., Jenk, T. M., Gäggeler, H. W., Synal, H. A., Hajdas, I., Bonani, G., and Saurer, M.: THEODORE, a two-step heating system for the EC/OC determination of radiocarbon (14C) in the environment, Nuclear Instruments and Methods in Physics Research Section B: Beam Interactions with Materials and Atoms, 223–224, 829-836, http://dx.doi.org/10.1016/j.nimb.2004.04.153, 2004.

Szidat, S., Jenk, T. M., Synal, H. A., Kalberer, M., Wacker, L., Hajdas, I., Kasper‐Giebl, A., and Baltensperger, U.: Contributions of fossil fuel, biomass‐burning, and biogenic emissions to carbonaceous aerosols in Zurich as traced by 14C, Journal of Geophysical Research: Atmospheres (1984–2012), 111, 2006.

---

## Author Response (AR2)

**Comments to the Author:**

The authors have reasonably addressed the comments of the two anonymous referees and they have modified their manuscript accordingly. However, the comments given below should be addressed and many alterations are needed for the Main text and Supplement before the manuscript can be published in ACP.

**Response:** Thank you for providing us the opportunity to revise and improve our manuscript. The comments on the main text and supplement are addressed accordingly. Detailed responses to the comments are provided in blue. Attached please also find the marked-up manuscript to track the changes in the revised manuscript.

**1. Comments on the main text and supplement:**

**Main text:**

Page 2, line 24: "Stuiver and Polach, 1977" is missing in the Reference list.

**Response:** This reference is added in the Reference list:

"Stuiver, M. and Polach, H. A.: Discussion: Reporting of $^{14}$C data, Radiocarbon, 19, 355–363, 1977."

Page 18, line 23: Abbreviations and acronyms, here "SD", should be defined (written full-out) when first used.

**Response:** "SD" is defined when first used:

"This suggests that from 2008 to 2013, biomass burning contributions to EC remained rather stable, although with a slight decrease from 24 % (22–26 %) to 20 % (standard deviation, SD = 7.9 %)."

Page 19, lines 16-19: The three sentences on enrichment factor (EF) are unclear. I presume that the EFs are crustal EFs, thus relative to some crustal composition, but what was this crustal composition and which crustal reference element was used to obtain the EFs? Both should be specified.

**Response:** Thank you for this comment. Enrichment factors (EFs) were calculated as the ratios of trace element concentrations to Fe, which are normalized to the same elemental ratios of a reference material such as the earth's upper continental crust:

$$EF = \frac{[X/Fe]_{sample}}{[X/Fe]_{crust}}$$

where $[X/Fe]_{sample}$ is the concentration ratio of an element of interest (X) to the reference element Fe in a sample, $[X/Fe]_{crust}$ represents the concentration ratio of X to the reference element Fe in earth crust.

The revised manuscript shows:

> "The decreased coal combustion emissions are also evidenced from the declined Fe-referenced enrichment factors (EFs, normalized to composition of earth crust) of As and Pb. As and Pb can indicate coal combustion, as Pb-containing gasoline has been forbidden since 2000 in Xi'an (Xu et al., 2012). Annual EFs of As and Pb dropped from 802 and 804 in 2008 to 465 and 490 in 2010, respectively (Xu et al., 2016)."

**Supplement**

Page S3, line 4: "De Rooi et al., 2010" is missing in the Reference list. Also, should it not be "De Rooij" instead of "De Rooi"?

**Response:** Thank you for spotting this. It should be "de Rooij et al., 2010", and we add it in the Reference list:

> "de Rooij, M., van der Plicht, J., and Meijer, H.: Porous iron pellets for AMS $^{14}$C analysis of small samples down to ultra-microscale size (10–25 µgC), Nucl. Instrum. Meth. B, 268, 947–951, 2010."

Page S3, line 6: "Dusek et al., 2014" is missing in the Reference list.

**Response:** It is added in the Reference list:

> "Dusek, U., Monaco, M., Prokopiou, M., Gongriep, F., Hitzenberger, R., Meijer, H.A.J. and Röckmann, T.: Evaluation of a two-step thermal method for separating organic and elemental carbon for radiocarbon analysis, Atmos. Meas. Tech., 7(7), 1943–1955, https://doi.org/10.5194/amt-7-1943-2014, 2014."

Page S14, caption of Figure S10(b): It looks like there is something wrong with the indication of the green colours in the caption; within the Figure S3 and S4 are dark green and light green, respectively. Please, check also whether everything on this is correct in the Main text, especially for Figure 4.

**Response:** The legend and color in Figure S10(b) are indicated correctly. But the caption of Figure S10 (b) was wrong, in the revised text, now it reads:

> "Bars filled with green colour indicate the relative contribution of biomass burning, including C4 plants (light green) and C3 plants (dark green)."

We also check the main text including Figure 4, and they are good.

Pages S18 and S19, captions of Figures S14 and S15: "Wang et al., 2016" is missing in the Reference list.

**Response:** "Wang et al., 2016" is added in the Reference list:

"Wang, Q., Huang, R. J., Zhao, Z., Cao, J., Ni, H., Tie, X., Zhao, S., Su, X., Han, Y., Shen, Z., Wang, Y., and Zhang, N.: Physicochemical characteristics of black carbon aerosol and its radiative impact in a polluted urban area of China, J.Geophys. Res. -Atmos., 121, 2016."

Page S33: The "Streets et al., 2003" reference is incomplete. There should be an article number and maybe also a doi number before the publication year.

**Response:** The article number and doi number are added for the "Streets et al., 2003" reference:

"Streets, D., Yarber, K., Woo, J. H., and Carmichael, G.: Biomass burning in Asia: Annual and seasonal estimates and atmospheric emissions, Global Biogeochem. Cy., 17, 1099, doi:10.1029/2003GB002040, 2003."

**2. Response:** Alterations and corrections are made following the co-editor's comments. Below are the alterations made in the main text and supplement.

**Main text**

Page 1, line 23: Replace "EC are from" by "EC is from".

Page 2, line 13: Replace "non-fossil sources for" by "non-fossil source for".

Page 2, line 14: Replace "other non-fossil sources" by "other sources" and replace "that exclusively" by "that is exclusively".

Page 5, line 7: Replace "replicated analysis" by "replicated analyses".

Page 5, line 9: Replace "Organic makers" by "Organic markers".

Page 7, line 19: Replace "and is 0" by "and 0".

Page 8, line 4: Replace "is zero" by "are zero".

Page 8, line 28: Replace "reported values" by "reported".

Page 9, line 29: Replace "ratios" by "ratio" and replace "concentrations from" by "concentration from".

Page 9, line 30: Replace "were" by "was".

Page 10, line 22: Replace "falls into" by "fall into".

Page 10, line 24: Replace "has some" by "there are some".

Page 11, line 29: Replace "are reported in" by "was reported for" and replace "2013), Xiamen" by "2013) and Xiamen".

Page 12, lines 20 and 21: Replace "for example" by "for example,".

Page 13, line 18: Replace "overlaps with" by "overlap with".

Page 13, line 30: Replace "signatures for" by "signature for".

Page 13, line 31: Replace "burning are" by "burning is".

Page 14, line 18: Replace "compared the" by "compared to the".

Page 14, line 20: Replace "contributions of" by "contribution of".

Page 15, line 4: Replace "are much" by "is much".

Page 16, line 9: Replace "conentrations" by "concentrations".

Page 16, line 20: Replace "owning to" by "owing to".

Page 16, line 21: Replace "e.g. median" by "e.g., median" and replace "range, (1.9-4)" by "range, (1.9-4))".

Page 16, line 24: Replace "are consistently" by "is consistently" and replace "correlations" by "correlation".

Page 16, line 25: Replace "were observed" by "was observed".

Page 17, line 11: Replace "investigating, whether" by "investigating whether" and replace "for example" by "for example,".

Page 17, line 13: Replace "For examples" by "For example".

Page 17, line 14: Replace "is 2 times" by "are 2 times".

Page 17, line 18: Replace "than that total" by "than total".

Page 17, line 19: Replace "of0.85" by "of 0.85".

Page 17, line 20: Abbreviations and acronyms, here "VOCs", should be defined (written full-out) when first used. Since "VOC" is not used elsewhere in the text, I suggest replacing "semi-VOCs" by "semi-volatile organic compounds".

Page 18, line 18: Replace "contributions from" by "contribution from".

Page 18, line 20: Replace "Taken into account of the" by "Taking into account the".

Page 18, lines 28 and 29: Replace "contributions" by "contribution".

Page 18, line 30: Replace "ranges for" by "range for" on two occasions.

Page 19, line 2: Replace "give a more" by "gives a more" and replace "Figure S14 and S15 shows" by "Figures S14 and S15 show".

Page 20, line 4: Replace "Chinses cities" by "Chinese cities".

Page 21, line 6: Replace "by the KNAW" by "by a KNAW".

Page 23, line 26: Insert a space between the semicolon and "Limits".

Page 29, line 1: Replace "in China" by "in China.".

Page 30, line 5: Replace "is shown" by "are shown".

Page 34, line 3: Replace "was calculated" by "were calculated".

Page 34, line 7: Replace "signatures of" by "signature of" and replace "is not shown on" by "and is not shown on the".

Page 36, line 5: Replace "are shown" by "is shown".

Page 36, line 6: Replace "denotes different" by "denote different".

Page 36, lines 7-8: Replace "shown but not" by "indicated but are too small to be".

**Supplement:**

Page S2, line 1 below heading "S2 Measurement ...": Replace "Organic makers" by "Organic markers".

Page S2, line 2 above heading "S3 Determination ...": Abbreviations and acronyms, here "MDLs", should be defined (written full-out) when first used. Since "MDL" is not used elsewhere in the Supplement, I suggest replacing "The MDLs" by "The minimum detection limit".

Page S3, line 3 above heading "S4 Primary ...": Replace "is calculated" by "are calculated".

Page S3, line 1 above heading "S4 Primary ...": Replace "for sample" by "for a sample".

Page S4, line 4: Replace "burning subtype" by "burning subtypes".

Page S4, line 9: Replace "(2015)" by "(2015))".

Page S4, line 10: Replace "(2016)" by "(2016))".

Page S4, line 14: Replace "partially" by "which partially".

Page S4, lines 20-21: Replace "respectively overlaps most of data in literatures" by "respectively, overlap with most of the data in the literature".

Page S4, line 21: Replace "ratios for coal" by "ratio for coal".

Page S6, caption of Figure S2: Replace "air backward-in-time air" by "backward-in-time air".

Page S16, caption of Figure S12, line 6: Replace "denotes different" by "denote different".

Page S16, caption of Figure S12, last line: Replace "shown but not" by "indicated but are too small to be".

Page S21, caption of Figure S17, line 3: Replace "represents" by "represent".

Page S22, line 1: Replace "studies" by "studies.".

Page S22, footnote of Table S1, line 1: Replace "signatures" by "signature".

Page S22, footnote of Table S1, line 2: Replace "are applied" by "is applied".

Page S23, footnote of Table S2: Replace "number in the parentheses is" by "numbers in parentheses are".

Page S24, footnotes of Table S3, line 1: Replace "uncertainties" by "uncertainty".

Page S29, line 1: Replace "The contribution" by "Contribution".

Page S30, line 1: Replace "The measured" by "Measured".

Page S30, footnotes of Table S9, line 2: Replace "ACS" by "the aerosol combustion system".

Page S30, footnotes of Table S9, last line: Replace "correct contamination" by "correct for contamination".

[revised manuscript text omitted]

F$^{14}$C of aerosols samples was corrected for contamination that occurred during graphitization and AMS measurement. For AMS measurements, samples are usually analysed together with varying amounts of reference material covering the range of sample mass. Two such materials with known $^{14}$C content are used: the oxalic acid OXII calibration material (F$^{14}$C = 1.3406) and a $^{14}$C-free CO$_2$ gas (F$^{14}$C = 0).

Contamination during the graphitization and AMS measurement results into the differences between measured and nominal $F^{14}C$ values. The magnitude of these deviations can be used to quantify the contamination with fossil carbon ($F^{14}C_F = 0$) and modern carbon ($F^{14}C_M = 1$), which in turn are used for correcting the sample values (de Rooij et al., 2010).

The contamination with fossil carbon and modern carbon is quantified using isotope mass balance (Dusek et al., 2014):

$$F^{14}C_m \cdot M_m = F^{14}C_{st} \cdot M_{st} + F^{14}C_F \cdot M_F + F^{14}C_M \cdot M_M. \qquad (S1)$$

$M_m$ and $M_{st}$ stand for the experimentally determined mass and the mass of reference materials either the oxalic acid OXII calibration material ($F^{14}C = 1.3406$) or a $^{14}C$-free $CO_2$ gas ($F^{14}C = 0$) with a unit of µgC, respectively. $F^{14}C_m$ and $F^{14}C_{st}$ represent the experimentally determined $F^{14}C$ measured by AMS and nominal $F^{14}C$ of reference materials (Table S9).

The relationships among all masses are described as Eq. (S2):

$$M_m = M_{st} + M_F + M_M, \qquad (S2)$$

where $M_M$ is calculated using Eq. (S1) by substituting $F^{14}C_{st} = 0$ for a $^{14}C$-free $CO_2$ gas as:

$$M_M = F^{14}C_m \cdot M_m. \qquad (S3)$$

Substitute $F^{14}C_{st} = 1.3406$ for OXII and the derived $M_M$ from Eq. (S3), $M_F$ is derived by combining Eq. (S1) and Eq. (S2) as:

$$M_F = ((1.3406 - F^{14}C_m) \cdot M_m - (1.3406 - 1) \cdot M_M)/1.3406. \qquad (S4)$$

$M_M$ and $M_F$ is are calculated by applying Eq. (S3) and Eq. (S4), and they are mass dependent. The modern carbon contamination ($M_M$) 
[revised manuscript text omitted]

[Figure]

**Figure S10.** Sources of EC in different seasons. Results from the F$^{14}$C and δ$^{13}$C based Bayesian source apportionment calculations of EC. The numbers in the bars represent the median contribution of liquid fossil fuel, coal and biomass burning. (a) results from the MCMC3 model, including C3 plants as biomass, coal and liquid fossil fuel; (b) Impact of C4 plants burning on EC source apportionment is tested by including C4 biomass into the calculations (MCMC4). For MCMC4, the PDF for C3 and C4 plants is combined and named as biomass burning. Bars filled with green colour indicate the relative contribution of biomass burning, including  C4 plants (light green) and  C3 plants (dark green). In winter, the sample taken on Chinese New Year eve (25 January 2009) was excluded.

[Figure]

**Figure S11.** MCMC4-derived source contributions to EC for each data point computed using the Bayesian Markov chain Monte Carlo approach. (a). biomass burning from C3 plants; (b). biomass burning from C4 plants; (c). liquid fossil fuel combustion; (d). coal combustion. Range of 95 % credible intervals (Bayesian analogue of confidence intervals) and interquartile range (25th-75th percentile) from the computed probability density functions (PDF) and shown in black and green error bars, respectively. To better compare results with MCMC3, we did a posteriori combination of PDF for C3 biomass (a) and C4 biomass (b) and named the combined PDF as biomass burning (e).

[Figure]

**Figure S12.** Estimated primary OC based on MCMC3 results. (a) measured OC concentrations (blue line and diamond symbols) with observational uncertainties (vertical bar) and estimated OC mass (OC$_{pri,e}$, circle and triangular symbols) from apportioned EC and OC/EC ratios for different sources (Eq. (10)). (b) $^{14}$C-based fraction of non-fossil OC ($f_{nf}(OC)$) and modelled non-fossil fraction in OC$_{pri,e}$ ($f_{bb}(OC_{pri,e})$) derived from Eq. (11). Interquartile range (25$^{th}$-75$^{th}$ percentile) of the median OC$_{pri,e}$ and $f_{bb}(OC_{pri,e})$ are shown in purple (A), red (B) and green (C) vertical bars. "A" and "B" denotes different OC/EC ratios applied to primary biomass burning emissions ($r_{bb}$): A. $r_{bb}$ = 5 (3–7, minimum-maximum), B. $r_{bb}$ = 4 (3–5). "C" denotes 80 % $r_{liq.fossil}$ applied in summer with $r_{bb}$ = 5. $f_{nf}(OC)$ uncertainties are  indicated but  visible.

[Figure]

**Figure S13.** Observed and estimated OC concentrations. Modelled $OC_{e,min}$ is the sum of $OC_{pri,e}$ and $OC_{o,nf}$. $OC_{o,nf}$ accounts for the differences between $f_{nf}(OC)$ and $f_{bb}(OC_{pri,e})$, with an unrealistic assumption of no secondary fossil OC, leading to minimum addition to $OC_{pri,e}$. Coral area shows the $POC_{bb,e}$ and $OC_{o,nf}$, green area the $POC_{coal,e}$ and blue area the $POC_{liq.fossil,e}$. Estimation is based on MCMC3 results for EC source apportionment and primary OC/EC ratios corresponding to case (A) in Fig. S12.

[Figure]

**Figure S14.** Probability density functions (PDF) of the relative source contributions of coal combustion to EC in winter in the year 2008/2009 (this study, shown in grey; this is also shown in Fig. S8) and 2012/2013 by Wang et al. (2016), shown in yellow.

**Figure S15.** Probability density functions (PDF) of the relative source contributions of vehicle emissions to EC in winter in the year 2008/2009 (this study, shown in grey; this is also shown in Fig. S8) and 2012/2013 by Wang et al. (2016), shown in yellow.

[Figure]

**Figure S16**. OC/EC ratios estimated from OC and EC emission amounts from biomass burning emission inventories specific to China. y-axis on the right side indicates the year of estimation. The range applied in OC estimation (Sect. 4.4 in main text) is shown by dashed vertical lines, and the mean is indicated by a full vertical line. Data sources: Streets et al. (2003); Yan et al. (2006); Zhang et al. (2006, 2009); Cao et al. (2006, 2011a), Qin and Xie (2011), Zhang et al. (2013), Zhou et al. (2017).

[Figure]

**Figure S17**. Literature reported OC/EC ratios for combustion of bituminous coal and anthracite coal. Boxplots show the median (thick line across the box), interquartile range (25th-75th percentile, vertical ends of the box). Outliers are shown as triangles. Blue dots (averages) with error bars (one standard deviation) represents OC/EC ratios measured by IMPROVE_A protocol reported by Tian et al. (2017). Data sources: Chen et al. (2005, 2006, 2015), Zhang et al., (2008, 2012), Zhi et al. (2008), Shen et al. (2010, 2015), Li et al. (2016b).

**Table S1.** Range of $\delta^{13}C$ values for each source reported in previous studies.

| Sources | $\delta^{13}C$ of emissions from sources (ranges) | Source signatures of $\delta^{13}C$ used in the source apportionment calculations of EC (mean ± standard deviation) | Reference |
|---|---|---|---|
| Corn stalk | -19.30 ‰ to -13.6 ‰[a] | -16.4 ± 1.4 ‰[a] | (Martinelli et al., 2002; Das et al., 2010; Chen et al., 2012; Kawashima and Haneishi, 2012; Liu et al., 2014; Guo et al., 2016) |
| C3 plants (wood, wheat straw, etc.) | -35 ‰ to -24 ‰ | -26.7 ± 1.8 ‰ | Andersson et al. (2015) and references therein |
| coal | -25 ‰ to -21 ‰ | -23.4 ± 1.3 ‰ | Andersson et al. (2015) and references therein |
| liquid fuel (e.g., gasoline, diesel, and oil) | -28 ‰ to -24 ‰ | -25.5 ± 1.3 ‰ | Andersson et al. (2015) and references therein |

[a] $\delta^{13}C$ source signatures for EC from burning corn stalk (C4 plant) of -16.4 ± 1.4 ‰ (mean ± standard deviation)  is applied in MCMC4 calculations.  In this study, $\delta^{13}C$ for corn stalk is used as it is the dominant C4 plant in Xi'an and its surrounding areas (Sun et al., 2017; Zhu et al., 2017), with little sugarcane and other C4 plants.  See details on selection of $\delta^{13}C$ signature for C4 plants in the study area (corn stalk) in Sect.4.3.1 and Fig. S6.

**Table S2.** Mass concentrations of PM$_{2.5}$, OC and EC in Xi'an, China from July 2008 to June 2009.

|  | PM$_{2.5}$ (µg m$^{-3}$) | OC (µg m$^{-3}$) | EC (µg m$^{-3}$) |
|---|---|---|---|
| Spring (n=13) | 124.0 ± 40.4 *(55.9–193.4)[a]* | 14.4 ± 9.6 *(3.3–33.8)* | 5.7 ± 2.3 *(2.0–8.8)* |
| Summer (n=15) | 83.0 ± 30.7 *(31.8–139.2)* | 12.7 ± 4.5 *(4.0–20.6)* | 6.3 ± 2.0 *(2.7–10.0)* |
| Autumn (n=12) | 125.1 ± 69.3 *(41.0–212.6)* | 22.2 ± 13.6 *(3.6–34.2)* | 8.4 ± 2.9 *(3.5–11.3)* |
| Winter (n=18) | 213.4 ± 91.8 *(73.1–408.5)* | 39.0 ±17.8 *(10.8–67.0)* | 9.1 ± 3.1 *(5.6–16.3)* |
| Annual | 142.0 ± 82.4 *(31.8–408.5)* | 21.5 ± 16.6 *(3.3–67.0)* | 7.6 ± 3.0 *(2.0–16.3)* |

[a] average ± standard deviation, the numbers in  parentheses  are the range of each dataset.

**Table S3.** Average fraction modern ($F^{14}C$) and stable carbon signature ($\delta^{13}C$, ‰) of OC and EC for selected samples.

| Date | $F^{14}C_{(OC)}$ | $F^{14}C_{(EC)}$ | $\delta^{13}C_{OC}$ | $\delta^{13}C_{EC}$ | Season |
|---|---|---|---|---|---|
| 7/17/2008[a] | $0.466 \pm 0.010$ | $0.178 \pm 0.003$ | -26.80 | -26.50 | summer |
| 7/23/2008 | $0.489 \pm 0.008$ | $0.164 \pm 0.003$ | -25.94 | -26.33 | summer |
| 8/4/2008 | $0.546 \pm 0.007$ | $0.153 \pm 0.002$ | -25.86 | -26.16 | summer |
| 8/11/2008 | $0.512 \pm 0.008$ | $0.141 \pm 0.003$ | -25.21 | -25.53 | summer |
| 9/3/2008 | $0.549 \pm 0.006$ | $0.129 \pm 0.002$ | -25.94 | -26.23 | autumn |
| 10/3/2008 | $0.581 \pm 0.006$ | $0.166 \pm 0.002$ | -24.55 | -25.51 | autumn |
| 10/16/2008 | $0.659 \pm 0.007$ | $0.188 \pm 0.002$ | -23.70 | -24.31 | autumn |
| 10/21/2008 | $0.610 \pm 0.005$ | $0.301 \pm 0.003$ | -24.51 | -24.92 | autumn |
| 11/2/2008 | $0.651 \pm 0.006$ | $0.172 \pm 0.002$ | -24.94 | -25.10 | autumn |
| 11/14/2008 | $0.579 \pm 0.007$ | $0.200 \pm 0.004$ | -25.48 | -24.79 | autumn |
| 11/26/2008 | $0.671 \pm 0.009$ | $0.245 \pm 0.004$ | -24.71 | -22.93 | winter |
| 12/20/2008 | $0.696 \pm 0.008$ | $0.225 \pm 0.002$ | -24.06 | -22.81 | winter |
| 1/1/2009 | $0.693 \pm 0.007$ | $0.317 \pm 0.004$ | -23.23 | -23.12 | winter |
| 1/25/2009 | $0.745 \pm 0.005$ | $0.505 \pm 0.008$ | -23.39 | -23.07 | winter |
| 2/6/2009 | $0.671 \pm 0.007$ | $0.318 \pm 0.005$ | -23.92 | -23.72 | winter |
| 3/5/2009 | $0.572 \pm 0.006$ | $0.183 \pm 0.003$ | -25.44 | -23.53 | winter |
| 3/17/2009 | $0.545 \pm 0.004$ | $0.177 \pm 0.002$ | -25.72 | -26.03 | spring |
| 3/29/2009 | $0.547 \pm 0.006$ | $0.153 \pm 0.002$ | -26.91 | -25.38 | spring |
| 4/16/2009 | $0.545 \pm 0.007$ | $0.166 \pm 0.003$ | -27.42 | -25.05 | spring |
| 4/22/2009 | $0.535 \pm 0.006$ | $0.175 \pm 0.004$ | -26.33 | -25.27 | spring |
| 4/28/2009 | $0.330 \pm 0.021$ | $0.175 \pm 0.005$ | -26.41 | -25.33 | spring |
| 5/4/2009 | $0.544 \pm 0.004$ | $0.180 \pm 0.003$ | -26.66 | -25.35 | spring |
| 6/9/2009 | $0.549 \pm 0.006$ | $0.132 \pm 0.003$ | -24.24 | -25.37 | summer |
| 6/21/2009 | $0.489 \pm 0.006$ | $0.124 \pm 0.002$ | -26.30 | -25.73 | summer |
| summer[b] | $0.509 \pm 0.033$ | $0.149 \pm 0.020$ | $-25.7 \pm 0.9$ | $-25.9 \pm 0.5$ | |
| autumn | $0.605 \pm 0.044$ | $0.193 \pm 0.058$ | $-24.9 \pm 0.8$ | $-25.1 \pm 0.7$ | |
| winter | $0.675 \pm 0.057$ | $0.299 \pm 0.114$ | $-24.1 \pm 0.8$ | $-23.2 \pm 0.4$ | |
| spring | $0.508 \pm 0.087$ | $0.171 \pm 0.010$ | $-26.6 \pm 0.6$ | $-25.4 \pm 0.4$ | |

[a] Daily $F^{14}C$ values are given in average ± measurement uncertainty;

[revised manuscript text omitted]

[c]Sample taken from Chinese New Year eve (25 January 2009) was excluded.

**Table S9.**  Measured F$^{14}$C values and masses of the standards with their nominal F$^{14}$C values.

| Standards | | nominal F$^{14}$C | measured F$^{14}$C (F$^{14}$C$_m$) | measured mass (M$_m$, µgC) |
|---|---|---|---|---|
| Combustion processes[a] | OXII | 1.3406 | 1.327 ± 0.022 | 65 |
| | OXII | 1.3406 | 1.321 ± 0.012 | 117 |
| | anthracite | 0 | 0.020 ± 0.001 | 51 |
| | anthracite | 0 | 0.002 ± 0.001 | 75 |
| | anthracite | 0 | 0.004 ± 0.001 | 219 |
| | anthracite | 0 | 0.005 ± 0.001 | 254 |
| | | | | |
| Graphitization and $^{14}$C measurements[b] | $^{14}$C-free CO$_2$ gas | 0 | 0.008 ± 0.001 | 42 |
| | $^{14}$C-free CO$_2$ gas | 0 | 0.004 ± 0.000 | 81 |
| | $^{14}$C-free CO$_2$ gas | 0 | 0.005 ± 0.000 | 91 |
| | $^{14}$C-free CO$_2$ gas | 0 | 0.004 ± 0.000 | 123 |
| | $^{14}$C-free CO$_2$ gas | 0 | 0.003 ± 0.000 | 162 |
| | $^{14}$C-free CO$_2$ gas | 0 | 0.002 ± 0.000 | 186 |
| | $^{14}$C-free CO$_2$ gas | 0 | 0.003 ± 0.000 | 287 |
| | OXII | 1.3406 | 1.268 ± 0.013 | 45 |
| | OXII | 1.3406 | 1.270 ± 0.012 | 81 |
| | OXII | 1.3406 | 1.280 ± 0.011 | 96 |
| | OXII | 1.3406 | 1.305 ± 0.010 | 128 |
| | OXII | 1.3406 | 1.337 ± 0.010 | 162 |
| | OXII | 1.3406 | 1.306 ± 0.006 | 214 |
| | OXII | 1.3406 | 1.311 ± 0.005 | 321 |

[a] For combustion processes, two sets of standard material: the oxalic acid HOxII and anthracite with known $^{14}$C contents (F$^{14}$C = 1.3406 and F$^{14}$C = 0, respectively) were combusted using the aerosol combustion system  and used for quality control;

[revised manuscript text omitted]